# Accelerated Federated Optimization with Quantization

**Yeojoon Youn**
Georgia Institute of Technology
Atlanta, GA 30332
yjyoun92@gatech.edu

**Bhuvesh Kumar**
Georgia Institute of Technology
Atlanta, GA 30332
bhuvesh@gatech.edu

**Jacob Abernethy**
Georgia Institute of Technology
Atlanta, GA 30332
prof@gatech.edu

## Abstract

Federated optimization is a new form of distributed training on very large datasets that leverages many devices each containing local data. While decentralized computation can lead to significant speed-ups due to parallelization, some centralization is still required: devices must aggregate their parameter updates through synchronization across the network. The potential for communication bottleneck is significant. The two main methods to tackle this issue are (a) smarter optimization that decreases the frequency of communication rounds and (b) using *compression* techniques such as quantization and sparsification to reduce the number of bits machines need to transmit. In this paper, we provide a novel algorithm, **Fed**erated optimization algorithm with **A**cceleration and **Q**uantization (FedAQ), with improved theoretical guarantees by combining an accelerated method of federated averaging, reducing the number of training and synchronization steps, with an efficient quantization scheme that significantly reduces communication complexity. We show that in a homogeneous strongly convex setting, FedAQ achieves a linear speedup in the number of workers $M$ with only $\tilde{\mathcal{O}}(M^{\frac{1}{3}})$ communication rounds, significantly smaller than what is required by other quantization-based federated optimization algorithms. Moreover, we empirically verify that our algorithm performs better than current methods.

## 1 Introduction

Federated learning (FL) has attracted much attention from both academia and industry due to the increasing demand for large-scale distributed machine learning systems and preserving privacy-sensitive data on local devices such as smartphones and IoT devices. In federated learning, a number of clients collaboratively learn the global objective function by communicating with a central server without sharing any locally stored data in each local device. The research in Federated learning has identified four major challenges: communication efficiency, systems heterogeneity, statistical heterogeneity, and privacy (Li et al., 2020a). In this paper, we focus on communication efficiency that is of primary interest in cross-device settings when there is a heavy communication burden with many edge computing devices and limited network bandwidth. Two of the most widely used methods to reduce the communication cost are federated averaging optimization and randomized compression techniques.

In federated averaging (FedAvg) (McMahan et al., 2017), also called *local SGD*, each client locally updates its model with multiple Stochastic gradient descent (SGD) steps, and a server aggregates

Workshop on Federated Learning: Recent Advances and New Challenges, in Conjunction with NeurIPS 2022 (FL-NeurIPS'22). This workshop does not have official proceedings and this paper is non-archival.

| Algorithm | Convergence rate | Communication rounds for $\tilde{\mathcal{O}}(\frac{1}{T})$ convergence with linear speedup | Bits communicated for linear speedup |
|---|---|---|---|
| Reisizadeh et al. (2020) | $\mathcal{O}(\frac{1+q}{K} + \frac{T}{K^2})$ | Not possible | Not possible |
| Haddadpour et al. (2021) | $\tilde{\mathcal{O}}(\frac{1+q}{MT} + \frac{1}{TK})$ | $\tilde{\mathcal{O}}(\frac{M}{1+q})$ | $\tilde{\mathcal{O}}(\frac{M}{1+q}) \cdot d_{\text{quant}}$ |
| Yuan and Ma (2020) | $\tilde{\mathcal{O}}(\frac{1}{MT} + \frac{1}{TK^3})$ | $\tilde{\mathcal{O}}(M^{\frac{1}{3}})$ | $\tilde{\mathcal{O}}(M^{\frac{1}{3}}) \cdot 2d_{\text{full}}$ |
| **FedAQ** (corollary F.8) | $\tilde{\mathcal{O}}(\frac{1+q}{MT} + \frac{1+q}{TK^3})$ | $\tilde{\mathcal{O}}(M^{\frac{1}{3}})$ | $\tilde{\mathcal{O}}(M^{\frac{1}{3}}) \cdot 2d_{\text{quant}}$ |

Table 1: Summary of results on the convergence rate and communication required for linear speedup. $M$ is the number of devices, $T$ is the number of total parallel iterations, and $K$ is the number of communication rounds, $q$ is a quantization parameter (assumption 2.1), $d_{\text{quant}}$ is the number of bits used to quantize, $d_{\text{full}}$ is the number of bits required when there is no quantization ($d_{\text{full}} \gg d_{\text{quant}}$). Yuan and Ma (2020) and FedAQ send two iterates per communication round as other algorithms to achieve acceleration (See line 11 in Algorithm 1), we multiply $d_{\text{full}}$ and $d_{\text{quant}}$ by 2 for bits communicated for a linear speedup. The presented results of Haddadpour et al. (2021) are newly obtained (app. E).

model updates of clients. The server updates its own model parameters by averaging client models and then broadcasts the server parameters to all clients. This enables FL systems to achieve high communication efficiency with infrequent synchronization while showing better performance than distributed large mini-batch SGD (Lin et al., 2018). Due to the significant empirical success of FedAvg, researchers have proposed an interesting theoretical question: To what extent can we minimize the number of synchronizations in order to both guarantee convergence and achieve linear speedup in the number of workers $M$[1]? For the strongly-convex and homogeneous settings, Khaled et al. (2020) was able to achieve a linear speedup in $M$ with $\tilde{\mathcal{O}}(M)$ communication rounds, which is the state-of-the-art result for FedAvg convergence analysis. However, even with this progress on theoretical guarantees of FedAvg, it remains unclear whether further improvements on convergence time and communication efficiency can be achieved.

Applying acceleration methods to FL has led to improved convergence, with Yuan and Ma (2020) providing a faster version of FedAvg with provably stronger bounds. For the strongly-convex and homogeneous setting, their algorithm achieves a linear speedup in $M$ with only $\tilde{\mathcal{O}}(M^{\frac{1}{3}})$ communication rounds. Hence, the accelerated version of federated averaging requires a much smaller number of communication rounds than FedAvg to achieve the same accuracy. At present, this remains the best result for strongly-convex and homogeneous local data distribution settings. In addition to reducing the required number of communication rounds, another powerful way to build communication-efficient FL systems is to reduce the number of bits that need to be transmitted at each synchronization. Reisizadeh et al. (2020); Haddadpour et al. (2021) have shown that such compression techniques, which include *quantization*, reduce communication costs and guarantee convergence (See Table 1). More related works can be found in Appx. A

In this work, we provide a novel algorithm, **Fed**erated optimization algorithm with **A**cceleration and **Q**uantization (FedAQ), to solve the severe communication bottleneck problem in FL systems. FedAQ is the first federated optimization algorithm that successfully incorporates *multiple local update schemes*, *acceleration*, and *quantization* for master-worker topology. Although these three key desiderata of Federal Learning systems have individually been shown to build communication-efficient FL systems, it is not obvious if or how acceleration techniques can lead to faster convergence even for quantization based methods. We answer this question by showing that FedAQ converges for strongly-convex and homogeneous local data distribution settings without any additional strong assumptions.

Let $T$ be the number of total parallel iterations, $K$ be the number of total communication rounds. We compare our results to previous methods in Table 1, and highlight the following contributions:

1. FedAQ has a convergence rate of $\tilde{\mathcal{O}}(\frac{1+q}{MT} + \frac{1+q}{TK^3})$ which is better than the $\tilde{\mathcal{O}}(\frac{1+q}{MT} + \frac{1}{TK})$ convergence of Haddadpour et al. (2021), the state of the art in quantization based methods.

---

[1]Linear speedup in the number of workers is a desirable property in parallel computing which implies that the task takes half as much time if the number of workers are doubled.

Here $q$ is a parameter that measures the effectiveness of the quantization scheme (see assumption 2.1). This allows FedAQ to obtain linear speedup with only $\tilde{\mathcal{O}}(M^{\frac{1}{3}})$ communication rounds whereas Haddadpour et al. (2021) requires $\tilde{\mathcal{O}}(\frac{M}{1+q})$ rounds. The faster convergence in number of communication rounds also implies that FedAQ can achieve better convergence than Haddadpour et al. (2021) by using much fewer communication rounds. That is, while the per-round communication costs are the same, FedAQ requires much less communication overall due to the reduction in synchronization rounds.

2. When comparing FedAQ to Accelerated Federated learning, we observe that FedAQ has similar convergence and requires the same number of communication rounds as Yuan and Ma (2020). In each communication round of Yuan and Ma (2020), every client sends the complete iterates to the server without any quantization. To effectively obtain a convergence rate of $\tilde{\mathcal{O}}(\frac{1}{MT})$, it needs to send each value with a precision of $\tilde{\mathcal{O}}(\frac{1}{MT})$, requiring $d_{\text{full}} = \mathcal{O}(\log{(MT)})$ bits. In comparison, if we use the low precision quantizer (Appx. E.3) given by Alistarh et al. (2017), FedAQ needs to send only $d_{\text{quant}} = O(\log{\frac{1}{q}})$ bits [2] for each value. Since $q$ is a constant, $d_{\text{quant}} \ll d_{\text{full}}$. The extra $1+q$ term in the convergence for FedAQ can be offset by scaling the number of local updates by $1+q$, which is cheaper than expensive data communication. Thus, FedAQ can obtain the same convergence as Yuan and Ma (2020) using the same number of communication rounds but by sending much fewer bits in each round.

Finally, we empirically verify that our algorithm exhibits better performance than baselines, FedPAQ (Reisizadeh et al., 2020), FedCOMGATE (Haddadpour et al., 2021), FedAC (Yuan and Ma, 2020), and FedAvg (McMahan et al., 2017) on classical vision datasets such as MNIST (LeCun, 1998) and CIFAR-10 (Krizhevsky et al., 2009).

## 2 Problem setup

In this paper, we build our algorithm based on federated learning with captain-worker topology where $M$ local devices contain their own local data, and a server aggregates local parameter updates without sharing any data during synchronization rounds. Since we focus on *homogeneous* local data distribution settings for the convergence analysis of our algorithm, we define the distributed stochastic optimization problem as below.

$$\min_{w\in\mathbb{R}^d} F(w) := \mathbb{E}_{z\sim\mathcal{D}}[f(w;z)]$$

In our convergence analysis, we assume $F$ is *strongly-convex*. Each client can access $F$ at $w$ via oracle $\nabla f(w;z)$ because all clients have the same loss function $f$. Also, every local device has the same local data distribution $\mathcal{D}$. Moreover, we use the *full participation* of nodes for local updates and synchronizations.

### 2.1 Assumptions

Let us clarify assumptions on the unbiased quantizer $Q$, the global objective function $F$, and the unbiased gradient estimator $\nabla f$.

**Assumption 2.1.** The variance of the unbiased quantizer $Q$ is bounded by the squared of $l_2$-norm of its argument, i.e., $\mathbb{E}[Q(x)|x] = x$, $\mathbb{E}[\|Q(x)-x\|^2|x] \leq q\|x\|^2$.

For example, a well-known randomized quantizer which satisfies Assumption 2.1 is low-precision quantizer (Appx. E.3) in Alistarh et al. (2017).

**Assumption 2.2.** F is $\mu$-strongly convex, i.e., $F(w_1) \geq F(w_2) + \langle\nabla F(w_2), w_1 - w_2\rangle + \frac{1}{2}\mu\|w_1 - w_2\|^2$ for any $w_1, w_2 \in \mathbb{R}^d$.

**Assumption 2.3.** F is L-smooth, i.e., $F(w_1) \leq F(w_2) + \langle\nabla F(w_2), w_1 - w_2\rangle + \frac{1}{2}L\|w_1 - w_2\|^2$ for any $w_1, w_2 \in \mathbb{R}^d$.

**Assumption 2.4.** $\nabla f(w;\xi)$ is unbiased and variance bounded, i.e., $\mathbb{E}_\xi[\nabla f(w;\xi)] = \nabla F(w)$, $\mathbb{E}_\xi[\|\nabla f(w;\xi) - \nabla F(w)\|^2] \leq \sigma^2$ for any $w \in \mathbb{R}^d$.

---

[2] More details on this are discussed in appendix E.3

## 2.2 Notation

We use $\tau, K$ to respectively denote the number of local updates and total communication rounds, which means the total number of iterations $T$ at each node satisfies $T = K\tau$. Since we consider a strongly-convex case, we can find the optimal point $w^*$ and denote the optimal function value as $F^* := F(w^*)$. The local parameter $w_{k,t}^m$ indicates the parameter of the $m$-th local model after $k$th synchronization followed by $t$ local SGD updates. There are other types of parameters such as $w_{k,t}^{\mathrm{ag},m}$ and $w_{k,t}^{\mathrm{md},m}$, and we obtain two types of parameters $w_k$ and $w_k^{\mathrm{ag}}$ in the server side after $k$th synchronization. More details on these parameters will be discussed in the next section.

# 3 FedAQ algorithm

We propose a novel communication efficient algorithm that combines an accelerated variant of federated averaging and an efficient quantization scheme. Our FedAQ algorithm has two main parts: (1) multiple accelerated local updates and (2) communication with quantization. Both components contribute to achieving better communication efficiency than other previous federated algorithms. The entire process is summarized in Algorithm 1 (See Appx. B).

## 3.1 Multiple accelerated local updates

The FedAvg algorithm, proposed by McMahan et al. (2017), is widely used for federated learning to improve communication efficiency by reducing communication rounds with multiple local SGD updates. Yuan and Ma (2020) provide FedAC that replaces the stochastic gradient updates of FedAVG by accelerated version of SGD by Ghadimi and Lan (2012) resulting in a linear speedup in $M$ with smaller communication rounds than FedAvg.

Thus, we apply the FedAC scheme to multiple updates of each local model. Since previous quantization-based federated optimization algorithms are FedAvg variants with no acceleration, the accelerated method enables our algorithm to gain better communication efficiency than others.

As you can see in Algorithm 1, we need two more local parameters $w_{k,t}^{\mathrm{ag},m}$ and $w_{k,t}^{\mathrm{md},m}$ for acceleration in addition to the main local parameter $w_{k,t}^m$. $w_{k,t}^{\mathrm{ag},m}$ aggregates the past iterates, and the gradients are queried at the auxiliary parameter $w_{k,t}^{\mathrm{md},m}$. While typical FL algorithms without acceleration only have a learning rate $\eta$ as their hyperparameter, the general acceleration scheme makes our algorithm flexible due to four hyperparameters $\alpha, \beta, \eta, \gamma$. The flexibility of hyperparameters enables the fast convergence speed of FedAQ, but naively chosen hyperparameters also cause unstable training of FedAQ. We discuss the exact choice of hyperparameters in Appx. B.1. Unlike FedAC, that requires each client to communicate the exact iterates to the server with high precision, we discuss in the following subsection how FedAQ incorporates quantization techniques to reduce communication cost.

## 3.2 Communication with quantization

In cross-device federated learning, a large amount of communicated messages from a number of devices and the limited communication bandwidth can lead to severe communication bottleneck. Therefore, in this scenario, an efficient quantization scheme can significantly reduce the size of communicated messages and make communication between local devices and a server faster. We apply the same unbiased quantizer used in Haddadpour et al. (2021) that satisfies Assumption 2.1.

In contrast with other quantization-based federated optimization algorithms (Reisizadeh et al., 2020; Haddadpour et al., 2021), FedAQ requires quantizing two weight updates for each local device during a synchronization step because two weight updates should be sent from local devices to a server in the FedAC framework. To be specific, after each client $m$ obtains $w_{k,\tau}^m, w_{k,\tau}^{\mathrm{ag},m}$ through $\tau$ accelerated local iterations, each client quantizes the difference between $w_{k,\tau}^m, w_{k,\tau}^{\mathrm{ag},m}$ and the most recent server models $w_k, w_k^{\mathrm{ag}}$. Then, a server aggregates $Q(w_{k,\tau}^m - w_k), Q(w_{k,\tau}^{\mathrm{ag},m} - w_k^{\mathrm{ag}})$ from all clients. After dequantizing those messages, the server obtains the following new models $w_{k+1}, w_{k+1}^{\mathrm{ag}}$

and broadcasts them back to each client.

$$w_{k+1} = w_k + \frac{1}{M}\sum_{m=1}^{M} Q(w_{k,\tau}^m - w_k), \ w_{k+1}^{\mathrm{ag}} = w_k^{\mathrm{ag}} + \frac{1}{M}\sum_{m=1}^{M} Q(w_{k,\tau}^{\mathrm{ag},m} - w_k^{\mathrm{ag}})$$

## 4 Convergence analysis

We analyze the FedAQ algorithm under assumptions mentioned in Section 2. In Appx. B.1, we define two condition sets of hyperparameters that ensure the convergence guarantees of FedAQ. FedAQ shows the better performance in experiments (See Strongly convex case in Section 5.1) under the parameter condition set (1) than the parameter condition set (2), while the parameter condition set (2) leads to better convergence rate. Thus, in this section, we provide the convergence analysis of FedAQ under the parameter condition set (2) that leads to the better convergence rate $\tilde{\mathcal{O}}(\frac{1+q}{MT} + \frac{1+q}{TK^3})$. The full proofs of lemmas, theorems, and corollaries under two parameter condition sets are elaborated in Appx. D (condition set (1)) and Appx. F (condition set (2)).

The decentralized potential $\Phi_{k,t}$ (Yuan and Ma, 2020) is used for our convergence analysis. People commonly use this potential for acceleration analysis (Bansal and Gupta, 2019).

$$\Phi_{k,t} = F(\bar{w}_{k,t}^{\mathrm{ag}}) - F^* + \frac{1}{6}\mu\|\bar{w}_{k,t} - w^*\|^2$$

$\bar{w}_{k,t}$ and $\bar{w}_{k,t}^{\mathrm{ag}}$ is respectively the average of $w_{k,t}^m$ and $w_{k,t}^{\mathrm{ag,\,m}}$ for all $m$. Here, we additionally define $\Phi_k$ as below.

$$\Phi_k := \Phi_{k,0} = F(w_k^{\mathrm{ag}}) - F^* + \frac{1}{6}\mu\|w_k - w^*\|^2$$

Since $w_k$ and $w_k^{\mathrm{ag}}$ are parameters obtained after $k$th synchronization in a server side, $\Phi_k$ can be considered as the potential of server models. $\Phi_k$ is essential to show the convergence of FedAQ because there is the computation of the quantizer between $\Phi_{k-1,\tau}$ and $\Phi_{k,0}$. Thus, we should not naively track $\Phi_{k,t}$ but track $\Phi_k$ for our analysis. Obtaining $\Phi_k \leq \epsilon$ would imply that $F(w_k^{\mathrm{ag}}) - F^* \leq \epsilon$ and since $F^* \leq F(w_k^{\mathrm{ag}})$, it would also imply that $\|w_k - w^*\|^2 = O(\epsilon)$, thus obtaining convergence in terms of both the objective value and the iterate.

Our goal is to show the convergence of FedAQ and derive the simplified convergence rate so that we can get the number of communication rounds to achieve a linear speedup in $M$. When it comes to a high-level proof sketch, there are three big steps: the relationship between two consecutive potential functions($\Phi_k, \Phi_{k+1}$) at a server-side (Lemma F.1), how $\Phi_K$ decreases from the initial potential $\Phi_0$ as a communication round $K$ increases (Theorem F.7), obtaining an intuitive convergence rate to analyze a linear speedup by tuning a local learning rate $\eta$ properly in Theorem F.7 (Corollary F.8). In Lemma F.1, we get the inequality between $\Phi_k$ and $\Phi_{k+1}$ by finding the upper bounds of error terms due to multiple($\tau$) local steps and the quantization step. The upper bound of the error caused by multiple local steps is obtained with the help of the analysis in Yuan and Ma (2020) (See Proposition F.3). How we find the upper bound of the error due to quantization and why this part is challenging is explained in Appx. F.4. Next, we introduce the simplified form of Corollary F.8 to analyze a linear speedup in $M$.

**Corollary 4.1.** *(Simplified form of Corollary F.8) Note that $T = K\tau$. For $\eta = \min(\frac{1}{L}, \tilde{\Theta}(\frac{\tau}{\mu T^2}))$, FedAQ yields*

$$\mathbb{E}[\Phi_K] \leq \min\left(\exp(-\frac{\mu T}{6L}), \exp(-\frac{\mu^{\frac{1}{2}}T}{6L^{\frac{1}{2}}\tau^{\frac{1}{2}}})\right)\Phi_0 + \tilde{\mathcal{O}}(\underbrace{\frac{(1+q)\sigma^2}{\mu MT}}_{I} + \underbrace{\frac{(1+q)L^2\tau^3\sigma^2}{\mu^3 T^4}}_{II} + \underbrace{\frac{qL^3\tau^2\sigma^2}{\mu^4 MT^3}}_{III})$$

**Remark 4.2.** The above convergence rate is worse than the convergence rate of FedAC-II according to Theorem C.13 in Yuan and Ma (2020) because there are additive terms related to the quantization noise $q$ in our case. Let's figure out the dominant terms with $\tilde{\mathcal{O}}$ notation from the above convergence rate. Here, we replace $\tau$ with $\frac{T}{K}$. At first, we can ignore the first term because it decreases exponentially. The second term I would be $\tilde{\mathcal{O}}(\frac{1+q}{MT})$. Then, the third term II becomes $\tilde{\mathcal{O}}(\frac{(1+q)\tau^3}{T^4}) = \tilde{\mathcal{O}}(\frac{1+q}{TK^3})$. Finally, the last term III turns into $\tilde{\mathcal{O}}(\frac{q\tau^2}{MT^3}) = \tilde{\mathcal{O}}(\frac{q}{MTK^2})$. Thus, the overall convergence rate of

FedAQ under the condition set (2) would be $\tilde{\mathcal{O}}(\frac{1+q}{MT} + \frac{1+q}{TK^3})$. Similarly, we obtain the simplified convergence rate of FedAQ under the condition set (1) from three terms (13), (14), (15) of Corollary D.8. In this case, the convergence rate of FedAQ is $\tilde{\mathcal{O}}(\frac{1+q}{MT} + \frac{1}{TK^2})$, and the required number of communication rounds to achieve a linear speedup in $M$ is $\tilde{\mathcal{O}}((\frac{M}{1+q})^{\frac{1}{2}})$.

## 5 Experiments

In this section, we provide experimental results of FedAQ in homogeneous local data distribution settings. We compare FedAQ with other quantization-based federated optimization algorithms, FedPAQ (Reisizadeh et al., 2020) and FedCOMGATE (Haddadpour et al., 2021). FedAvg (McMahan et al., 2017) and FedAC (Yuan and Ma, 2020), federated optimization algorithms without quantization, are also our baselines. We empirically validate the performance of 5 algorithms on classical classification tasks on MNIST(LeCun, 1998) and CIFAR-10(Krizhevsky et al., 2009) datasets in the distributed learning environment. We consider three objective functions i) A strongly convex objective of $l_2$-regularized logistic regression model on the MNIST dataset, ii) A non convex objective of training a multilayer perceptron on the MNIST data, and iii) A non convex objective of training a convolution neural network (CNN) on the CIFAR-10 dataset. The details of the implementation environment, datasets, training models, hyperparameter choices, quantization bits, and new time metric are elaborated in Appx. C.1.

### 5.1 Experimental results

In our experiments on both MNIST and CIFAR-10, we verify how the global training loss and test accuracy of five algorithms change with respect to communication rounds, the number of bits communicated between one client and the server during the uplink, and human time. We consider both computation time and communication time to estimate human time defined by the new time metric (See Appx. C.1). All MNIST experimental plots and CIFAR-10 plots can be found in Appx. C.3. Also, we provide quantitative results for plots in Appx. C.3.4.

**Strongly convex case.** In this experiment, we compare FedAQ under the condition set (1) and set (2) with FedAvg, FedPAQ, FedCOMGATE, and FedAC-I. We denote each FedAQ as FedAQ-I and FedAQ-II. As we observe the theoretical benefits of FedAQ over other methods in Section 4, FedAQ-I outperforms all other quantization-based federated optimization algorithms and FedAC-I in all plots (See Figure 1). However, although FedAQ-II shows the fast convergence speed, the training process is unstable. Thus, we only use FedAQ-I for further non-convex experiments. FedAC and FedAQ in non-convex experiments indicate FedAC-I and FedAQ-I.

**Non-convex case.** Figure 2, 3 clearly demonstrates that FedAQ with 4 bits quantization outperforms other algorithms in all plots. In terms of communication rounds, accelerated algorithms, FedAQ and FedAC, converge faster than other algorithms. We also observe that quantization does not lead to slower convergence, which means we can apply an efficient quantization scheme to make communication efficient FL systems without sacrificing convergence speed. The plots related to communicated bits are helpful to interpret how algorithms work well in situations with heavy communication. FedAQ with 8 bits quantization shows comparable performance relative to FedPAQ and FedCOMGATE with the help of acceleration, even though FedAQ sends more updates during every synchronization. When we use 4 bits quantization for FedAQ to make the number of communicated bits the same for all quantization-based algorithms during synchronization, FedAQ shows a much faster convergence speed with regard to the number of communicated bits. However, plots of communicated bits fail to reflect how algorithms converge in real estimated time for FL scenarios, which consists of both communication and computation. Thus, we further analyze algorithms with human time. We observe that FedAQ with 8 quantization bits performs slightly better than FedPAQ and FedCOMGATE for both MNIST and CIFAR-10. This occurs because while all quantization-based algorithms send the same number of communicated bits, the number of communication rounds for FedAQ is much smaller than others. Then, this also indicates that FedAQ takes less computation time than other methods while reaching the same accuracy.

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

# A  Related works

The practical usefulness of federated learning arouses researchers' curiosity why FedAvg, the simplest form of federated optimization algorithms indispensable for communication efficient distributed training, works well in the real world. The first convergence guarantee that FedAvg converges at the same rate as mini-batch SGD on IID local data distribution in strongly convex scenarios is proposed by Stich (2018). The further convergence analysis of FedAvg for non-convex functions is done by several works (Wang and Joshi, 2018; Haddadpour et al., 2019b; Yu et al., 2019b). Wang and Joshi (2018); Stich and Karimireddy (2019); Haddadpour et al. (2019a); Khaled et al. (2020); Woodworth et al. (2020) remove unnecessary assumptions such as uniformly bounded gradients and achieve better convergence rate. Moreover, Li et al. (2018); Haddadpour and Mahdavi (2019); Li et al. (2019); Khaled et al. (2020); Karimireddy et al. (2020b) individually define the quantity of non-IID and analyze the convergence of FedAvg and its variants in heterogeneous settings.

Reducing the transmitted bits between a server and clients through compression techniques is pivotal to saving communication costs in federated learning. This motivates researchers to develop various compression techniques such as sparsification and quantization without significantly sacrificing accuracy (Konečný et al., 2016; Alistarh et al., 2017; Suresh et al., 2017; Wangni et al., 2017; Bernstein et al., 2018; Wang et al., 2018; Vogels et al., 2019; Horvath et al., 2019; Basu et al., 2019; Rothchild et al., 2020). Reisizadeh et al. (2020) show near-optimal theoretical guarantees of the first federated optimization algorithm that incorporates federated averaging, partial node participation and quantization in homogeneous local data distribution settings. Haddadpour et al. (2021) further provide improved convergence rates for both homogeneous and heterogeneous settings.

Furthermore, we can achieve better communication efficiency by applying acceleration methods into client updates. Yuan and Ma (2020) proposes the first provable acceleration of FedAvg that achieves a linear speedup with the smallest communication rounds. Several other works aim to achieve communication efficiency by using momentum or adaptive optimizers (Yu et al., 2019a; Karimireddy et al., 2020a; Wang et al., 2021b). Our work is not the first to combine acceleration and quantization. Li et al. (2020b); Li and Richtárik (2021) propose compressed and accelerated gradient distributed optimization methods that are neither stochastic nor FedAvg variants. Singh et al. (2021) propose communication efficient momentum SGD for decentralized optimization with the first theoretical analysis. However, to the best of our knowledge, FedAQ is the first accelerated version of federated averaging for master-worker topology that successfully integrates a quantization scheme and provides rigorous convergence guarantees.

# B  Algorithm details

---

**Algorithm 1** Federated Accelerated SGD with Quantization (FedAQ)

---

1: **Input:** $\alpha, \beta, \eta, \gamma$, initial vector $w_0 = w_{0,0}^{\mathrm{ag},m} = w_{0,0}^m$ for all devices $m \in [M]$
2: **for** $k = 0, \cdots, K-1$ **do**
3:    **for** each client $m$ in parallel **do**
4:       $w_{k,0}^m \leftarrow w_k, w_{k,0}^{\mathrm{ag},m} \leftarrow w_k^{\mathrm{ag}}$
5:       **for** $t = 0, \cdots, \tau - 1$ **do**
6:          $w_{k,t}^{\mathrm{md},m} \leftarrow \beta^{-1} w_{k,t}^m + (1 - \beta^{-1}) w_{k,t}^{\mathrm{ag},m}$
7:          $g_{k,t}^m \leftarrow \nabla f(w_{k,t}^{\mathrm{md},m}, \xi_{k,t}^m)$
8:          $w_{k,t+1}^{\mathrm{ag},m} \leftarrow w_{k,t}^{\mathrm{md},m} - \eta g_{k,t}^m$
9:          $w_{k,t+1}^m \leftarrow (1 - \alpha^{-1}) w_{k,t}^m + \alpha^{-1} w_{k,t}^{\mathrm{md},m} - \gamma g_{k,t}^m$
10:       **end for**
11:       send $Q(w_{k,\tau}^m - w_k), Q(w_{k,\tau}^{\mathrm{ag},m} - w_k^{\mathrm{ag}})$
12:    **end for**
13:    server finds $w_{k+1} \leftarrow w_k + \frac{1}{M} \sum\limits_{m=1}^{M} Q(w_{k,\tau}^m - w_k), \ w_{k+1}^{\mathrm{ag}} \leftarrow w_k^{\mathrm{ag}} + \frac{1}{M} \sum\limits_{m=1}^{M} Q(w_{k,\tau}^{\mathrm{ag},m} - w_k^{\mathrm{ag}})$
14: **end for**

---

## B.1 Two parameter condition sets

We carefully determine two parameter condition sets that theoretically ensure the convergence guarantees. The first one is

$$\eta, \gamma \in \left(0, \frac{1}{L}\right], \gamma = \max\left(\sqrt{\frac{\eta}{\mu\tau}}, \eta\right), \alpha = \frac{1}{\gamma\mu}, \beta = \alpha + 1 \tag{1}$$

We add one more condition $\gamma \in (0, \frac{1}{L}]$ to the FedAC-I condition (Yuan and Ma, 2020) and create our parameter condition set (1). The second one is

$$\eta, \gamma \in \left(0, \frac{1}{L}\right], \gamma = \max\left(\sqrt{\frac{\eta}{\mu\tau}}, \eta\right), \alpha = \frac{3}{2\gamma\mu} - \frac{1}{2}, \beta = \frac{2\alpha^2 - 1}{\alpha - 1}, \gamma\mu \le \frac{3}{4} \tag{2}$$

We add two more conditions $\gamma \in (0, \frac{1}{L}]$ and $\gamma\mu \le \frac{3}{4}$ to the FedAC-II condition to build our parameter condition set (2). Even though quantization adds complexity to the algorithm, these weak assumptions are the only additional requirements for showing the convergence of FedAQ. Moreover, although the better convergence rate $\tilde{\mathcal{O}}(\frac{1+q}{MT} + \frac{1+q}{TK^3})$ is obtained from the condition set (2), we also analyze the convergence of FedAQ under the condition set (1) because this set empirically leads to more stable training and better performance in experiments than the condition set (2) (See Strongly convex case in Section 5.1).

# C  More details and results about experiments

## C.1  Experimental setup details

**Implementation environment.**  We follow the implementation setup in Haddadpour et al. (2021). We use the Distributed library of PyTorch to implement our algorithm because this library allows us to simulate real-world communication and distributed training. The 18 cores of Intel Xeon E5-2676 CPU are used as computing sources. Each core is considered as one local client. We use 16 cores for strongly convex MNIST, 18 cores for the non-convex MNIST, and 8 cores for the CIFAR-10. For MNIST, the strongly convex experiment and the non-convex one respectively run for 300 rounds of communication with 20 local updates and 50 rounds of communication with 100 local updates. The CIFAR-10 experiment runs for 100 rounds of communication with 100 local updates.

**Datasets.**  For image classification tasks, we choose two main classical image datasets: MNIST and CIFAR-10. Since we assume homogeneous settings, data is distributed homogeneously among clients, which also means each device has access to all 10 classes.

**Training models.**  For MNIST, we use a $l_2$-regularized logistic regression model for the strongly convex case and a multilayer perceptron (MLP) with two hidden layers for the non-convex case. For CIFAR-10, we use a Convolutional Neural Network (CNN). Here, we note that the number of parameters in a neural network model is directly related to the number of communicated bits. We discuss more on this in Appx. C.2.

**Hyperparameter choice.**  The important hyperparmeters in our experiments are learning rates for each algorithm. For the client learning rate $\eta$, we respectively use 0.002, 0.1, and 0.01 for strongly convex MNIST, non-convex MNIST, and CIFAR-10 for all algorithms. For FedAQ and FedAC, once we set the value of $\mu$, other hyperparameters ($\gamma, \alpha, \beta$) are automatically determined (See condition set (1) and (2)). Thus, we choose 0.1, 0.01, and 0.2 for $\mu$ value for strongly convex MNIST, non-convex MNIST, and CIFAR-10. Since too large $\mu$ leads to slow convergence and too small $\mu$ leads to unstable training, we get these $\mu$ values by tuning $\mu$ appropriately. FedCOMGATE has a server learning rate, and we set this value as 1 for all experiments.

**Quantization bits.**  We have three quantization-based federated algorithms: FedAQ, FedPAQ, FedCOMGATE. We quantize the updates from 32 bits to 8 bits for all quantization-based algorithms in both MNIST and CIFAR-10. Additionally, particularly for FedAQ in non-convex experiments, we consider 4 bits quantization as well. Since FedAQ sends twice as many messages as FedPAQ or FedCOMGATE at every synchronization when we use 8 bits quantization for all cases, we apply 4 bits quantization to FedAQ to let FedAQ send the same amount of information in each communication round as other quantization-based algorithms for a fair comparison.

**New time metric.** In our experiments, communication between CPU cores is very fast, so it is hard to say that the environment of our experiments fully reflects the real-world federated learning when there is a heavy communication burden. Thus, we use a linear model to estimate the execution time $T_{\text{round}}(\mathcal{A})$ between two consecutive communication rounds for real federated learning scenarios (Wang et al., 2021a).

$$T_{\text{round}}(\mathcal{A}) = T_{\text{comm}}(\mathcal{A}) + T_{\text{comp}}(\mathcal{A}), \qquad T_{\text{comm}}(\mathcal{A}) = \frac{S_{\text{down}(\mathcal{A})}}{B_{\text{down}}} + \frac{S_{\text{up}(\mathcal{A})}}{B_{\text{up}}}$$

$$T_{\text{comp}}(\mathcal{A}) = \max_j T^j_{\text{client}}(\mathcal{A}) + T_{\text{server}}(\mathcal{A}), \qquad T^j_{\text{client}}(\mathcal{A}) = R_{\text{comp}} T^j_{\text{sim}}(\mathcal{A}) + C_{\text{comp}}$$

Since $T_{\text{server}}(\mathcal{A})$ is relatively smaller than $T^j_{\text{client}}(\mathcal{A})$, we ignore $T_{\text{server}}(\mathcal{A})$ in our experiments. We get client download size $S_{\text{down}(\mathcal{A})}$ and upload size $S_{\text{up}(\mathcal{A})}$ from the number of neural network parameters. $\max_j T^j_{\text{sim}}(\mathcal{A})$ is the computation time in our simulation.

$$B_{\text{down}} \sim 0.75\text{MB/secs}, \; B_{\text{up}} \sim 0.25\text{B/secs}, \; R_{\text{comp}} \sim 7, \; C_{\text{comp}} \sim 10\text{secs}$$

Wang et al. (2021a) estimate each value of the above parameters from a real world cross-device FL system. The upload bandwidth $B_{\text{up}}$ is generally smaller than download bandwidth $B_{\text{down}}$. We define human time as the parallel time estimated by this new time metric.

## C.2 Neural network parameters & communicated bits

### C.2.1 MLP model for MNIST

As you can see in Appx. C.1 (Training models), we use a multilayer perceptron (MLP) with two hidden layers. Each hidden layer consists of 200 neurons with ReLU activations. Thus, we compute the total number of parameters in this MLP model as below.

$$
\begin{aligned}
(\text{\# of MLP parameters}) &= (\text{\# of input features}) \times (\text{\# of neurons in the 1st layer}) \\
&+ (\text{\# of neurons in the 1st layer}) \times (\text{\# of neurons in the 2nd layer}) \\
&+ (\text{\# of neurons in the 2nd layer}) \times (\text{\# of MNIST classes}) \\
&+ (\text{\# of neurons in the 1st layer}) + (\text{\# of neurons in the 2nd layer}) \\
&+ (\text{\# of MNIST classes}) \\
&= 28 \times 28 \times 200 + 200 \times 200 + 200 \times 10 + 200 + 200 + 10 = 199210
\end{aligned}
$$

Finally, we derive $S_{\text{up}}(\mathcal{A})(= S_{\text{down}}(\mathcal{A}))$, defined in Appx. C.1 (New time metric), by using the above fact. We use 32 bits floating-point if there is no quantization.

$$
\begin{aligned}
S_{\text{up}}(\mathcal{A}) &= (\text{\# of device}) \times (\text{\# of MLP parameters}) \times (\text{\# of bits}) \\
&= 18 \times 199210 \times 32 = 114744960
\end{aligned}
$$

The FedAvg algorithm follows the above calculation. If we use 8 bits quantization for FedPAQ, FedCOMGATE, and FedAQ, (# of bits) in the above equation will respectively be 8, 8, and 16. Since FedAQ sends twice as many messages as others at every communication round, (# of bits) for FedAQ is 16. Similarly, (# of bits) for FedAC, which has no quantization, is 64.

### C.2.2 CNN model for CIFAR-10

We use a CNN model, which consists of two 2-dimensional convolutional layers, two max pooling layers, and two fully connected layers. The ReLU activations are used in this CNN model. Let's clarify (# of input channel, # of output channel, kernel size, stride) for convolutional layers. We respectively use (3, 20, 5, 1), (20, 50, 5, 1) for the 1st and 2nd convolutional layer. Let's denote each convolutional layer and fully connected layer as CONV1, CONV2, FC3, FC4. At first, the activation shape of input layer for CIFAR-10 is (32, 32, 3). Then, we get the activation shape after CONV1 and

the number of parameters for CONV1.

$$\text{(width of activation shape)} = \frac{\text{(width of previous activation shape)} - \text{kernel size} + 1}{\text{stride}}$$

$$= \frac{32 - 5 + 1}{1} = 28 \Rightarrow \text{activation shape} = (28, 28, 20)$$

$$\text{(\# of CONV1 parameters)} = \Big(\text{kernel size} \times \text{kernel size}$$

$$\times \text{(\# of filters in the previous layer)} + 1\Big)$$

$$\times \text{(\# of filters in the current layer)}$$

$$= (5 \times 5 \times 3 + 1) \times 20 = 1520$$

The activation shape becomes (14, 14, 20) after max pooling. There are no learnable parameters in pooling layers. We do similar calculation for CONV2.

$$\text{(width of activation shape)} = \frac{\text{(width of previous activation shape)} - \text{kernel size} + 1}{\text{stride}}$$

$$= \frac{14 - 5 + 1}{1} = 10 \Rightarrow \text{activation shape} = (10, 10, 50)$$

$$\text{(\# of CONV2 parameters)} = \Big(\text{kernel size} \times \text{kernel size} \times \text{(\# of filters in the previous layer)}$$

$$+ 1\Big) \times \text{(\# of filters in the current layer)}$$

$$= (5 \times 5 \times 20 + 1) \times 50 = 25050$$

The activation shape becomes (5, 5, 50) after second max pooling. Then, we calculate the number of parameters in FC3 and FC4 similar to Appx. C.2.1.

$$\text{(\# of FC3 parameters )} = (5 \times 5 \times 50) \times 512 + 512 = 640512$$

$$\text{(\# of FC4 parameters )} = 512 \times 10 + 10 = 5130$$

Thus, the total number of parameters in this CNN model is

$$\text{(\# of CNN parameters)} = \text{(\# of CONV1 parameters)} + \text{(\# of CONV2 parameters)}$$

$$+ \text{(\# of FC3 parameters)} + \text{(\# of FC4 parameters)}$$

$$= 1520 + 25050 + 640512 + 5130 = 672212$$

Finally, we derive $S_{\text{up}}(\mathcal{A})(= S_{\text{down}}(\mathcal{A}))$ in this case.

$$S_{\text{up}}(\mathcal{A}) = \text{(\# of device)} \times \text{(\# of CNN parameters)} \times \text{(\# of bits)}$$

$$= 8 \times 672212 \times 32 = 172086272$$

We can do the similar discussion in Appx. C.2.1 when it comes to applying this to quantization-based federated optimization algorithms.

## C.3 Experimental plots

### C.3.1 Strongly convex case for MNIST

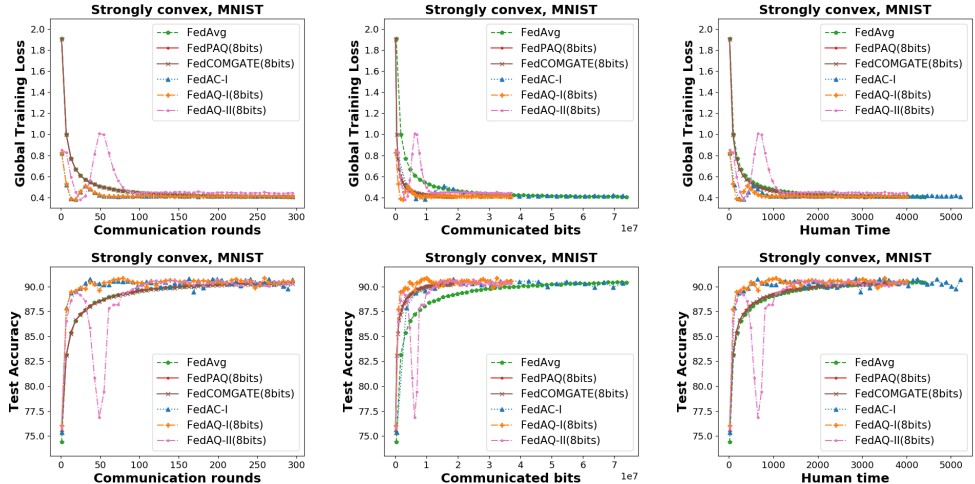

Figure 1: Comparing FedAQ with FedAvg, FedPAQ, FedCOMGATE, and FedAC-I on MNIST with strongly convex settings. We observe how the global training loss and test accuracy change across communication rounds (first column), communicated bits (second column), and human time (third column). We use a $l_2$-regularized logistic regression model for the strongly convex MNIST experiment. FedAQ-I outperforms other algorithms in all plots.

### C.3.2    Non-convex case for MNIST

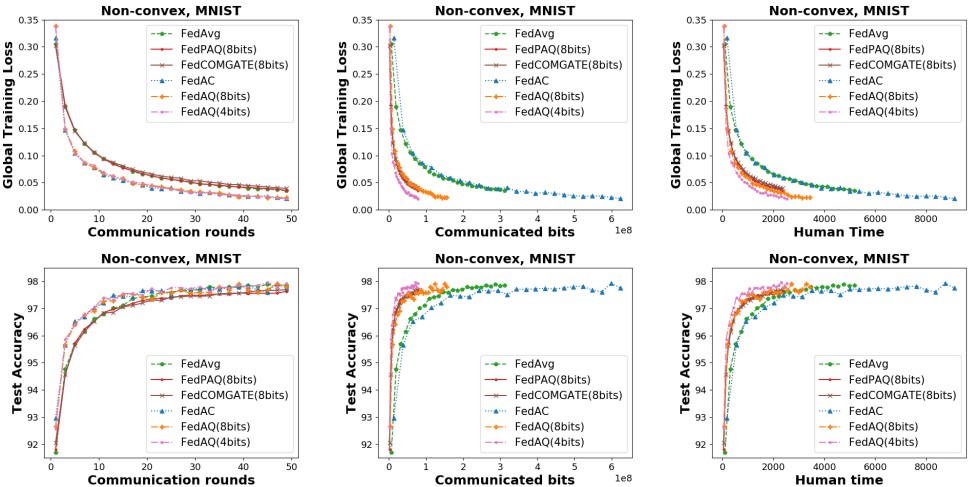

Figure 2: Comparing FedAQ with FedAvg, FedPAQ, FedCOMGATE, and FedAC on MNIST with non-convex settings. We observe how the global training loss and test accuracy change across communication rounds (first column), communicated bits (second column), and human time (third column). We use a MLP model for the non-convex MNIST experiment. FedAQ(4bits) sends the same number of communicated bits as FedPAQ(8bits) and FedCOMGATE(8bits) in each communication round, which indicates a fair comparison (See Quantization bits in Appx. C.1). FedAQ(4bits) outperforms other algorithms in all plots.

### C.3.3    Non-convex case for CIFAR-10

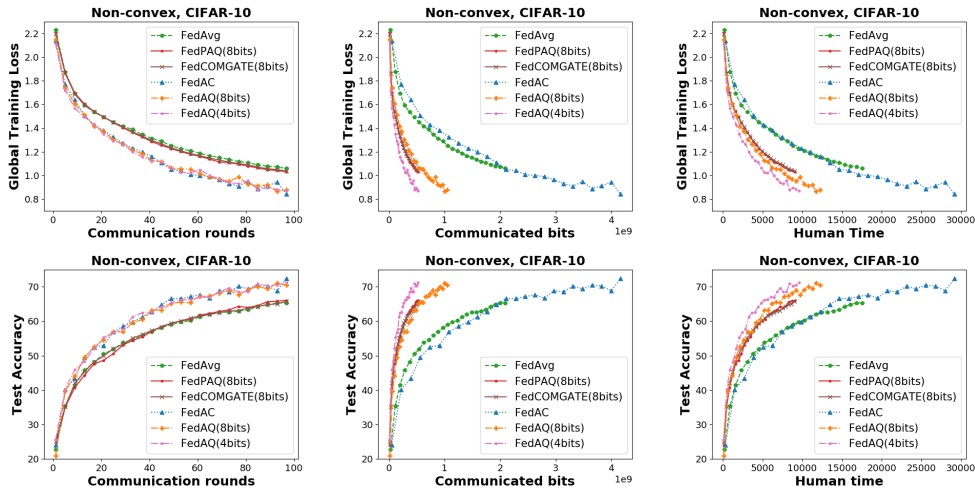

Figure 3: Comparing FedAQ with FedAvg, FedPAQ, FedCOMGATE, and FedAC on CIFAR-10. We observe how the global training loss and test accuracy change across communication rounds (first column), communicated bits (second column), and human time (third column). We use a CNN model for CIFAR-10. Similar to the MNIST experiment, FedAQ (4 bits) outperforms all other algorithms in every case.

#### C.3.4 Quantitative results for plots

We provide quantitative results to help readers understand plots better. To be specific, for all plots, we observe the number of communication rounds, the number of communicated bits, and the human time required to achieve a particular test accuracy by each federated optimization algorithm.

For the strongly convex experiment on MNIST (See Figure 1), the number of communication rounds required to achieve 90.28% test accuracy by FedAvg, FedPAQ(8bits), FedCOMGATE(8bits), FedAC-I, FedAQ-I(8bits), FedAQ-II(8bits) are respectively 217, 216, 260, 28, 26, 99. The number of communicated bits required to achieve the same accuracy are respectively 5.4e7, 1.4e7, 1.6e7, 1.4e7, 3.3e6, 1.2e7. Lastly, the required human time are respectively 3220s, 2760s, 3336s, 484s, 344s, 1323s. In this experiment, FedAQ-I(8bits) requires the smallest number of communication rounds, the smallest number of communicated bits, and the shortest human time to achieve the same test accuracy. These experimental results support the validity of our theoretical analysis on strongly convex cases.

For the non-convex experiment on MNIST (See Figure 2), the number of communication rounds required to achieve 97.6% test accuracy by FedAvg, FedPAQ(8bits), FedCOMGATE(8bits), FedAC, FedAQ(8bits), FedAQ(4bits) are respectively 23, 48, 38, 18, 18, 16. The number of communicated bits required to achieve the same accuracy are respectively 1.5e8, 7.6e7, 6.1e7, 2.3e8, 5.7e7, 2.5e7. Finally, the required human time are respectively 2424s, 2311s, 1834s, 3327s, 1248s, 805s. Thus, we conclude that FedAQ(4bits) outperforms other algorithms, and even FedAQ(8bits) needs smaller number of communicated bits/less human time to achieve the goal accuracy than Fed-PAQ(8bits)/FedCOMGATE(8bits).

For the non-convex experiment on CIFAR-10 (See Figure 3), the number of communication rounds required to achieve 65.4% test accuracy by FedAvg, FedPAQ(8bits), FedCOMGATE(8bits), FedAC, FedAQ(8bits), FedAQ(4bits) are respectively 98, 89, 95, 49, 50, 48. The number of communicated bits required to achieve the same accuracy are respectively 2.1e9, 4.8e8, 5.1e8, 2.1e9, 5.4e8, 2.6e8. Finally, the required human time are respectively 31798s, 11526s, 12240s, 28720s, 9902s, 6464s. As with the non-convex experiment on MNIST, FedAQ(4bits) outperforms other algorithms, and even FedAQ(8bits) requires less human time to achieve the same accuracy than FedPAQ(8bits)/FedCOMGATE(8bits).

**Remark C.1.** Our current experimental setup only allows us to scale the number of clients up to the number of CPU cores in our machine. Since FedAQ achieves linear speed up in the number of

workers with much fewer communication rounds than other quantization based methods, we expect FedAQ to outperform other methods by an even larger margin as we scale the number of workers.

### C.3.5 Additional experiment on linear speedup in terms of the number of clients

We do further experiments about a linear speedup that can validate our theoretical claims. We follow the experimental setup in Yuan and Ma (2020) for this experiment. We compare the performance of FedAQ with FedAvg, FedPAQ, and FedAC on $l_2$-regularized logistic regression for UCI a9a dataset (Dua and Graff, 2017). We set regularization strength as $10^{-3}$. In this experiment, FedAC and FedAQ respectively stand for FedAC-I and FedAQ-I. The number of quantization levels for FedAQ and FedPAQ is 16. We do experiments with $M = 1, 4, 16, 128, 512, 2048$ workers and $K = 1, 8, 32, 64, 128, 256$ synchronization intervals. We choose $\eta$ for each point that leads to the best suboptimality. We can say that an algorithm achieves a linear speedup when the best suboptimality decreases as the number of workers ($M$) increases. From Figure 4, we observe that FedAQ and FedAC achieve a linear speedup even when the synchronization interval $K$ is large, while FedAvg and FedPAQ lose a linear speedup when $K = 8$. This observation aligns well with our theoretical results that FedAQ and FedAC require a small number of communication rounds to achieve a linear speedup. That's why FedAQ and FedAC are more robust to infrequent communication in these experiments. Furthermore, the plots of FedAQ and FedAC are very similar. This empirical result reminds us of our theoretical result that the added quantization scheme does not hurt the convergence of the original algorithm that much.

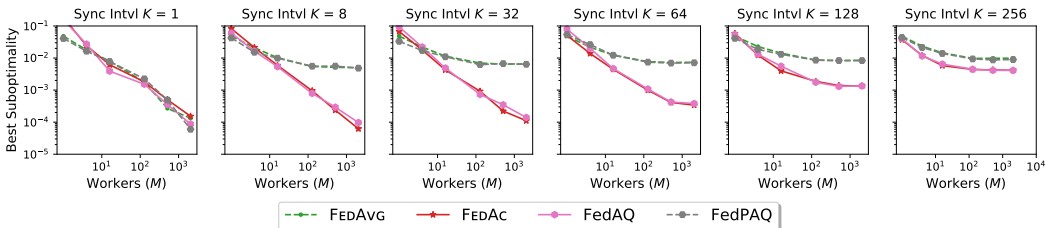

Figure 4: Comparing FedAQ with FedAvg, FedPAQ, and FedAC on UCI a9a dataset. We can say that an algorithm achieves a linear speedup when the best suboptimality decreases as the number of workers ($M$) increases. FedAQ and FedAC achieve a linear speedup even when the synchronization interval $K$ is large.

## D  Missing proofs for FedAQ under the parameter condition set (1)

Before diving into proof details, we define $\bar{w}_{k,\tau}, \bar{w}_{k,\tau}^{\mathrm{ag}}, \Psi_{k,t}^m, \Psi_{k,t}, \Psi_k, A_{k,t}^m$ as below.

$$\bar{w}_{k,\tau} = \frac{1}{M} \sum_{m=1}^{M} w_{k,\tau}^m$$

$$\bar{w}_{k,\tau}^{\mathrm{ag}} = \frac{1}{M} \sum_{m=1}^{M} w_{k,\tau}^{\mathrm{ag},m}$$

$$\Psi_{k,t}^m = F(w_{k,t}^{\mathrm{ag},m}) - F^* + \frac{1}{2}\mu\|w_{k,t}^m - w^*\|^2$$

$$\Psi_{k,t} = \frac{1}{M} \sum_{m=1}^{M} F(w_{k,t}^{\mathrm{ag},m}) - F^* + \frac{1}{2}\mu\|\bar{w}_{k,t} - w^*\|^2$$

$$\Psi_k := \Psi_{k,0} = F(w_k^{\mathrm{ag}}) - F^* + \frac{1}{2}\mu\|w_k - w^*\|^2$$

$$A_{k,t}^m = \frac{\gamma^2\mu^2(\mu+L)}{(1+\gamma\mu)^2}\|w_{k,t}^m - w_{k,t}^{\mathrm{ag},m}\|^2 + \gamma^2(\mu+L)\frac{2L}{1+\gamma\mu}\Psi_{k,t}^m$$

The above notations are essential to our convergence analysis. Intuitively, if the FedAQ algorithm converges to the optimal point, $\bar{w}_{k,\tau}, \bar{w}_{k,\tau}^{\mathrm{ag}}$ become $w^*$, and $\Psi_{k,t}^m, \Psi_{k,t}, \Psi_k, A_{k,t}^m$ become 0. In order to denote the $\sigma$-algebra generated by $\{w_{k',t'}^m, w_{k',t'}^{\mathrm{ag},m}\}_{(k'<k) \text{ or } (k'=k,t'\le t), m\in[M]}$, we use $\mathcal{F}_{k,t}$.

## D.1   Proof of Lemma D.1

**Lemma D.1.** *Let $F$ be $\mu$-strongly convex, and assume Assumption 2.1, 2.2, 2.3, 2.4, then for $\alpha = \frac{1}{\gamma\mu}, \beta = \alpha + 1, \gamma \in [\eta, \sqrt{\frac{\eta}{\mu}}], \eta, \gamma \in (0, \frac{1}{L}], \tau \geq 2$, FedAQ yields*

$$\mathbb{E}[\Psi_{k+1}] \leq C(\gamma, \tau)\mathbb{E}[\Psi_k] + \frac{1}{2}(\eta^2 L + \frac{\gamma^2\mu}{M})\tau\sigma^2$$

$$+ \gamma\mu L\tau \cdot \max_{0 \leq t < \tau} \mathbb{E}[\frac{1}{M}\sum_{m=1}^{M}\|\bar{w}_{k,t}^{md} - w_{k,t}^{md,m}\|\|\frac{1}{1+\gamma\mu}(\bar{w}_{k,t} - w_{k,t}^m) + \frac{\gamma\mu}{1+\gamma\mu}(\bar{w}_{k,t}^{ag} - w_{k,t}^{ag,m})\|]$$

$$+ \underbrace{\frac{q}{M}(\gamma^2\mu + \eta^2 L)\tau\sigma^2 + \frac{q}{2M}\Big(\frac{(\gamma - \eta)^2\gamma^2\mu^2(\mu + L)}{(1+\gamma\mu)^2} + \frac{\gamma^4(\mu + L)^2 L}{1+\gamma\mu}\Big)\tau^3\sigma^2}_{\text{Additional terms due to quantization}}$$

*Where $C(\gamma, \tau)$ is defined as*

$$C(\gamma, \tau) = (1 - \gamma\mu)^\tau + \underbrace{\frac{q}{M}\Big(\frac{4\gamma^2\mu(\mu + L)}{(1+\gamma\mu)^2} + \frac{2L\gamma^2(\mu + L)}{1+\gamma\mu}\Big)\tau^2}_{\text{Additional terms due to quantization}}$$

In this section, we first introduce five crucial Propositions for proving Lemma D.1. Then, we prove Lemma D.1 by using Propositions in the last part of this section.

**Proposition D.2.** *Let Assumption 2.1 hold and consider any $k$ synchronization round. Then, we can decompose the expectation as follows:*

$$\mathbb{E}[\|w_{k+1} - w^*\|^2] = \mathbb{E}[\|w_{k+1} - \bar{w}_{k,\tau}\|^2] + \mathbb{E}[\|\bar{w}_{k,\tau} - w^*\|^2]$$

$$\mathbb{E}[F(w_{k+1}^{ag}) - F^*] = \mathbb{E}[F(w_{k+1}^{ag}) - \frac{1}{M}\sum_{m=1}^{M}F(w_{k,\tau}^{ag,m})] + \mathbb{E}[\frac{1}{M}\sum_{m=1}^{M}F(w_{k,\tau}^{ag,m}) - F^*]$$

*Proof of Proposition D.2*   The second equality is trivial. Let's focus on the first equality. By Assumption 2.1, the quantizer $Q$ is unbiased and we get,

$$\mathbb{E}_Q[w_{k+1}] = w_k + \frac{1}{M}\sum_{m=1}^{M}\mathbb{E}_Q Q(w_{k,\tau}^m - w_k)$$

$$= \frac{1}{M}\sum_{m=1}^{M}w_{k,\tau}^m$$

$$= \bar{w}_{k,\tau}$$

Thus, we finally obtain

$$\mathbb{E}[\|w_{k+1} - w^*\|^2] = \mathbb{E}[\|w_{k+1} - \bar{w}_{k,\tau} + \bar{w}_{k,\tau} - w^*\|^2]$$

$$= \mathbb{E}[\|w_{k+1} - \bar{w}_{k,\tau}\|^2] + \mathbb{E}[\|\bar{w}_{k,\tau} - w^*\|^2]$$

**Proposition D.3.** *Let $F$ be $\mu$-strongly convex, and assume Assumption 2.2, 2.3, 2.4, then for $\alpha = \frac{1}{\gamma\mu}, \beta = \alpha + 1, \gamma \in [\eta, \sqrt{\frac{\eta}{\mu}}], \eta \in (0, \frac{1}{L}]$, FedAQ yields*

$$\mathbb{E}[\Psi_{k,\tau}] \leq (1 - \gamma\mu)^\tau\mathbb{E}[\Psi_k] + \frac{1}{2}(\eta^2 L + \frac{\gamma^2\mu}{M})\tau\sigma^2 + \gamma\mu L\tau$$

$$\cdot \max_{0 \leq t < \tau} \mathbb{E}[\frac{1}{M}\sum_{m=1}^{M}\|\bar{w}_{k,t}^{md} - w_{k,t}^{md,m}\|\|\frac{1}{1+\gamma\mu}(\bar{w}_{k,t} - w_{k,t}^m) + \frac{\gamma\mu}{1+\gamma\mu}(\bar{w}_{k,t}^{ag} - w_{k,t}^{ag,m})\|]$$

*Proof of Proposition D.3*   We refer to the proof of Lemma B.2 in Yuan and Ma (2020). There is no quantization between $\Psi_{k,\tau}$ and $\Psi_k$. Thus, we can directly apply useful inequalities in the proof of

Lemma B.2 in Yuan and Ma (2020) to our proof. Then, we obtain

$$\mathbb{E}[\Psi_{k,t+1}|\mathcal{F}_{k,t}] \le (1-\gamma\mu)\Psi_{k,t} + \frac{1}{2}(\eta^2 L + \frac{\gamma^2\mu}{M})\sigma^2 + \gamma\mu L$$

$$\cdot \frac{1}{M}\sum_{m=1}^{M} \|\bar{w}_{k,t}^{\mathrm{md}} - w_{k,t}^{\mathrm{md},m}\|\|\frac{1}{1+\gamma\mu}(\bar{w}_{k,t} - w_{k,t}^m) + \frac{\gamma\mu}{1+\gamma\mu}(\bar{w}_{k,t}^{\mathrm{ag}} - w_{k,t}^{\mathrm{ag},m})\|$$

From the above relationship between $\Psi_{k,t+1}$ and $\Psi_{k,t}$, we get

$$\mathbb{E}[\Psi_{k,\tau}] \le (1-\gamma\mu)^\tau \mathbb{E}[\Psi_k] + \Big(\sum_{t=0}^{\tau-1}(1-\gamma\mu)^t\Big)\frac{1}{2}(\eta^2 L + \frac{\gamma^2\mu}{M})\sigma^2 + \gamma\mu L \cdot \sum_{t=0}^{\tau-1}\Big\{(1-\gamma\mu)^{\tau-t-1}$$

$$\mathbb{E}[\frac{1}{M}\sum_{m=1}^{M}\|\bar{w}_{k,t}^{\mathrm{md}} - w_{k,t}^{\mathrm{md},m}\|\|\frac{1}{1+\gamma\mu}(\bar{w}_{k,t} - w_{k,t}^m) + \frac{\gamma\mu}{1+\gamma\mu}(\bar{w}_{k,t}^{\mathrm{ag}} - w_{k,t}^{\mathrm{ag},m})\|]\Big\}$$

$$\le (1-\gamma\mu)^\tau \mathbb{E}[\Psi_k] + \frac{1}{2}(\eta^2 L + \frac{\gamma^2\mu}{M})\tau\sigma^2 + \gamma\mu L\tau$$

$$\cdot \max_{0 \le t < \tau} \mathbb{E}[\frac{1}{M}\sum_{m=1}^{M}\|\bar{w}_{k,t}^{\mathrm{md}} - w_{k,t}^{\mathrm{md},m}\|\|\frac{1}{1+\gamma\mu}(\bar{w}_{k,t} - w_{k,t}^m) + \frac{\gamma\mu}{1+\gamma\mu}(\bar{w}_{k,t}^{\mathrm{ag}} - w_{k,t}^{\mathrm{ag},m})\|]$$

**Proposition D.4.** *Let Assumption 2.1 hold. Then, we have*

$$\mathbb{E}[\|w_{k+1} - \bar{w}_{k,\tau}\|^2] \le \frac{q}{M^2}\sum_{m=1}^{M}\mathbb{E}[\|w_{k,\tau}^m - w_k\|^2]$$

$$\mathbb{E}[F(w_{k+1}^{ag}) - \frac{1}{M}\sum_{m=1}^{M}F(w_{k,\tau}^{ag,m})] \le \frac{qL}{2M^2}\sum_{m=1}^{M}\mathbb{E}[\|w_{k,\tau}^{ag,m} - w_k^{ag}\|^2]$$

*Proof of Proposition D.4*   First, let's consider the first inequality. According to Assumption 2.1, we get

$$\mathbb{E}[\|w_{k+1} - \bar{w}_{k,\tau}\|^2] = \mathbb{E}[\|w_k + \frac{1}{M}\sum_{m=1}^{M}Q(w_{k,\tau}^m - w_k) - \frac{1}{M}\sum_{m=1}^{M}w_{k,\tau}^m\|^2]$$

$$= \mathbb{E}[\|\frac{1}{M}\sum_{m=1}^{M}Q(w_{k,\tau}^m - w_k) - (w_{k,\tau}^m - w_k)\|^2]$$

$$= \frac{1}{M^2}\sum_{m=1}^{M}\mathbb{E}[\|Q(w_{k,\tau}^m - w_k) - (w_{k,\tau}^m - w_k)\|^2]$$

$$\le \frac{q}{M^2}\sum_{m=1}^{M}\mathbb{E}\|w_{k,\tau}^m - w_k\|^2$$

The third equality comes from the unbiasedness of $Q$, and the last inequality stems from the variance assumption of $Q$. Similarly, we obtain

$$\mathbb{E}[F(w_{k+1}^{\mathrm{ag}}) - \frac{1}{M}\sum_{m=1}^{M} F(w_{k,\tau}^{\mathrm{ag},m})] = \mathbb{E}[F(w_k^{\mathrm{ag}} + \frac{1}{M}\sum_{m=1}^{M} Q(w_{k,\tau}^{\mathrm{ag},m} - w_k^{\mathrm{ag}})) - \frac{1}{M}\sum_{m=1}^{M} F(w_{k,\tau}^{\mathrm{ag},m})]$$

$$= \mathbb{E}[\frac{1}{M}\sum_{m=1}^{M} F(w_k^{\mathrm{ag}} + \frac{1}{M}\sum_{m=1}^{M} Q(w_{k,\tau}^{\mathrm{ag},m} - w_k^{\mathrm{ag}})) - F(w_{k,\tau}^{\mathrm{ag},m})]$$

$$\leq \mathbb{E}\Big[\frac{1}{M}\sum_{m=1}^{M}\langle \nabla F(w_{k,\tau}^{\mathrm{ag},m}), \frac{1}{M}\sum_{m=1}^{M}\Big(Q(w_{k,\tau}^{\mathrm{ag},m} - w_k^{\mathrm{ag}}) - (w_{k,\tau}^{\mathrm{ag},m}$$

$$- w_k^{\mathrm{ag}})\Big)\rangle + \frac{L}{2}\|\frac{1}{M}\sum_{m=1}^{M} Q(w_{k,\tau}^{\mathrm{ag},m} - w_k^{\mathrm{ag}}) - (w_{k,\tau}^{\mathrm{ag},m} - w_k^{\mathrm{ag}})\|^2\Big]$$

$$= \frac{L}{2}\mathbb{E}[\|\frac{1}{M}\sum_{m=1}^{M} Q(w_{k,\tau}^{\mathrm{ag},m} - w_k^{\mathrm{ag}}) - (w_{k,\tau}^{\mathrm{ag},m} - w_k^{\mathrm{ag}})\|^2]$$

$$= \frac{L}{2M^2}\sum_{m=1}^{M}\mathbb{E}[\|Q(w_{k,\tau}^{\mathrm{ag},m} - w_k^{\mathrm{ag}}) - (w_{k,\tau}^{\mathrm{ag},m} - w_k^{\mathrm{ag}})\|^2]$$

$$\leq \frac{qL}{2M^2}\sum_{m=1}^{M}\mathbb{E}[\|w_{k,\tau}^{\mathrm{ag},m} - w_k^{\mathrm{ag}}\|^2]$$

**Proposition D.5.** *Let F be $\mu$-strongly convex, and assume Assumption 2.2, 2.3, 2.4, then for $\alpha = \frac{1}{\gamma\mu}, \beta = \alpha + 1, \gamma \in [\eta, \sqrt{\frac{\eta}{\mu}}], \eta, \gamma \in (0, \frac{1}{L}]$, we get*

$$\mathbb{E}[A_{k,t}^m] \leq \mathbb{E}[A_{k,0}^m] + \Big(\frac{(\gamma-\eta)^2(\mu+L)}{1+\gamma\mu} + \frac{\gamma^2(\mu+L)^2 L}{\mu^2}\Big)\cdot\Big(1 - (1-\gamma\mu + \frac{\gamma\mu}{1+\gamma\mu})^t\Big)\sigma^2$$

*Proof of Proposition D.5* From the notation mentioned in the beginning of Section D,

$$\mathbb{E}[A_{k,t+1}^m|\mathcal{F}_{k,t}] = \frac{\gamma^2\mu^2(\mu+L)}{(1+\gamma\mu)^2}\mathbb{E}[\|w_{k,t+1}^m - w_{k,t+1}^{\mathrm{ag},m}\|^2|\mathcal{F}_{k,t}] + \gamma^2(\mu+L)\frac{2L}{1+\gamma\mu}\mathbb{E}[\Psi_{k,t+1}^m|\mathcal{F}_{k,t}]$$

$$(3)$$

Thus, let's sequentially compute $\mathbb{E}[\|w_{k,t+1}^m - w_{k,t+1}^{\mathrm{ag},m}\|^2|\mathcal{F}_{k,t}]$ and $\mathbb{E}[\Psi_{k,t+1}^m|\mathcal{F}_{k,t}]$.

$$\mathbb{E}[\|w_{k,t+1}^m - w_{k,t+1}^{\mathrm{ag},m}\|^2|\mathcal{F}_{k,t}] = \mathbb{E}[\|(1-\alpha^{-1})w_{k,t}^m + \alpha^{-1}w_{k,t}^{\mathrm{md},m} - \gamma g_{k,t}^m - w_{k,t}^{\mathrm{md},m} + \eta g_{k,t}^m\|^2|\mathcal{F}_{k,t}]$$

$$= \mathbb{E}[\|(1-\alpha^{-1})(w_{k,t}^m - w_{k,t}^{\mathrm{md},m}) - (\gamma-\eta)g_{k,t}^m\|^2|\mathcal{F}_{k,t}]\ (\leftarrow \gamma \geq \eta)$$

$$= \|(1-\alpha^{-1})(w_{k,t}^m - w_{k,t}^{\mathrm{md},m}) - (\gamma-\eta)\nabla F(w_{k,t}^{\mathrm{md},m})\|^2$$

$$+ (\gamma-\eta)^2\mathbb{E}[\|\nabla F(w_{k,t}^{\mathrm{md},m}) - g_{k,t}^m\|^2|\mathcal{F}_{k,t}]$$

$$\leq (1-\alpha^{-1})^2\|w_{k,t}^m - w_{k,t}^{\mathrm{md},m}\|^2 + (\gamma-\eta)^2\|\nabla F(w_{k,t}^{\mathrm{md},m})\|^2$$

$$+ (\gamma-\eta)^2\sigma^2 - 2(\gamma-\eta)\langle(1-\alpha^{-1})(w_{k,t}^m - w_{k,t}^{\mathrm{md},m}), \nabla F(w_{k,t}^{\mathrm{md},m})\rangle$$

$$\leq (1-\alpha^{-1})^2(1+\gamma\mu)\|w_{k,t}^m - w_{k,t}^{\mathrm{md},m}\|^2$$

$$+ (\gamma-\eta)^2(1+\frac{1}{\gamma\mu})\|\nabla F(w_{k,t}^{\mathrm{md},m})\|^2 + (\gamma-\eta)^2\sigma^2$$

$$= \frac{(1-\gamma\mu)^2}{1+\gamma\mu}\|w_{k,t}^m - w_{k,t}^{\mathrm{ag},m}\|^2 + (\gamma-\eta)^2\frac{1+\gamma\mu}{\gamma\mu}\|\nabla F(w_{k,t}^{\mathrm{md},m})\|^2$$

$$+ (\gamma-\eta)^2\sigma^2$$

Here, we need to bound $\|\nabla F(w_{k,t}^{\text{md},m})\|^2$.

$$
\begin{aligned}
\|\nabla F(w_{k,t}^{\text{md},m})\|^2 &\leq 2L(F(w_{k,t}^{\text{md},m}) - F^*) \;(\because \text{ Assumption 2.3}) \\
&\leq 2L\Big(\beta^{-1}(F(w_{k,t}^m) - F(w^*)) + (1 - \beta^{-1})(F(w_{k,t}^{\text{ag},m}) - F^*)\Big) \\
&\leq \beta^{-1}L^2\|w_{k,t}^m - w^*\|^2 + 2(1 - \beta^{-1})L(F(w_{k,t}^{\text{ag},m}) - F^*) \\
&= \frac{\gamma\mu L^2}{1 + \gamma\mu}\|w_{k,t}^m - w^*\|^2 + \frac{2L}{1 + \gamma\mu}(F(w_{k,t}^{\text{ag},m}) - F^*) \\
&\leq \frac{\mu L}{1 + \gamma\mu}\|w_{k,t}^m - w^*\|^2 + \frac{2L}{1 + \gamma\mu}(F(w_{k,t}^{\text{ag},m}) - F^*) = \frac{2L}{1 + \gamma\mu}\Psi_{k,t}^m \quad (4)
\end{aligned}
$$

The last inequality comes from the fact $\gamma \in [0, \frac{1}{L})$. Therefore, we finally get

$$
\begin{aligned}
&\mathbb{E}[\|w_{k,t+1}^m - w_{k,t+1}^{\text{ag},m}\|^2|\mathcal{F}_{k,t}] \\
&\leq \frac{(1 - \gamma\mu)^2}{1 + \gamma\mu}\|w_{k,t}^m - w_{k,t}^{\text{ag},m}\|^2 + (\gamma - \eta)^2\frac{1 + \gamma\mu}{\gamma\mu}\|\nabla F(w_{k,t}^{\text{md},m})\|^2 + (\gamma - \eta)^2\sigma^2 \\
&\leq \frac{(1 - \gamma\mu)^2}{1 + \gamma\mu}\|w_{k,t}^m - w_{k,t}^{\text{ag},m}\|^2 + (\gamma - \eta)^2\frac{1 + \gamma\mu}{\gamma\mu}\Big(\frac{2L}{1 + \gamma\mu}\Psi_{k,t}^m\Big) + (\gamma - \eta)^2\sigma^2 \quad (5)
\end{aligned}
$$

Now, let's compute $\mathbb{E}[\Psi_{k,t+1}^m|\mathcal{F}_{k,t}]$. We need to compute $\mathbb{E}[\|w_{k,t+1}^m - w^*\|^2|\mathcal{F}_{k,t}]$ and $\mathbb{E}[F(w_{k,t+1}^{\text{ag},m}) - F^*|\mathcal{F}_{k,t}]$ first.

$$
\begin{aligned}
&\mathbb{E}[\|w_{k,t+1}^m - w^*\|^2|\mathcal{F}_{k,t}] \\
&= \mathbb{E}[\|(1 - \alpha^{-1})w_{k,t}^m + \alpha^{-1}w_{k,t}^{\text{md},m} - \gamma g_{k,t}^m - w^*\|^2|\mathcal{F}_{k,t}] \\
&\leq \|(1 - \alpha^{-1})w_{k,t}^m + \alpha^{-1}w_{k,t}^{\text{md},m} - w^*\|^2 + \gamma^2\|\nabla F(w_{k,t}^{\text{md},m})\|^2 + \gamma^2\sigma^2 \\
&\quad - 2\gamma\langle(1 - \alpha^{-1})w_{k,t}^m + \alpha^{-1}w_{k,t}^{\text{md},m} - w^*, \nabla F(w_{k,t}^{\text{md},m})\rangle \\
&\leq (1 - \alpha^{-1})\|w_{k,t}^m - w^*\|^2 + \alpha^{-1}\|w_{k,t}^{\text{md},m} - w^*\|^2 + \gamma^2\|\nabla F(w_{k,t}^{\text{md},m})\|^2 + \gamma^2\sigma^2 \\
&\quad - 2\gamma\langle(1 - \alpha^{-1}(1 - \beta^{-1}))w_{k,t}^m + \alpha^{-1}(1 - \beta^{-1})w_{k,t}^{\text{ag},m} - w^*, \nabla F(w_{k,t}^{\text{md},m})\rangle \\
&= (1 - \gamma\mu)\|w_{k,t}^m - w^*\|^2 + \gamma\mu\|w_{k,t}^{\text{md},m} - w^*\|^2 + \gamma^2\|\nabla F(w_{k,t}^{\text{md},m})\|^2 + \gamma^2\sigma^2 \\
&\quad - 2\gamma\langle\frac{1}{1 + \gamma\mu}w_{k,t}^m + \frac{\gamma\mu}{1 + \gamma\mu}w_{k,t}^{\text{ag},m} - w^*, \nabla F(w_{k,t}^{\text{md},m})\rangle
\end{aligned}
$$

$$\mathbb{E}[F(w_{k,t+1}^{\text{ag},m}) - F^* | \mathcal{F}_{k,t}]$$

$$\leq \mathbb{E}[F(w_{k,t}^{\text{md},m}) + \langle \nabla F(w_{k,t}^{\text{md},m}), w_{k,t+1}^{\text{ag},m} - w_{k,t}^{\text{md},m}\rangle + \frac{L}{2}\|w_{k,t+1}^{\text{ag},m} - w_{k,t}^{\text{md},m}\|^2 - F^* | \mathcal{F}_{k,t}]$$

$$\leq F(w_{k,t}^{\text{md},m}) - F^* - \eta\|\nabla F(w_{k,t}^{\text{md},m})\|^2 + \frac{\eta^2 L}{2}\|\nabla F(w_{k,t}^{\text{md},m})\|^2 + \frac{\eta^2 L}{2}\sigma^2$$

$$\leq F(w_{k,t}^{\text{md},m}) - F^* - \frac{\eta}{2}\|\nabla F(w_{k,t}^{\text{md},m})\|^2 + \frac{\eta^2 L}{2}\sigma^2 \ (\because 1 - \frac{\eta L}{2} \geq \frac{1}{2} \leftarrow \eta \in [0, \frac{1}{L}])$$

$$= (1 - \alpha^{-1})(F(w_{k,t}^{\text{ag},m}) - F^*) + \alpha^{-1}(F(w_{k,t}^{\text{md},m}) - F^*)$$

$$+ (1 - \alpha^{-1})(F(w_{k,t}^{\text{md},m}) - F(w_{k,t}^{\text{ag},m})) - \frac{\eta}{2}\|\nabla F(w_{k,t}^{\text{md},m})\|^2 + \frac{\eta^2 L}{2}\sigma^2$$

$$\leq (1 - \alpha^{-1})(F(w_{k,t}^{\text{ag},m}) - F^*) - \frac{\mu\alpha^{-1}}{2}\|w_{k,t}^{\text{md},m} - w^*\|^2$$

$$+ \alpha^{-1}\langle \nabla F(w_{k,t}^{\text{md},m}), w_{k,t}^{\text{md},m} - w^*\rangle + (1 - \alpha^{-1})\langle \nabla F(w_{k,t}^{\text{md},m}), w_{k,t}^{\text{md},m} - w_{k,t}^{\text{ag},m}\rangle$$

$$- \frac{\eta}{2}\|\nabla F(w_{k,t}^{\text{md},m})\|^2 + \frac{\eta^2 L}{2}\sigma^2$$

$$= (1 - \alpha^{-1})(F(w_{k,t}^{\text{ag},m}) - F^*) - \frac{\mu\alpha^{-1}}{2}\|w_{k,t}^{\text{md},m} - w^*\|^2 - \frac{\eta}{2}\|\nabla F(w_{k,t}^{\text{md},m})\|^2 + \frac{\eta^2 L}{2}\sigma^2$$

$$+ \alpha^{-1}\langle \nabla F(w_{k,t}^{\text{md},m}), \alpha\beta^{-1}w_{k,t}^m + (1 - \alpha\beta^{-1})w_{k,t}^{\text{ag},m} - w^*\rangle$$

$$= (1 - \gamma\mu)(F(w_{k,t}^{\text{ag},m}) - F^*) - \frac{\gamma\mu^2}{2}\|w_{k,t}^{\text{md},m} - w^*\|^2 - \frac{\eta}{2}\|\nabla F(w_{k,t}^{\text{md},m})\|^2 + \frac{\eta^2 L}{2}\sigma^2$$

$$+ \gamma\mu\langle \frac{1}{1+\gamma\mu}w_{k,t}^m + \frac{\gamma\mu}{1+\gamma\mu}w_{k,t}^{\text{ag},m} - w^*, \nabla F(w_{k,t}^{\text{md},m})\rangle$$

Then, we bound $\mathbb{E}[\Psi_{k,t+1}^m | \mathcal{F}_{k,t}]$ by using the above results.

$$\mathbb{E}[\Psi_{k,t+1}^m | \mathcal{F}_{k,t}] = \frac{\mu}{2}\mathbb{E}[\|w_{k,t+1}^m - w^*\|^2 | \mathcal{F}_{k,t}] + \mathbb{E}[F(w_{k,t+1}^{\text{ag},m}) - F^* | \mathcal{F}_{k,t}]$$

$$\leq (1 - \gamma\mu)\Psi_{k,t}^m - \frac{\eta - \gamma^2\mu}{2}\|\nabla F(w_{k,t}^{\text{md},m})\|^2 + \frac{\gamma^2\mu + \eta^2 L}{2}\sigma^2$$

$$\leq (1 - \gamma\mu)\Psi_{k,t}^m + \frac{\gamma^2\mu + \eta^2 L}{2}\sigma^2 \ (\because \gamma \leq \sqrt{\frac{\eta}{\mu}})$$

$$\leq (1 - \gamma\mu)\Psi_{k,t}^m + \frac{\gamma^2(\mu + L)}{2}\sigma^2 \tag{6}$$

Plugging (5), (6) in (3) yields,

$$\mathbb{E}[A_{k,t+1}^m | \mathcal{F}_{k,t}]$$

$$\leq \frac{\gamma^2\mu^2(\mu+L)}{(1+\gamma\mu)^2}\left(\frac{(1-\gamma\mu)^2}{1+\gamma\mu}\|w_{k,t}^m - w_{k,t}^{\text{ag},m}\|^2 + (\gamma - \eta)^2 \frac{1+\gamma\mu}{\gamma\mu}\left(\frac{2L}{1+\gamma\mu}\Psi_{k,t}^m\right) + (\gamma - \eta)^2\sigma^2\right)$$

$$+ \gamma^2(\mu+L)\frac{2L}{1+\gamma\mu}\left((1-\gamma\mu)\Psi_{k,t}^m + \frac{\gamma^2(\mu+L)}{2}\sigma^2\right)$$

$$= \frac{(1-\gamma\mu)^2}{1+\gamma\mu} \cdot \frac{\gamma^2\mu^2(\mu+L)}{(1+\gamma\mu)^2}\|w_{k,t}^m - w_{k,t}^{\text{ag},m}\|^2 + \left(\frac{\gamma\mu(\gamma-\eta)^2(\mu+L)}{1+\gamma\mu}\right.$$

$$+ \gamma^2(\mu+L)(1-\gamma\mu)\left)\frac{2L}{1+\gamma\mu}\Psi_{k,t}^m + \left(\frac{\gamma^2\mu^2(\gamma-\eta)^2(\mu+L)}{(1+\gamma\mu)^2} + \frac{\gamma^4(\mu+L)^2 L}{1+\gamma\mu}\right)\sigma^2 \tag{7}$$

Since $\eta \leq \gamma$, we get $(\gamma - \eta)^2 \leq \gamma^2$. By using this fact, we obtain

$$\frac{\gamma\mu(\gamma-\eta)^2(\mu+L)}{1+\gamma\mu} + \gamma^2(\mu+L)(1-\gamma\mu) \leq \frac{\gamma^3\mu(\mu+L)}{1+\gamma\mu} + \gamma^2(\mu+L)(1-\gamma\mu)$$

$$= \gamma^2(\mu+L)(1 - \gamma\mu + \frac{\gamma\mu}{1+\gamma\mu}) \tag{8}$$

It is easy to show that $1 - \gamma\mu + \frac{\gamma\mu}{1+\gamma\mu} < 1$. Also, we get

$$\frac{(1-\gamma\mu)^2}{1+\gamma\mu} < 1 - \gamma\mu < 1 - \gamma\mu + \frac{\gamma\mu}{1+\gamma\mu} \tag{9}$$

From (7), (8), and (9) we finally get

$$\mathbb{E}[A^m_{k,t+1}|\mathcal{F}_{k,t}] \le (1 - \gamma\mu + \frac{\gamma\mu}{1+\gamma\mu})A^m_{k,t} + \left(\frac{\gamma^2\mu^2(\gamma-\eta)^2(\mu+L)}{(1+\gamma\mu)^2} + \frac{\gamma^4(\mu+L)^2 L}{1+\gamma\mu}\right)\sigma^2$$

From this relationship between $A^m_{k,t+1}$ and $A^m_{k,t}$, we obtain the result of Proposition D.5.

$$
\begin{aligned}
\mathbb{E}[A^m_{k,t}] &\le (1 - \gamma\mu + \frac{\gamma\mu}{1+\gamma\mu})^t \mathbb{E}[A^m_{k,0}] + \left(\frac{\gamma^2\mu^2(\gamma-\eta)^2(\mu+L)}{(1+\gamma\mu)^2} + \frac{\gamma^4(\mu+L)^2 L}{1+\gamma\mu}\right)\sigma^2 \\
&\quad \cdot \frac{1 - (1-\gamma\mu+\frac{\gamma\mu}{1+\gamma\mu})^t}{1 - (1-\gamma\mu+\frac{\gamma\mu}{1+\gamma\mu})} \\
&= (1 - \gamma\mu + \frac{\gamma\mu}{1+\gamma\mu})^t \mathbb{E}[A^m_{k,0}] + \left(\frac{(\gamma-\eta)^2(\mu+L)}{1+\gamma\mu} + \frac{\gamma^2(\mu+L)^2 L}{\mu^2}\right)\sigma^2 \\
&\quad \cdot \left(1 - (1 - \gamma\mu + \frac{\gamma\mu}{1+\gamma\mu})^t\right) \\
&\le \mathbb{E}[A^m_{k,0}] + \left(\frac{(\gamma-\eta)^2(\mu+L)}{1+\gamma\mu} + \frac{\gamma^2(\mu+L)^2 L}{\mu^2}\right) \cdot \left(1 - (1 - \gamma\mu + \frac{\gamma\mu}{1+\gamma\mu})^t\right)\sigma^2
\end{aligned}
$$

**Proposition D.6.** *Let F be $\mu$-strongly convex, and assume Assumption 2.2, 2.3, 2.4, then for $\alpha = \frac{1}{\gamma\mu}, \beta = \alpha + 1, \gamma \in [\eta, \sqrt{\frac{\eta}{\mu}}], \eta, \gamma \in (0, \frac{1}{L}], \tau \ge 2$, FedAQ yields*

$$
\begin{aligned}
\frac{\mu}{2}\mathbb{E}[\|w^m_{k,\tau} - w_k\|^2] + \frac{L}{2}\mathbb{E}[\|w^{ag,m}_{k,\tau} - w^{ag}_k\|^2] &\le \left(\frac{4\gamma^2\mu(\mu+L)}{(1+\gamma\mu)^2} + \frac{2L\gamma^2(\mu+L)}{1+\gamma\mu}\right)\tau^2\mathbb{E}[\Psi_k] \\
&\quad + (\gamma^2\mu + \eta^2 L)\tau\sigma^2 \\
&\quad + \left(\frac{(\gamma-\eta)^2\gamma^2\mu^2(\mu+L)}{(1+\gamma\mu)^2} + \frac{\gamma^4(\mu+L)^2 L}{1+\gamma\mu}\right)\frac{\tau^3\sigma^2}{2}
\end{aligned}
$$

*Proof of Proposition D.6* Let's first bound $\mathbb{E}[\|w_{k,\tau}^m - w_k\|^2]$ and $\mathbb{E}[\|w_{k,\tau}^{\mathrm{ag},m} - w_k^{\mathrm{ag}}\|^2]$ individually.

$$
\begin{aligned}
\mathbb{E}[\|w_{k,\tau}^m - w_k\|^2] &= \mathbb{E}[\|(w_{k,\tau}^m - w_{k,\tau-1}^m) + \cdots + (w_{k,1}^m - w_{k,0}^m)\|^2] \\
&= \mathbb{E}\Big[\Big\|\sum_{t=0}^{\tau-1}\Big((1-\alpha^{-1})w_{k,t}^m + \alpha^{-1}w_{k,t}^{\mathrm{md,\ m}} - w_{k,t}^m - \gamma g_{k,t}^m\Big)\Big\|^2\Big] \\
&= \mathbb{E}\Big[\Big\|\alpha^{-1}\sum_{t=0}^{\tau-1}(w_{k,t}^{\mathrm{md},m} - w_{k,t}^m) - \gamma\sum_{t=0}^{\tau-1}g_{k,t}^m\Big\|^2\Big] \\
&\leq 2\alpha^{-2}\mathbb{E}[\|\sum_{t=0}^{\tau-1}(w_{k,t}^{\mathrm{md},m} - w_{k,t}^m)\|^2] + 2\gamma^2\mathbb{E}[\|\sum_{t=0}^{\tau-1}g_{k,t}^m\|^2] \\
&\leq 2\alpha^{-2}\tau\sum_{t=0}^{\tau-1}\mathbb{E}[\|w_{k,t}^{\mathrm{md},m} - w_{k,t}^m\|^2] + 2\gamma^2\mathbb{E}[\|\sum_{t=0}^{\tau-1}\nabla F(w_{k,t}^{\mathrm{md},m})\|^2] \\
&\quad + 2\gamma^2\mathbb{E}[\|\sum_{t=0}^{\tau-1}(g_{k,t}^m - \nabla F(w_{k,t}^{\mathrm{md},m}))\|^2] \\
&\leq 2\alpha^{-2}(1-\beta^{-1})^2\tau\sum_{t=0}^{\tau-1}\mathbb{E}[\|w_{k,t}^m - w_{k,t}^{\mathrm{ag},m}\|^2] + 2\gamma^2\tau\sum_{t=0}^{\tau-1}\mathbb{E}[\|\nabla F(w_{k,t}^{\mathrm{md},m})\|^2] \\
&\quad + 2\gamma^2\sum_{t=0}^{\tau-1}\mathbb{E}[\|g_{k,t}^m - \nabla F(w_{k,t}^{\mathrm{md},m})\|^2] \\
&= \tau\Big(\sum_{t=0}^{\tau-1}2\alpha^{-2}(1-\beta^{-1})^2\mathbb{E}[\|w_{k,t}^m - w_{k,t}^{\mathrm{ag},m}\|^2] + 2\gamma^2\mathbb{E}[\|\nabla F(w_{k,t}^{\mathrm{md},m})\|^2]\Big) \\
&\quad + 2\tau\gamma^2\sigma^2
\end{aligned}
$$

$$
\begin{aligned}
\mathbb{E}[\|w_{k,\tau}^{\mathrm{ag},m} - w_k^{\mathrm{ag}}\|^2] &= \mathbb{E}[\|\sum_{t=0}^{\tau-1}(w_{k,t+1}^{\mathrm{ag},m} - w_{k,t}^{\mathrm{ag},m})\|^2] \\
&= \mathbb{E}[\|\sum_{t=0}^{\tau-1}(w_{k,t}^{\mathrm{md},m} - w_{k,t}^{\mathrm{ag},m} - \eta g_{k,t}^m)\|^2] \\
&\leq 2\mathbb{E}[\|\sum_{t=0}^{\tau-1}(w_{k,t}^{\mathrm{md},m} - w_{k,t}^{\mathrm{ag},m})\|^2] + 2\eta^2\mathbb{E}[\|\sum_{t=0}^{\tau-1}g_{k,t}^m\|^2] \\
&= 2\beta^{-2}\mathbb{E}[\|\sum_{t=0}^{\tau-1}(w_{k,t}^m - w_{k,t}^{\mathrm{ag},m})\|^2] + 2\eta^2\mathbb{E}[\|\sum_{t=0}^{\tau-1}\nabla F(w_{k,t}^{\mathrm{md},m})\|^2] \\
&\quad + 2\eta^2\mathbb{E}[\|\sum_{t=0}^{\tau-1}(g_{k,t}^m - \nabla F(w_{k,t}^{\mathrm{md},m}))\|^2] \\
&\leq 2\beta^{-2}\tau\sum_{t=0}^{\tau-1}\mathbb{E}[\|w_{k,t}^m - w_{k,t}^{\mathrm{ag},m}\|^2] + 2\eta^2\tau\sum_{t=0}^{\tau-1}\mathbb{E}[\|\nabla F(w_{k,t}^{\mathrm{md},m})\|^2] \\
&\quad + 2\eta^2\sum_{t=0}^{\tau-1}\mathbb{E}[\|g_{k,t}^m - \nabla F(w_{k,t}^{\mathrm{md},m})\|^2] \\
&= \tau\Big(\sum_{t=0}^{\tau-1}2\beta^{-2}\mathbb{E}[\|w_{k,t}^m - w_{k,t}^{\mathrm{ag},m}\|^2] + 2\eta^2\mathbb{E}[\|\nabla F(w_{k,t}^{\mathrm{md},m})\|^2]\Big) + 2\tau\eta^2\sigma^2
\end{aligned}
$$

Thus, by using the above results, we get

$$\frac{\mu}{2}\mathbb{E}[\|w_{k,\tau}^m - w_k\|^2] + \frac{L}{2}\mathbb{E}[\|w_{k,\tau}^{\mathrm{ag},m} - w_k^{\mathrm{ag}}\|^2]$$

$$\leq \tau \sum_{t=0}^{\tau-1} \left\{ \left(\mu\alpha^{-2}(1-\beta^{-1})^2 + L\beta^{-2}\right)\mathbb{E}[\|w_{k,t}^m - w_{k,t}^{\mathrm{ag},m}\|^2] + (\gamma^2\mu + \eta^2 L)\mathbb{E}[\|\nabla F(w_{k,t}^{\mathrm{md},m})\|^2] \right\}$$

$$+ (\gamma^2\mu + \eta^2 L)\tau\sigma^2$$

$$\leq \tau \sum_{t=0}^{\tau-1} \left\{ \left(\mu\alpha^{-2}(1-\beta^{-1})^2 + L\beta^{-2}\right)\mathbb{E}[\|w_{k,t}^m - w_{k,t}^{\mathrm{ag},m}\|^2] + (\gamma^2\mu + \eta^2 L)\frac{2L}{1+\gamma\mu}\mathbb{E}[\Psi_{k,t}^m] \right\}$$

$$+ (\gamma^2\mu + \eta^2 L)\tau\sigma^2 \ (\because (4))$$

$$\leq \tau \sum_{t=0}^{\tau-1} \left\{ \frac{\gamma^2\mu^2(\mu+L)}{(1+\gamma\mu)^2}\mathbb{E}[\|w_{k,t}^m - w_{k,t}^{\mathrm{ag},m}\|^2] + \gamma^2(\mu+L)\frac{2L}{1+\gamma\mu}\mathbb{E}[\Psi_{k,t}^m] \right\} + (\gamma^2\mu + \eta^2 L)\tau\sigma^2$$

$$= \tau\left(\sum_{t=0}^{\tau-1}\mathbb{E}[A_{k,t}^m]\right) + (\gamma^2\mu + \eta^2 L)\tau\sigma^2$$

By Proposition D.5 and the fact $\Psi_{k,0}^m = \Psi_k$, we obtain

$$\frac{\mu}{2}\mathbb{E}[\|w_{k,\tau}^m - w_k\|^2] + \frac{L}{2}\mathbb{E}[\|w_{k,\tau}^{\mathrm{ag},m} - w_k^{\mathrm{ag}}\|^2]$$

$$\leq \tau\left\{ \sum_{t=0}^{\tau-1}\mathbb{E}[A_{k,0}^m] + \left(\frac{(\gamma-\eta)^2(\mu+L)}{1+\gamma\mu} + \frac{\gamma^2(\mu+L)^2 L}{\mu^2}\right)\cdot\left(1 - (1-\gamma\mu+\frac{\gamma\mu}{1+\gamma\mu})^t\right)\sigma^2 \right\}$$

$$+ (\gamma^2\mu + \eta^2 L)\tau\sigma^2$$

$$= \tau^2\left(\frac{\gamma^2\mu^2(\mu+L)}{(1+\gamma\mu)^2}\mathbb{E}[\|w_k - w_k^{\mathrm{ag}}\|^2] + \gamma^2(\mu+L)\frac{2L}{1+\gamma\mu}\mathbb{E}[\Psi_k]\right)$$

$$+ \tau\left(\frac{(\gamma-\eta)^2(\mu+L)}{1+\gamma\mu} + \frac{\gamma^2(\mu+L)^2 L}{\mu^2}\right)\left(\sum_{t=0}^{\tau-1}1 - (1-\gamma\mu+\frac{\gamma\mu}{1+\gamma\mu})^t\right)\sigma^2 + (\gamma^2\mu + \eta^2 L)\tau\sigma^2$$

Before we get to the final result, let's find the upper bound for $\|w_k - w_k^{\mathrm{ag}}\|^2$, $\sum_{t=0}^{\tau-1}\left(1 - (1-\gamma\mu + \frac{\gamma\mu}{1+\gamma\mu})^t\right)$

$$\|w_k - w_k^{\mathrm{ag}}\|^2 = \|w_k - w^* - (w_k^{\mathrm{ag}} - w^*)\|^2$$

$$\leq 2\|w_k - w^*\|^2 + 2\|w_k^{\mathrm{ag}} - w^*\|^2$$

$$\leq 2\|w_k - w^*\|^2 + 2\cdot\frac{2}{\mu}\left(F(w_k^{\mathrm{ag}}) - F^* - \langle\nabla F(w^*), w_k^{\mathrm{ag}} - w^*\rangle\right)$$

$$= 2\|w_k - w^*\|^2 + \frac{4}{\mu}(F(w_k^{\mathrm{ag}}) - F^*) = \frac{4}{\mu}\Psi_k$$

$$\sum_{t=0}^{\tau-1}\left(1 - (1-\gamma\mu + \frac{\gamma\mu}{1+\gamma\mu})^t\right) = \tau - \sum_{t=0}^{\tau-1}(1-\gamma\mu + \frac{\gamma\mu}{1+\gamma\mu})^t$$

$$= \tau - \frac{1 - (1-\gamma\mu + \frac{\gamma\mu}{1+\gamma\mu})^\tau}{1 - (1-\gamma\mu + \frac{\gamma\mu}{1+\gamma\mu})}$$

$$\leq \tau - \frac{1 - (1 - \frac{\gamma^2\mu^2}{1+\gamma\mu}\tau + (\frac{\gamma^2\mu^2}{1+\gamma\mu})^2\frac{\tau(\tau-1)}{2})}{\frac{\gamma^2\mu^2}{1+\gamma\mu}}$$

$$= \frac{\gamma^2\mu^2}{1+\gamma\mu}\cdot\frac{\tau(\tau-1)}{2} \leq \frac{\gamma^2\mu^2}{1+\gamma\mu}\cdot\frac{\tau^2}{2}$$

Therefore, we conclude as below

$$\frac{\mu}{2}\mathbb{E}[\|w_{k,\tau}^m - w_k\|^2] + \frac{L}{2}\mathbb{E}[\|w_{k,\tau}^{\mathrm{ag},m} - w_k^{\mathrm{ag}}\|^2] \le \Big(\frac{4\gamma^2\mu(\mu+L)}{(1+\gamma\mu)^2} + \frac{2L\gamma^2(\mu+L)}{1+\gamma\mu}\Big)\tau^2\mathbb{E}[\Psi_k]$$
$$+ (\gamma^2\mu + \eta^2 L)\tau\sigma^2$$
$$+ \Big(\frac{(\gamma-\eta)^2\gamma^2\mu^2(\mu+L)}{(1+\gamma\mu)^2} + \frac{\gamma^4(\mu+L)^2 L}{1+\gamma\mu}\Big)\frac{\tau^3\sigma^2}{2}$$

*Proof of Lemma D.1*  By the definition of $\Psi_k$, $\Psi_{k,t}$ and Proposition D.2,

$$\mathbb{E}[\Psi_{k+1}] = \mathbb{E}[\Psi_{k,\tau}] + \frac{\mu}{2}\mathbb{E}[\|w_{k+1} - \bar{w}_{k,\tau}\|^2] + \mathbb{E}[F(w_{k+1}^{\mathrm{ag}}) - \frac{1}{M}\sum_{m=1}^M F(w_{k,\tau}^{\mathrm{ag},m})]$$

Applying Proposition D.3 and Proposition D.4, we have

$$\mathbb{E}[\Psi_{k+1}]$$
$$\le (1-\gamma\mu)^\tau\mathbb{E}[\Psi_k] + \frac{1}{2}(\eta^2 L + \frac{\gamma^2\mu}{M})\tau\sigma^2$$
$$+ \gamma\mu L\tau \cdot \max_{0\le t<\tau}\mathbb{E}[\frac{1}{M}\sum_{m=1}^M \|\bar{w}_{k,t}^{\mathrm{md}} - w_{k,t}^{\mathrm{md},m}\|\|\frac{1}{1+\gamma\mu}(\bar{w}_{k,t} - w_{k,t}^m) + \frac{\gamma\mu}{1+\gamma\mu}(\bar{w}_{k,t}^{\mathrm{ag}} - w_{k,t}^{\mathrm{ag},m})\|]$$
$$+ \frac{q\mu}{2M^2}\sum_{m=1}^M \mathbb{E}[\|w_{k,\tau}^m - w_k\|^2] + \frac{qL}{2M^2}\sum_{m=1}^M \mathbb{E}[\|w_{k,\tau}^{\mathrm{ag},m} - w_k^{\mathrm{ag}}\|^2]$$
$$\le (1-\gamma\mu)^\tau\mathbb{E}[\Psi_k] + \frac{1}{2}(\eta^2 L + \frac{\gamma^2\mu}{M})\tau\sigma^2$$
$$+ \gamma\mu L\tau \cdot \max_{0\le t<\tau}\mathbb{E}[\frac{1}{M}\sum_{m=1}^M \|\bar{w}_{k,t}^{\mathrm{md}} - w_{k,t}^{\mathrm{md},m}\|\|\frac{1}{1+\gamma\mu}(\bar{w}_{k,t} - w_{k,t}^m) + \frac{\gamma\mu}{1+\gamma\mu}(\bar{w}_{k,t}^{\mathrm{ag}} - w_{k,t}^{\mathrm{ag},m})\|]$$
$$+ \frac{q}{M}\Big[\Big(\frac{4\gamma^2\mu(\mu+L)}{(1+\gamma\mu)^2} + \frac{2L\gamma^2(\mu+L)}{1+\gamma\mu}\Big)\tau^2\mathbb{E}[\Psi_k] + (\gamma^2\mu + \eta^2 L)\tau\sigma^2$$
$$+ \Big(\frac{(\gamma-\eta)^2\gamma^2\mu^2(\mu+L)}{(1+\gamma\mu)^2} + \frac{\gamma^4(\mu+L)^2 L}{1+\gamma\mu}\Big)\frac{\tau^3\sigma^2}{2}\Big]$$
$$= \Big\{(1-\gamma\mu)^\tau + \frac{q}{M}\Big(\frac{4\gamma^2\mu(\mu+L)}{(1+\gamma\mu)^2} + \frac{2L\gamma^2(\mu+L)}{1+\gamma\mu}\Big)\tau^2\Big\}\mathbb{E}[\Psi_k] + \frac{1}{2}(\eta^2 L + \frac{\gamma^2\mu}{M})\tau\sigma^2$$
$$+ \frac{q}{M}(\gamma^2\mu + \eta^2 L)\tau\sigma^2 + \frac{q}{2M}\Big(\frac{(\gamma-\eta)^2\gamma^2\mu^2(\mu+L)}{(1+\gamma\mu)^2} + \frac{\gamma^4(\mu+L)^2 L}{1+\gamma\mu}\Big)\tau^3\sigma^2$$
$$+ \gamma\mu L\tau \cdot \max_{0\le t<\tau}\mathbb{E}[\frac{1}{M}\sum_{m=1}^M \|\bar{w}_{k,t}^{\mathrm{md}} - w_{k,t}^{\mathrm{md},m}\|\|\frac{1}{1+\gamma\mu}(\bar{w}_{k,t} - w_{k,t}^m) + \frac{\gamma\mu}{1+\gamma\mu}(\bar{w}_{k,t}^{\mathrm{ag}} - w_{k,t}^{\mathrm{ag},m})\|]$$

The second inequality comes from Proposition D.6. Then, let's define $C(\gamma, \tau)$ as

$$C(\gamma, \tau) = (1-\gamma\mu)^\tau + \frac{q}{M}\Big(\frac{4\gamma^2\mu(\mu+L)}{(1+\gamma\mu)^2} + \frac{2L\gamma^2(\mu+L)}{1+\gamma\mu}\Big)\tau^2$$

Finally, we obtain

$$\mathbb{E}[\Psi_{k+1}] \le C(\gamma, \tau)\mathbb{E}[\Psi_k] + \frac{1}{2}(\eta^2 L + \frac{\gamma^2\mu}{M})\tau\sigma^2 + \frac{q}{M}(\gamma^2\mu + \eta^2 L)\tau\sigma^2$$
$$+ \frac{q}{2M}\Big(\frac{(\gamma-\eta)^2\gamma^2\mu^2(\mu+L)}{(1+\gamma\mu)^2} + \frac{\gamma^4(\mu+L)^2 L}{1+\gamma\mu}\Big)\tau^3\sigma^2 + \gamma\mu L\tau$$
$$\cdot \max_{0\le t<\tau}\mathbb{E}[\frac{1}{M}\sum_{m=1}^M \|\bar{w}_{k,t}^{\mathrm{md}} - w_{k,t}^{\mathrm{md},m}\|\|\frac{1}{1+\gamma\mu}(\bar{w}_{k,t} - w_{k,t}^m) + \frac{\gamma\mu}{1+\gamma\mu}(\bar{w}_{k,t}^{\mathrm{ag}} - w_{k,t}^{\mathrm{ag},m})\|]$$

## D.2 Proof of Theorem D.7

**Theorem D.7.** *Let $F$ be $\mu$-strongly convex, and assume Assumption 2.1, 2.2, 2.3, 2.4, then for $\alpha = \frac{1}{\gamma\mu}, \beta = \alpha + 1, \gamma = \max(\eta, \sqrt{\frac{\eta}{\mu\tau}}), \eta, \gamma \in (0, \frac{1}{L}], \tau \geq 2$, if the learning rate $\gamma$ satisfies*

$$\Big(\mu^2 + \frac{q}{M}(\mu + L)(4\mu + 2L)\Big)\gamma\tau \leq \frac{1}{2}\mu \tag{10}$$

*FedAQ yields*

$$\mathbb{E}[\Psi_K] \leq \exp\Big(-\frac{1}{2}\max(\eta\mu, \sqrt{\frac{\eta\mu}{\tau}})K\tau\Big)\Psi_0 + (2q+1)(\frac{\eta^{\frac{1}{2}}\sigma^2}{\mu^{\frac{1}{2}}M\tau^{\frac{1}{2}}} + \frac{\eta\sigma^2}{M}) + 14\eta^2 L\tau\sigma^2$$

$$+ \frac{(780 + \frac{2q}{M})\eta^{\frac{3}{2}}L\tau^{\frac{1}{2}}\sigma^2}{\mu^{\frac{1}{2}}} + \frac{(\mu + L)(\mu^2 + \mu L + L^2)q\eta^{\frac{3}{2}}\tau^{\frac{1}{2}}\sigma^2}{\mu^{\frac{5}{2}}M} + \frac{q\eta^3\tau^2(\mu + L)^2 L\sigma^2}{\mu M}$$

*Proof of Theorem D.7* At first, due to the condition (10) in Theorem D.7, we get

$$C(\gamma, \tau) = (1 - \gamma\mu)^\tau + \frac{q}{M}\Big(\frac{4\gamma^2\mu(\mu + L)}{(1 + \gamma\mu)^2} + \frac{2L\gamma^2(\mu + L)}{1 + \gamma\mu}\Big)\tau^2$$

$$\leq 1 - \gamma\mu\tau + \gamma^2\mu^2\tau^2 + \frac{q}{M}\gamma^2(\mu + L)(4\mu + 2L)\tau^2$$

$$= 1 - \gamma\mu\tau + \Big(\mu^2 + \frac{q}{M}(\mu + L)(4\mu + 2L)\Big)\gamma^2\tau^2$$

$$\leq 1 - \frac{1}{2}\gamma\mu\tau \ (\because \text{ condition (10)})$$

The first inequality comes from the fact that $(1 - \gamma\mu)^\tau \leq e^{-\gamma\mu\tau} \leq 1 - \gamma\mu\tau + \gamma^2\mu^2\tau^2$ when $0 \leq \gamma\mu \leq 1$. Also, it is trivial that $\gamma = \max(\eta, \sqrt{\frac{\eta}{\mu\tau}}) \in [\eta, \sqrt{\frac{\eta}{\mu}}]$. Thus, we can use Lemma D.1. By using Lemma D.1 and the above result, we obtain

$$\mathbb{E}[\Psi_{k+1}] \leq (1 - \frac{1}{2}\gamma\mu\tau)\mathbb{E}[\Psi_k] + \frac{1}{2}(\eta^2 L + \frac{\gamma^2\mu}{M})\tau\sigma^2$$

$$+ \frac{q}{M}(\gamma^2\mu + \eta^2 L)\tau\sigma^2 + \frac{q}{2M}\Big(\frac{(\gamma - \eta)^2\gamma^2\mu^2(\mu + L)}{(1 + \gamma\mu)^2} + \frac{\gamma^4(\mu + L)^2 L}{1 + \gamma\mu}\Big)\tau^3\sigma^2 + \gamma\mu L\tau$$

$$\cdot \max_{0 \leq t < \tau} \mathbb{E}[\frac{1}{M}\sum_{m=1}^M \|\bar{w}_{k,t}^{\text{md}} - w_{k,t}^{\text{md},m}\| \|\frac{1}{1 + \gamma\mu}(\bar{w}_{k,t} - w_{k,t}^m) + \frac{\gamma\mu}{1 + \gamma\mu}(\bar{w}_{k,t}^{\text{ag}} - w_{k,t}^{\text{ag},m})\|]$$

$$\tag{11}$$

By the Lemma B.3 in Yuan and Ma (2020), we know that the below quantity is bounded.

$$\max_{0 \leq t < \tau} \mathbb{E}[\frac{1}{M}\sum_{m=1}^M \|\bar{w}_{k,t}^{\text{md}} - w_{k,t}^{\text{md},m}\| \|\frac{1}{1 + \gamma\mu}(\bar{w}_{k,t} - w_{k,t}^m) + \frac{\gamma\mu}{1 + \gamma\mu}(\bar{w}_{k,t}^{\text{ag}} - w_{k,t}^{\text{ag},m})\|] \leq B$$

$$B = \begin{cases} 7\eta\gamma\tau\sigma^2\Big(1 + \frac{2\gamma^2\mu}{\eta}\Big)^{2\tau}, & \text{if } \gamma \in \Big(\eta, \sqrt{\frac{\eta}{\mu}}\Big] \\ 7\eta^2\tau\sigma^2, & \text{if } \gamma = \eta \end{cases}$$

Telescoping (11) yields

$$\mathbb{E}[\Psi_K] \leq (1 - \frac{1}{2}\gamma\mu\tau)^K\Psi_0 + \Big(\sum_{k'=0}^{K-1}(1 - \frac{1}{2}\gamma\mu\tau)^{k'}\Big) \cdot \Big[\frac{1}{2}(\eta^2 L + \frac{\gamma^2\mu}{M})\tau\sigma^2 + \gamma\mu L\tau B$$

$$+ \frac{q}{M}(\gamma^2\mu + \eta^2 L)\tau\sigma^2 + \frac{q}{2M}\Big(\frac{(\gamma - \eta)^2\gamma^2\mu^2(\mu + L)}{(1 + \gamma\mu)^2} + \frac{\gamma^4(\mu + L)^2 L}{1 + \gamma\mu}\Big)\tau^3\sigma^2\Big]$$

$$\leq \exp\Big(-\frac{\gamma\mu\tau K}{2}\Big)\Psi_0 + \frac{\eta^2 L\sigma^2}{\gamma\mu} + \frac{\gamma\sigma^2}{M} + 2LB + 2q\Big(\frac{\gamma\sigma^2}{M} + \frac{\eta^2 L\sigma^2}{\gamma\mu M}\Big)$$

$$+ \frac{q}{M}\Big(\frac{(\gamma - \eta)^2\gamma\mu(\mu + L)}{(1 + \gamma\mu)^2} + \frac{\gamma^3(\mu + L)^2 L}{(1 + \gamma\mu)\mu}\Big)\tau^2\sigma^2$$

The last inequality comes from the fact that $\sum_{k'=0}^{K-1}(1 - \frac{1}{2}\gamma\mu\tau)^{k'} \leq \frac{2}{\gamma\mu\tau}$. Since we plug in $\gamma = \max(\eta, \sqrt{\frac{\eta}{\mu\tau}})$, we can use Lemma B.4 in Yuan and Ma (2020). Therefore, we obtain

$$\mathbb{E}[\Psi_K] \leq \exp\left(-\frac{1}{2}\max(\eta\mu, \sqrt{\frac{\eta\mu}{\tau}})K\tau\right)\Psi_0 + \frac{\eta^{\frac{1}{2}}\sigma^2}{\mu^{\frac{1}{2}}M\tau^{\frac{1}{2}}} + \frac{\eta\sigma^2}{M} + \frac{780\eta^{\frac{3}{2}}L\tau^{\frac{1}{2}}\sigma^2}{\mu^{\frac{1}{2}}} + 14\eta^2 L\tau\sigma^2$$

$$+ \max\left(\frac{2q\eta^{\frac{1}{2}}\sigma^2}{M\mu^{\frac{1}{2}}\tau^{\frac{1}{2}}}, \frac{2q\eta\sigma^2}{M}\right) + \min\left(\frac{2q\eta^{\frac{3}{2}}\tau^{\frac{1}{2}}L\sigma^2}{M\mu^{\frac{1}{2}}}, \frac{2q\eta L\sigma^2}{M\mu}\right)$$

$$+ \frac{q\tau^2\sigma^2}{M}\max\left(\frac{\eta^{\frac{3}{2}}\mu(\mu+L)}{\mu^{\frac{3}{2}}\tau^{\frac{3}{2}}} + \frac{\eta^{\frac{3}{2}}(\mu+L)^2 L}{\mu^{\frac{5}{2}}\tau^{\frac{3}{2}}}, \frac{\eta^3(\mu+L)^2 L}{\mu}\right)$$

The first term stems directly from Lemma B.4 in Yuan and Ma (2020). Also, the last term comes from the fact that

$$\frac{(\gamma-\eta)^2\gamma\mu(\mu+L)}{(1+\gamma\mu)^2} + \frac{\gamma^3(\mu+L)^2 L}{(1+\gamma\mu)\mu} \leq \begin{cases} \gamma^3\mu(\mu+L) + \frac{\gamma^3(\mu+L)^2 L}{\mu}, & \text{if } \gamma \neq \eta \\ \frac{\eta^3(\mu+L)^2 L}{\mu}, & \text{if } \gamma = \eta \end{cases}$$

Therefore, by simple inequalities such as $\max(a,b) \leq a+b$ and $\min(a,b) \leq a$, we ultimately get

$$\mathbb{E}[\Psi_K] \leq \exp\left(-\frac{1}{2}\max(\eta\mu, \sqrt{\frac{\eta\mu}{\tau}})K\tau\right)\Psi_0 + \frac{(2q+1)\eta^{\frac{1}{2}}\sigma^2}{\mu^{\frac{1}{2}}M\tau^{\frac{1}{2}}} + \frac{(2q+1)\eta\sigma^2}{M} + 14\eta^2 L\tau\sigma^2$$

$$+ \frac{(780+\frac{2q}{M})\eta^{\frac{3}{2}}L\tau^{\frac{1}{2}}\sigma^2}{\mu^{\frac{1}{2}}} + \frac{(\mu+L)(\mu^2+\mu L+L^2)q\eta^{\frac{3}{2}}\tau^{\frac{1}{2}}\sigma^2}{\mu^{\frac{5}{2}}M} + \frac{q\eta^3\tau^2(\mu+L)^2 L\sigma^2}{\mu M} \tag{12}$$

## D.3 Proof of Corollary D.8

**Corollary D.8.** *Let $C_1, C_2$, and $\eta_0$ as below. Note that $T = K\tau$.*

$$C_1 = \frac{(\mu+L)(\mu^2+\mu L+L^2)q}{\mu^{\frac{5}{2}}}, \; C_2 = \frac{q(\mu+L)^2 L}{\mu}$$

$$\eta_0 = \frac{4\tau}{\mu T^2}\log^2\left(e + \min(\frac{\mu M T\Psi_0}{(2q+1)\sigma^2}, \frac{\mu^2 T^3\Psi_0}{L\tau^2\sigma^2}, \frac{\mu^3 M T^3\Psi_0}{(\mu^{\frac{3}{2}}C_1+8C_2)\tau^2\sigma^2})\right)$$

*Then for $\eta = \min(\frac{1}{L}, \eta_0)$, FedAQ yields*

$$\mathbb{E}[\Psi_K] \leq \min\left(\exp(-\frac{\mu T}{2L}), \exp(-\frac{\mu^{\frac{1}{2}}T}{2L^{\frac{1}{2}}\tau^{\frac{1}{2}}})\right)\Psi_0$$

$$+ \frac{7(2q+1)\sigma^2}{\mu M T}\log^2\left(e + \frac{\mu M T\Psi_0}{(2q+1)\sigma^2}\right) \tag{13}$$

$$+ \frac{(6465+\frac{16q}{M})L\tau^2\sigma^2}{\mu^2 T^3}\log^4\left(e + \frac{\mu^2 T^3\Psi_0}{L\tau^2\sigma^2}\right) \tag{14}$$

$$+ \frac{9(\mu^{\frac{3}{2}}C_1+8C_2)\tau^2\sigma^2}{\mu^3 M T^3}\log^6\left(e + \frac{\mu^3 M T^3\Psi_0}{(\mu^{\frac{3}{2}}C_1+8C_2)\tau^2\sigma^2}\right) \tag{15}$$

*Proof of Corollary D.8* Let's decompose the final result (12) of the Theorem D.7 into a decreasing term and an increasing term. We denote the decreasing term $\psi_1$ and the increasing term $\psi_2$ as below.

$$\psi_1(\eta) = \exp\left(-\frac{1}{2}\max(\eta\mu, \sqrt{\frac{\eta\mu}{\tau}})T\right)\Psi_0$$

$$\psi_2(\eta) = \frac{(2q+1)\eta^{\frac{1}{2}}\sigma^2}{\mu^{\frac{1}{2}}M\tau^{\frac{1}{2}}} + \frac{(2q+1)\eta\sigma^2}{M} + \frac{(780+\frac{2q}{M})\eta^{\frac{3}{2}}L\tau^{\frac{1}{2}}\sigma^2}{\mu^{\frac{1}{2}}} + 14\eta^2 L\tau\sigma^2$$

$$+ \frac{(\mu+L)(\mu^2+\mu L+L^2)q\eta^{\frac{3}{2}}\tau^{\frac{1}{2}}\sigma^2}{\mu^{\frac{5}{2}}M} + \frac{q\eta^3\tau^2(\mu+L)^2 L\sigma^2}{\mu M}$$

Since $\psi_1$ is the decreasing term, we have

$$\psi_1(\eta) \le \psi_1(\frac{1}{L}) + \psi_1(\eta_0) \tag{16}$$

where

$$\psi_1(\frac{1}{L}) = \min\Big(\exp(-\frac{\mu T}{2L}), \exp(-\frac{\mu^{\frac{1}{2}}T}{2L^{\frac{1}{2}}\tau^{\frac{1}{2}}})\Big)\Psi_0$$

$$\psi_1(\eta_0) \le \exp\Big(-\frac{1}{2}\sqrt{\frac{\eta_0\mu}{\tau}}T\Big)$$

$$= \Big(e + \min(\frac{\mu MT\Psi_0}{(2q+1)\sigma^2}, \frac{\mu^2 T^3\Psi_0}{L\tau^2\sigma^2}, \frac{\mu^3 MT^3\Psi_0}{(\mu^{\frac{3}{2}}C_1 + 8C_2)\tau^2\sigma^2})\Big)^{-1}\Psi_0$$

$$\le \frac{(2q+1)\sigma^2}{\mu MT} + \frac{L\tau^2\sigma^2}{\mu^2 T^3} + \frac{(\mu^{\frac{3}{2}}C_1 + 8C_2)\tau^2\sigma^2}{\mu^3 MT^3}$$

Since $\psi_2$ is the increasing term, we have

$$\psi_2(\eta)$$
$$\le \psi_2(\eta_0)$$
$$\le \frac{2(2q+1)\sigma^2}{\mu MT}\log\Big(e + \frac{\mu MT\Psi_0}{(2q+1)\sigma^2}\Big) + \frac{4(2q+1)\tau\sigma^2}{\mu MT^2}\log^2\Big(e + \frac{\mu MT\Psi_0}{(2q+1)\sigma^2}\Big)$$
$$+ \frac{8(780 + \frac{2q}{M})L\tau^2\sigma^2}{\mu^2 T^3}\log^3\Big(e + \frac{\mu^2 T^3\Psi_0}{L\tau^2\sigma^2}\Big) + \frac{224L\tau^3\sigma^2}{\mu^2 T^4}\log^4\Big(e + \frac{\mu^2 T^3\Psi_0}{L\tau^2\sigma^2}\Big)$$
$$+ \frac{8C_1\tau^2\sigma^2}{\mu^{\frac{3}{2}}MT^3}\log^3\Big(e + \frac{\mu^3 MT^3\Psi_0}{(\mu^{\frac{3}{2}}C_1 + 8C_2)\tau^2\sigma^2}\Big) + \frac{64C_2\tau^5\sigma^2}{\mu^3 MT^6}\log^6\Big(e + \frac{\mu^3 MT^3\Psi_0}{(\mu^{\frac{3}{2}}C_1 + 8C_2)\tau^2\sigma^2}\Big)$$
$$\le \frac{6(2q+1)\sigma^2}{\mu MT}\log^2\Big(e + \frac{\mu MT\Psi_0}{(2q+1)\sigma^2}\Big) + \frac{(6464 + \frac{16q}{M})L\tau^2\sigma^2}{\mu^2 T^3}\log^4\Big(e + \frac{\mu^2 T^3\Psi_0}{L\tau^2\sigma^2}\Big)$$
$$+ \frac{8(\mu^{\frac{3}{2}}C_1 + 8C_2)\tau^2\sigma^2}{\mu^3 MT^3}\log^6\Big(e + \frac{\mu^3 MT^3\Psi_0}{(\mu^{\frac{3}{2}}C_1 + 8C_2)\tau^2\sigma^2}\Big) \tag{17}$$

The last inequality comes from $\frac{\tau}{T} \le 1$. Therefore, by combining (16) and (17), we finally get

$$\mathbb{E}[\Psi_K] \le \psi_1(\eta) + \psi_2(\eta)$$
$$\le \psi_1(\frac{1}{L}) + \psi_1(\eta_0) + \psi_2(\eta_0)$$
$$\le \min\Big(\exp(-\frac{\mu T}{2L}), \exp(-\frac{\mu^{\frac{1}{2}}T}{2L^{\frac{1}{2}}\tau^{\frac{1}{2}}})\Big)\Psi_0 + \frac{7(2q+1)\sigma^2}{\mu MT}\log^2\Big(e + \frac{\mu MT\Psi_0}{(2q+1)\sigma^2}\Big)$$
$$+ \frac{(6465 + \frac{16q}{M})L\tau^2\sigma^2}{\mu^2 T^3}\log^4\Big(e + \frac{\mu^2 T^3\Psi_0}{L\tau^2\sigma^2}\Big)$$
$$+ \frac{9(\mu^{\frac{3}{2}}C_1 + 8C_2)\tau^2\sigma^2}{\mu^3 MT^3}\log^6\Big(e + \frac{\mu^3 MT^3\Psi_0}{(\mu^{\frac{3}{2}}C_1 + 8C_2)\tau^2\sigma^2}\Big)$$

### D.4 Why the condition (10) is satisfied

The synchronization rounds $K$ required for linear speedup in $M$ for FedAQ is $\tilde{\mathcal{O}}((\frac{M}{1+q})^{\frac{1}{2}})$ (See Remark 4.2). Since we derive this result from Theorem D.7, we should show that $K = \tilde{\mathcal{O}}((\frac{M}{1+q})^{\frac{1}{2}})$ satisfies the condition (10) in Theorem D.7.

$$\Big(\mu^2 + \frac{q}{M}(\mu + L)(4\mu + 2L)\Big)\gamma\tau \le \frac{1}{2}\mu$$

We rewrite the above condition as below.

$$\gamma\tau \le \frac{\mu}{2\mu^2 + \frac{2q}{M}(\mu + L)(4\mu + 2L)} \tag{18}$$

We know $\gamma = \max(\eta, \sqrt{\frac{\eta}{\mu\tau}})$ and $\eta = \min(\frac{1}{L}, \eta_0)$. Since $\eta_0$ becomes smaller and smaller as T increases, we assume $\eta = \eta_0$ here. Therefore, we get

$$
\begin{aligned}
\gamma\tau &= \max(\eta_0\tau, \sqrt{\frac{\eta_0\tau}{\mu}}) \\
&= \max\Big(\frac{4\tau^2}{\mu T^2}\log^2\Big(e + \min(\frac{\mu MT\Psi_0}{(2q+1)\sigma^2}, \frac{\mu^2 T^3\Psi_0}{L\tau^2\sigma^2}, \frac{\mu^3 MT^3\Psi_0}{(\mu^{\frac{3}{2}}C_1 + 8C_2)\tau^2\sigma^2})\Big), \\
&\quad \frac{2\tau}{\mu T}\log\Big(e + \min(\frac{\mu MT\Psi_0}{(2q+1)\sigma^2}, \frac{\mu^2 T^3\Psi_0}{L\tau^2\sigma^2}, \frac{\mu^3 MT^3\Psi_0}{(\mu^{\frac{3}{2}}C_1 + 8C_2)\tau^2\sigma^2})\Big)\Big)
\end{aligned}
$$

Note that $K = \frac{T}{\tau} = \tilde{\mathcal{O}}((\frac{M}{1+q})^{\frac{1}{2}}) = C(\frac{M}{1+q})^{\frac{1}{2}}\log(T)$ because $\tilde{\mathcal{O}}$ contains hidden multiplicative polylog factors with respect to $T$. We can assume $T$ is sufficiently large here. Then, we have

$$
\begin{aligned}
\gamma\tau &= \max\Big(\frac{4(1+q)}{\mu C^2 M\log^2(T)}\log^2\Big(e + \min(\frac{\mu MT\Psi_0}{(2q+1)\sigma^2}, \frac{\mu^2 T^3\Psi_0}{L\tau^2\sigma^2}, \frac{\mu^3 MT^3\Psi_0}{(\mu^{\frac{3}{2}}C_1 + 8C_2)\tau^2\sigma^2})\Big), \\
&\quad \frac{2(1+q)^{\frac{1}{2}}}{\mu C M^{\frac{1}{2}}\log(T)}\log\Big(e + \min(\frac{\mu MT\Psi_0}{(2q+1)\sigma^2}, \frac{\mu^2 T^3\Psi_0}{L\tau^2\sigma^2}, \frac{\mu^3 MT^3\Psi_0}{(\mu^{\frac{3}{2}}C_1 + 8C_2)\tau^2\sigma^2})\Big)\Big) \\
&\leq \max\Big(\frac{4(1+q)}{\mu C^2 M\log^2(T)}\log^2\Big(\frac{2\mu MT\Psi_0}{(2q+1)\sigma^2}\Big), \frac{2(1+q)^{\frac{1}{2}}}{\mu C M^{\frac{1}{2}}\log(T)}\log\Big(\frac{2\mu MT\Psi_0}{(2q+1)\sigma^2}\Big)\Big)
\end{aligned}
$$

For an arbitrary constant $k_1 > 0$, it is easy to show that $\lim_{T\to\infty}\frac{\log(k_1 T)}{\log(T)} = 1$. Thus, we obtain

$$
\begin{aligned}
\gamma\tau &\leq \max\Big(\frac{4(1+q)}{\mu C^2 M\log^2(T)}\log^2\Big(\frac{2\mu MT\Psi_0}{(2q+1)\sigma^2}\Big), \frac{2(1+q)^{\frac{1}{2}}}{\mu C M^{\frac{1}{2}}\log(T)}\log\Big(\frac{2\mu MT\Psi_0}{(2q+1)\sigma^2}\Big)\Big) \\
&\simeq \max\Big(\frac{4(1+q)}{\mu C^2 M}, \frac{2(1+q)^{\frac{1}{2}}}{\mu C M^{\frac{1}{2}}}\Big) \\
&\leq \frac{\mu}{2\mu^2 + \frac{2q}{M}(\mu + L)(4\mu + 2L)}
\end{aligned}
$$

Finally, we conclude that there exists a constant $C$ that meets the last inequality. Therefore, $K = \tilde{\mathcal{O}}((\frac{M}{1+q})^{\frac{1}{2}})$ satisfies the condition (10).

# E    Details about Haddadpour et al. (2021) & contribution 2 in Introduction

### E.1    Why Haddadpour et al. (2021) cannot achieve a linear speedup

It is hard to say that Haddadpour et al. (2021) achieves a linear speedup in $M$ in strongly-convex and homogeneous settings. Let's first recap Corollary D.8 in Haddadpour et al. (2021). They let $\eta\gamma\mu\tau \leq \frac{1}{2}, \kappa = \frac{L}{\mu}, \gamma \geq M$ and tune $\eta$ as $\eta = \frac{1}{2L(\frac{q}{M}+1)\tau\gamma}$. Here, $\eta$ is the client learning rate, and $\gamma$ is the server learning rate. Other parameters are the same as we defined. Then, they obtain the below result.

$$
\mathbb{E}[F(w_K) - F^*] \leq \exp(-\eta\gamma\mu\tau K)(F(w_0) - F^*) + \frac{1}{\mu}\Big[\frac{1}{2}\tau L^2\eta^2\sigma^2 + (1+q)\frac{\gamma\eta L\sigma^2}{2M}\Big] \quad (19)
$$

$$
\leq \mathcal{O}\Big(\exp(-\frac{K}{2(\frac{q}{M}+1)\kappa})(F(w_0) - F^*) + \frac{\sigma^2}{\gamma^2\mu\tau} + \frac{(q+1)\sigma^2}{\mu(\frac{q}{M}+1)\tau M}\Big)
$$

$$
= \mathcal{O}\Big(\exp(-\frac{K}{2(\frac{q}{M}+1)\kappa})(F(w_0) - F^*) + \frac{\sigma^2 K}{\gamma^2\mu T} + \frac{(q+1)K\sigma^2}{\mu(\frac{q}{M}+1)TM}\Big)
$$

Let's focus on the second and third term. We assume $M$ is large enough and represent them only with $\gamma, K, T, M$ to easily check the linear speedup of this convergence rate. Then, we obtain

$$
\mathcal{O}\Big(\frac{K}{\gamma^2 T} + \frac{K}{MT}\Big) \leq \mathcal{O}\Big(\frac{K}{M^2 T} + \frac{K}{MT}\Big) \; (\because \gamma \geq M) \quad (20)
$$

Thus, it seemingly achieves a linear speedup in $M$ when $K$ is just a constant. However, we are missing the critical point in this analysis. To be specific, let's consider the case when $\gamma = 1$. Then, the convergence rate (20) changes into $\mathcal{O}\left(\frac{K}{T} + \frac{K}{MT}\right)$ that cannot achieve a linear speedup in $M$. This is implausible because the convergence rate (19) becomes tighter when $\gamma = 1$ than $\gamma \geq M$ (See the last term of (19)). Actually, we can achieve a linear speedup in $M$ when $\gamma = 1$ if we tune $\eta = \frac{1}{2L(\frac{q}{M}+1)\tau M}$. However, this is not an appropriate tuning because there is $M$ in the denominator. Similarly, Haddadpour et al. (2021) tunes $\eta = \frac{1}{2L(\frac{q}{M}+1)\tau\gamma}$ where $\gamma \geq M$. Even though there is no $M$ in the denominator, the condition $\gamma \geq M$ makes the convergence rate achieve a linear speedup without any theoretical benefits of the algorithm. Therefore, we cannot say their $\eta$ makes their algorithm achieve a linear speedup in $M$. We should tune in a different way that does not contain $M$ in a denominator. For reference, our tuning parameter $\eta$ for the FedAQ algorithm does not contain $M$ in the denominator (See Corollary D.8 and Corollary F.8).

### E.2 New convergence rate for Haddadpour et al. (2021)

We propose new $\eta$ and convergence rate for Haddadpour et al. (2021). This new $\eta$ makes the algorithm achieve a linear speedup in $M$. Let's denote $\Phi_0 = F(w_0) - F^*$. We also know that $T = K\tau$. Then, we choose $\eta$ as

$$\eta = \frac{1}{\gamma\mu T} \log\left(e + \min(\frac{\gamma^2\mu^3 T^2 \Phi_0}{\tau L^2\sigma^2}, \frac{\mu^2 MT\Phi_0}{(1+q)L\sigma^2})\right)$$

We plug in this $\eta$ to (19). We bound the first term as below.

$$\exp(-\eta\gamma\mu\tau K)(F(w_0) - F^*) = \left(e + \min(\frac{\gamma^2\mu^3 T^2\Phi_0}{\tau L^2\sigma^2}, \frac{\mu^2 MT\Phi_0}{(1+q)L\sigma^2})\right)^{-1}\Phi_0$$

$$\leq \frac{\tau L^2\sigma^2}{\gamma^2\mu^3 T^2} + \frac{(1+q)L\sigma^2}{\mu^2 MT}$$

The another terms are bounded as below.

$$\frac{1}{\mu}\left[\frac{1}{2}\tau L^2\eta^2\sigma^2 + (1+q)\frac{\gamma\eta L\sigma^2}{2M}\right] \leq \frac{\tau L^2\sigma^2}{2\gamma^2\mu^3 T^2}\log^2\left(e + \frac{\gamma^2\mu^3 T^2\Phi_0}{\tau L^2\sigma^2}\right)$$

$$+ \frac{(1+q)L\sigma^2}{2\mu^2 MT}\log\left(e + \frac{\mu^2 MT\Phi_0}{(1+q)L\sigma^2}\right)$$

Thus, we obtain a new convergence rate by combining the above two bounds.

$$\mathbb{E}[F(w_K) - F^*] \leq \exp(-\eta\gamma\mu\tau K)(F(w_0) - F^*) + \frac{1}{\mu}\left[\frac{1}{2}\tau L^2\eta^2\sigma^2 + (1+q)\frac{\gamma\eta L\sigma^2}{2M}\right]$$

$$\leq \frac{3\tau L^2\sigma^2}{2\gamma^2\mu^3 T^2}\log^2\left(e + \frac{\gamma^2\mu^3 T^2\Phi_0}{\tau L^2\sigma^2}\right) + \frac{3(1+q)L\sigma^2}{2\mu^2 MT}\log\left(e + \frac{\mu^2 MT\Phi_0}{(1+q)L\sigma^2}\right)$$

Here, we replace $\tau$ with $\frac{T}{K}$. Then, we represent the above convergence rate with only $T, K, M, q$.

$$\tilde{\mathcal{O}}(\frac{1}{TK} + \frac{1+q}{MT})$$

This is the new convergence rate we propose for Haddadpour et al. (2021). We also get $K = \tilde{\mathcal{O}}(\frac{M}{1+q})$ communication rounds make this algorithm achieve a linear speedup in $M$.

### E.3 More details on contribution 2 in Introduction

**Low-precision quantizer (Alistarh et al. (2017))** Given $x \in \mathbb{R}^d$, the quantizer $Q : \mathbb{R}^d \to \mathbb{R}^d$ is defined by

$$Q_i(x) = \text{sign}(x_i) \cdot \|x\| \cdot \xi_i(x, s), \quad i \in [d]$$

$\xi_i$ is defined as below.

$$\xi_i(x, s) = \begin{cases} \frac{l+1}{s}, & \text{with probability } \frac{|x_i|}{\|x\|}s - l \\ \frac{l}{s}, & \text{o/w} \end{cases}$$

$s$ is the number of quantization levels. $l \in [0, s)$ is an integer which satisfies $\frac{|x_i|}{\|x\|} \in [\frac{l}{s}, \frac{l+1}{s})$.

**More details on $d_{\text{quant}}$**   This paragraph explains why FedAQ needs to send only $d_{\text{quant}} = O(\log \frac{1}{q})$ bits for each value. We use the result of Lemma 3.1 in Alistarh et al. (2017). They show the below result with a low-precision quantizer.

$$\mathbb{E}[\|Q(x,s) - x\|_2^2] \leq \min(\frac{n}{s^2}, \frac{\sqrt{n}}{s})\|x\|_2^2$$

where $n$ is the dimension of $x$, and $s$ is the number of quantization levels. Then, we regard $q$ as

$$q = \frac{\sqrt{n}}{s} = \frac{\sqrt{n}}{2^{d_{\text{quant}}}} \tag{21}$$

Thus, we obtain the following conclusion.

$$d_{\text{quant}} = \frac{\frac{1}{2}\log n + \log \frac{1}{q}}{\log 2} = O(\log \frac{1}{q})$$

**Comparing FedAQ to FedAC**   We compare computation and communication efficiency of FedAC-II and FedAQ under the condition set (2) to achieve the same error. Let's recall the convergence rate of FedAC and FedAQ. The convergence rate of FedAC and FedAQ is respectively $\tilde{\mathcal{O}}(\frac{1}{MT} + \frac{1}{TK^3})$ and $\tilde{\mathcal{O}}(\frac{1+q}{MT} + \frac{1+q}{TK^3})$. Let's say FedAC requires $T$ iterations and $K = M^{\frac{1}{3}}$ communication rounds to achieve the error $\frac{1}{MT}$. Then, FedAQ requires

$$T' = (1 + q)T, \ K' = M^{\frac{1}{3}}$$

to achieve the same error $\frac{1}{MT}$. This means FedAQ needs $1 + q$ times more local steps and the same number of communication rounds to achieve the same error of FedAC. These local steps do no require any communication with the server hence can be performed without any additional communication overhead.

From discussion in the previous section, if we use the simple low-precision quantizer, we need only $d_{\text{quant}} = O(\log \frac{1}{q})$ bits for communicating values with enough precision that can lead to an error rate of $O(\frac{1}{MT})$. In comparison, FedAC would require $O(\log(MT))$ bits to maintain enough precision to achieve the same error rate. In a majority of tasks in the real world, 32 bits are usually enough for $d_{\text{full}}$ to achieve enough precision as we usually don't need converge to a very small error rate. Nonetheless, even if we compare FedAQ(8bits) with to FedAC(32bits), we argue that the overall benefit from less communication by quantization is more influential than the slowdown effect from quantization.

For example, if we consider a $l_2$-regularized logistic regression model for MNIST (strongly convex experiment) and quantize from 32 bits to $d_{\text{quant}} = 8$ bits. Here, $n = 784 \times 10$. We get the following results by using (21).

$$1 + q = 1 + \frac{\sqrt{n}}{2^{d_{\text{quant}}}} = 1 + \frac{\sqrt{7840}}{2^8} \simeq 1.346,$$

On the other hand, the ratio of data communicated by FedAC and FedAQ is

$$\frac{32}{d_{\text{quant}}} = 4$$

In contribution 2, we claim $1 + q \ll \frac{d_{\text{full}}}{d_{\text{quant}}}$ because $d_{\text{full}}$ is unbounded as $T$ goes to infinity. In the real world example, $\frac{d_{\text{full}}}{d_{\text{quant}}} = 4$ is still much greater than $1 + q$. Furthermore, since the local computation is much cheaper than data communication, we conclude that the benefit from less communication by quantization (4 times less bits) overwhelm the slowdown effect from quantization ($(1 + q)$ times more local computation).

## F   Further analysis of FedAQ under the parameter condition set (2)

We use notations defined in Appx. D here as well. We newly define $\Phi_{k,t}^m, \Phi_{k,t}, \Phi_k, B_{k,t}^m$ as below.

$$\Phi_{k,t}^m = F(w_{k,t}^{\mathrm{ag},m}) - F^* + \frac{1}{6}\mu\|w_{k,t}^m - w^*\|^2$$

$$\Phi_{k,t} = F(\bar{w}_{k,t}^{\mathrm{ag}}) - F^* + \frac{1}{6}\mu\|\bar{w}_{k,t} - w^*\|^2$$

$$\Phi_k := \Phi_{k,0} = F(w_k^{\mathrm{ag}}) - F^* + \frac{1}{6}\mu\|w_k - w^*\|^2$$

$$B_{k,t}^m = \left(\frac{\mu\alpha^{-2}}{3}(1 - \beta^{-1})^2 + L\beta^{-2}\right)\|w_{k,t}^m - w_{k,t}^{\mathrm{ag},m}\|^2 + \gamma^2(\frac{\mu}{3} + L)\frac{2\alpha^2 - \alpha}{2\alpha^2 - 1} \cdot 2L\Phi_{k,t}^m$$

The flow of proof is similar to Appx. D. We need one more condition $\gamma\mu \leq \frac{3}{4}$ to show the convergence of FedAQ under the parameter condition set (2).

### F.1   Proof of Lemma F.1

**Lemma F.1.** *Let $F$ be $\mu$-strongly convex, and assume Assumption 2.1, 2.2, 2.3, 2.4, then for* $\alpha = \frac{3}{2\gamma\mu} - \frac{1}{2}, \beta = \frac{2\alpha^2 - 1}{\alpha - 1}, \gamma \in [\eta, \sqrt{\frac{\eta}{\mu}}], \eta, \gamma \in (0, \frac{1}{L}], \gamma\mu \leq \frac{3}{4}, \tau \geq 2$, *FedAQ yields*

$$\mathbb{E}[\Phi_{k+1}]$$

$$\leq D(\gamma,\tau)\mathbb{E}[\Phi_k] + (\frac{\eta^2 L}{2} + \frac{\gamma^2\mu}{6})\frac{\tau\sigma^2}{M} + \gamma\tau \cdot \max_{0 \leq t < \tau} \mathbb{E}[\|\nabla F(\bar{w}_{k,t}^{md}) - \frac{1}{M}\sum_{m=1}^M \nabla F(w_{k,t}^{md,m})\|^2]$$

$$+ \underbrace{\frac{q}{M}(\frac{\gamma^2\mu}{3} + \eta^2 L)\tau\sigma^2 + \frac{q}{2M}\left(((\gamma - \eta)^2\gamma^2\mu^2(\frac{\mu}{3} + \frac{L}{4}) + \gamma^4(\frac{\mu}{3} + L)^2 L\right)\tau^3\sigma^2}_{\text{additional terms due to quantization}}$$

*Where $D(\gamma, \tau)$ is defined as*

$$D(\gamma,\tau) = (1 - \frac{1}{3}\gamma\mu)^\tau + \underbrace{\frac{q}{M}\left(\gamma^2\mu(\frac{8}{3}\mu + 2L) + 2\gamma^2 L(\frac{\mu}{3} + L)\right)\tau^2}_{\text{additional terms due to quantization}}$$

In order to prove Lemma F.1, we first introduce five crucial Propositions for proving Lemma F.1. Then, we prove Lemma F.1 by using Propositions in the last part of this section.

**Proposition F.2.** *Let Assumption 2.1 hold and consider any $k$ synchronization round. Then, we can decompose the expectation as follows:*

$$\mathbb{E}[\|w_{k+1} - w^*\|^2] = \mathbb{E}[\|w_{k+1} - \bar{w}_{k,\tau}\|^2] + \mathbb{E}[\|\bar{w}_{k,\tau} - w^*\|^2]$$
$$\mathbb{E}[F(w_{k+1}^{ag}) - F^*] = \mathbb{E}[F(w_{k+1}^{ag}) - F(\bar{w}_{k,\tau}^{ag})] + \mathbb{E}[F(\bar{w}_{k,\tau}^{ag}) - F^*]$$

*Proof of Proposition F.2*   The second equality is trivial. The first equality is the same as one in Proposition D.2.

**Proposition F.3.** *Let $F$ be $\mu$-strongly convex, and assume Assumption 2.2, 2.3, 2.4, then for $\alpha = \frac{3}{2\gamma\mu} - \frac{1}{2}, \beta = \frac{2\alpha^2 - 1}{\alpha - 1}, \gamma \in [\eta, \sqrt{\frac{\eta}{\mu}}], \eta \in (0, \frac{1}{L}]$, FedAQ yields*

$$\mathbb{E}[\Phi_{k,\tau}] \leq (1 - \frac{1}{3}\gamma\mu)^\tau \mathbb{E}[\Phi_k] + (\frac{\eta^2 L}{2} + \frac{\gamma^2\mu}{6})\frac{\tau\sigma^2}{M}$$

$$+ \gamma\tau \cdot \max_{0 \leq t < \tau} \mathbb{E}[\|\nabla F(\bar{w}_{k,t}^{md}) - \frac{1}{M}\sum_{m=1}^M \nabla F(w_{k,t}^{md,m})\|^2]$$

*Proof of Proposition F.3*   We refer to the proof of Lemma C.2 in Yuan and Ma (2020). There is no quantization between $\Phi_{k,\tau}$ and $\Phi_k$. Thus, we can directly apply useful inequalities in the proof of

Lemma C.2 in Yuan and Ma (2020) to our proof. Then, we obtain

$$\mathbb{E}[\Phi_{k,t+1}|\mathcal{F}_{k,t}] \leq (1 - \frac{1}{3}\gamma\mu)\Phi_{k,t} + (\frac{\eta^2 L}{2} + \frac{\gamma^2\mu}{6})\frac{\sigma^2}{M} + \gamma\|\nabla F(\bar{w}_{k,t}^{\mathrm{md}}) - \frac{1}{M}\sum_{m=1}^{M}\nabla F(w_{k,t}^{\mathrm{md},m})\|^2$$

From the above relationship between $\Phi_{k,t+1}$ and $\Phi_{k,t}$, we get

$$\mathbb{E}[\Phi_{k,\tau}] \leq (1 - \frac{1}{3}\gamma\mu)^{\tau}\mathbb{E}[\Phi_k] + \Big(\sum_{t=0}^{\tau-1}(1 - \frac{1}{3}\gamma\mu)^t\Big) \cdot (\frac{\eta^2 L}{2} + \frac{\gamma^2\mu}{6})\frac{\sigma^2}{M}$$

$$+ \gamma\sum_{t=0}^{\tau-1}\Big\{(1 - \frac{1}{3}\gamma\mu)^{\tau-t-1}\mathbb{E}[\|\nabla F(\bar{w}_{k,t}^{\mathrm{md}}) - \frac{1}{M}\sum_{m=1}^{M}\nabla F(w_{k,t}^{\mathrm{md},m})\|^2]\Big\}$$

$$\leq (1 - \frac{1}{3}\gamma\mu)^{\tau}\mathbb{E}[\Phi_k] + (\frac{\eta^2 L}{2} + \frac{\gamma^2\mu}{6})\frac{\tau\sigma^2}{M}$$

$$+ \gamma\tau \cdot \max_{0\leq t<\tau}\mathbb{E}[\|\nabla F(\bar{w}_{k,t}^{\mathrm{md}}) - \frac{1}{M}\sum_{m=1}^{M}\nabla F(w_{k,t}^{\mathrm{md},m})\|^2]$$

**Proposition F.4.** *Let Assumption 2.1 hold. Then, we have*

$$\mathbb{E}[\|w_{k+1} - \bar{w}_{k,\tau}\|^2] \leq \frac{q}{M^2}\sum_{m=1}^{M}\mathbb{E}[\|w_{k,\tau}^m - w_k\|^2]$$

$$\mathbb{E}[F(w_{k+1}^{ag}) - F(\bar{w}_{k,\tau}^{ag})] \leq \frac{qL}{2M^2}\sum_{m=1}^{M}\mathbb{E}[\|w_{k,\tau}^{ag,m} - w_k^{ag}\|^2]$$

*Proof of Proposition F.4* The first inequality is the same as one in Proposition D.4. The proof of the second inequality is similar to Proposition D.4 as well.

$$\mathbb{E}[F(w_{k+1}^{\mathrm{ag}}) - F(\bar{w}_{k,\tau}^{\mathrm{ag}})] = \mathbb{E}[F(w_k^{\mathrm{ag}} + \frac{1}{M}\sum_{m=1}^{M}Q(w_{k,\tau}^{\mathrm{ag},m} - w_k^{\mathrm{ag}})) - F(\frac{1}{M}\sum_{m=1}^{M}w_{k,\tau}^{\mathrm{ag},m})]$$

$$\leq \mathbb{E}\Big[\langle\nabla F(\frac{1}{M}\sum_{m=1}^{M}w_{k,\tau}^{\mathrm{ag},m}), \frac{1}{M}\sum_{m=1}^{M}\Big(Q(w_{k,\tau}^{\mathrm{ag},m} - w_k^{\mathrm{ag}})$$

$$- (w_{k,\tau}^{\mathrm{ag},m} - w_k^{\mathrm{ag}})\Big)\rangle + \frac{L}{2}\|\frac{1}{M}\sum_{m=1}^{M}Q(w_{k,\tau}^{\mathrm{ag},m} - w_k^{\mathrm{ag}}) - (w_{k,\tau}^{\mathrm{ag},m} - w_k^{\mathrm{ag}})\|^2\Big]$$

$$= \frac{L}{2}\mathbb{E}[\|\frac{1}{M}\sum_{m=1}^{M}Q(w_{k,\tau}^{\mathrm{ag},m} - w_k^{\mathrm{ag}}) - (w_{k,\tau}^{\mathrm{ag},m} - w_k^{\mathrm{ag}})\|^2]$$

$$= \frac{L}{2M^2}\sum_{m=1}^{M}\mathbb{E}[\|Q(w_{k,\tau}^{\mathrm{ag},m} - w_k^{\mathrm{ag}}) - (w_{k,\tau}^{\mathrm{ag},m} - w_k^{\mathrm{ag}})\|^2]$$

$$\leq \frac{qL}{2M^2}\sum_{m=1}^{M}\mathbb{E}[\|w_{k,\tau}^{\mathrm{ag},m} - w_k^{\mathrm{ag}}\|^2]$$

**Proposition F.5.** *Let F be $\mu$-strongly convex, and assume Assumption 2.2, 2.3, 2.4, then for $\alpha = \frac{3}{2\gamma\mu} - \frac{1}{2}, \beta = \frac{2\alpha^2-1}{\alpha-1}, \gamma \in [\eta, \sqrt{\frac{\eta}{\mu}}], \eta, \gamma \in (0, \frac{1}{L}], \gamma\mu \leq \frac{3}{4}$, we get*

$$\mathbb{E}[B_{k,t}^m] \leq \mathbb{E}[B_{k,0}^m] + \Big(\Big(\frac{\mu}{3}(\frac{2\alpha-1}{2\alpha^2-1})^2 + L(\frac{\alpha-1}{2\alpha^2-1})^2\Big) \cdot (\gamma-\eta)^2 + \gamma^4(\frac{\mu}{3} + L)^2\frac{2\alpha^2-\alpha}{2\alpha^2-1}L\Big)$$

$$\cdot \frac{1 + \frac{1}{2}\alpha^{-1}}{\frac{1}{4}\alpha^{-2}} \cdot \Big(1 - (1 - \frac{1}{2}\alpha^{-1} + \frac{\frac{1}{2}\alpha^{-1}}{1 + \frac{1}{2}\alpha^{-1}})^t\Big)\sigma^2$$

*Proof of Proposition F.5*  From the notation mentioned in the beginning of Section F,

$$\mathbb{E}[B_{k,t+1}^m|\mathcal{F}_{k,t}] = \left(\frac{\mu\alpha^{-2}}{3}(1-\beta^{-1})^2 + L\beta^{-2}\right)\mathbb{E}[\|w_{k,t+1}^m - w_{k,t+1}^{\mathrm{ag},m}\|^2|\mathcal{F}_{k,t}]$$
$$+ \gamma^2(\frac{\mu}{3}+L)\frac{2\alpha^2-\alpha}{2\alpha^2-1}\cdot 2L\mathbb{E}[\Phi_{k,t+1}^m|\mathcal{F}_{k,t}] \tag{22}$$

Thus, let's sequentially compute $\mathbb{E}[\|w_{k,t+1}^m - w_{k,t+1}^{\mathrm{ag},m}\|^2|\mathcal{F}_{k,t}]$ and $\mathbb{E}[\Phi_{k,t+1}^m|\mathcal{F}_{k,t}]$.

$$\mathbb{E}[\|w_{k,t+1}^m - w_{k,t+1}^{\mathrm{ag},m}\|^2|\mathcal{F}_{k,t}] = \mathbb{E}[\|(1-\alpha^{-1})w_{k,t}^m + \alpha^{-1}w_{k,t}^{\mathrm{md},m} - \gamma g_{k,t}^m - w_{k,t}^{\mathrm{md},m} + \eta g_{k,t}^m\|^2|\mathcal{F}_{k,t}]$$
$$= \mathbb{E}[\|(1-\alpha^{-1})(w_{k,t}^m - w_{k,t}^{\mathrm{md},m}) - (\gamma-\eta)g_{k,t}^m\|^2|\mathcal{F}_{k,t}]\ (\leftarrow \gamma \geq \eta)$$
$$= \|(1-\alpha^{-1})(w_{k,t}^m - w_{k,t}^{\mathrm{md},m}) - (\gamma-\eta)\nabla F(w_{k,t}^{\mathrm{md},m})\|^2$$
$$+ (\gamma-\eta)^2\mathbb{E}[\|\nabla F(w_{k,t}^{\mathrm{md},m}) - g_{k,t}^m\|^2|\mathcal{F}_{k,t}]$$
$$\leq (1-\alpha^{-1})^2\|w_{k,t}^m - w_{k,t}^{\mathrm{md},m}\|^2 + (\gamma-\eta)^2\|\nabla F(w_{k,t}^{\mathrm{md},m})\|^2$$
$$+ (\gamma-\eta)^2\sigma^2 - 2(\gamma-\eta)\langle(1-\alpha^{-1})(w_{k,t}^m - w_{k,t}^{\mathrm{md},m}), \nabla F(w_{k,t}^{\mathrm{md},m})\rangle$$
$$\leq (1-\alpha^{-1})^2(1+2\alpha^{-1})\|w_{k,t}^m - w_{k,t}^{\mathrm{md},m}\|^2$$
$$+ (\gamma-\eta)^2(1+\frac{\alpha}{2})\|\nabla F(w_{k,t}^{\mathrm{md},m})\|^2 + (\gamma-\eta)^2\sigma^2$$

Here, we need to bound $\|\nabla F(w_{k,t}^{\mathrm{md},m})\|^2$.

$$\|\nabla F(w_{k,t}^{\mathrm{md},m})\|^2 \leq 2L(F(w_{k,t}^{\mathrm{md},m}) - F^*)\ (\because \text{ Assumption 2.3})$$
$$\leq 2L\left(\beta^{-1}(F(w_{k,t}^m) - F(w^*)) + (1-\beta^{-1})(F(w_{k,t}^{\mathrm{ag},m}) - F^*)\right)$$
$$\leq \beta^{-1}L^2\|w_{k,t}^m - w^*\|^2 + 2(1-\beta^{-1})L(F(w_{k,t}^{\mathrm{ag},m}) - F^*)$$
$$= \frac{\alpha-1}{2\alpha^2-1}L^2\|w_{k,t}^m - w^*\|^2 + 2L\cdot\frac{2\alpha^2-\alpha}{2\alpha^2-1}(F(w_{k,t}^{\mathrm{ag},m}) - F^*)$$
$$\leq \frac{\frac{\mu}{3}(2\alpha^2-\alpha)}{2\alpha^2-1}L\|w_{k,t}^m - w^*\|^2 + 2L\cdot\frac{2\alpha^2-\alpha}{2\alpha^2-1}(F(w_{k,t}^{\mathrm{ag},m}) - F^*)$$
$$= \frac{2\alpha^2-\alpha}{2\alpha^2-1}\cdot 2L\Phi_{k,t}^m \tag{23}$$

It is easy to show $(\alpha-1)L \leq \frac{\mu}{3}(2\alpha^2-\alpha)$ by using the fact $\gamma L \leq 1$. Therefore, we finally get

$$\mathbb{E}[\|w_{k,t+1}^m - w_{k,t+1}^{\mathrm{ag},m}\|^2|\mathcal{F}_{k,t}]$$
$$\leq (1-\alpha^{-1})^2(1+2\alpha^{-1})\|w_{k,t}^m - w_{k,t}^{\mathrm{md},m}\|^2 + (\gamma-\eta)^2(1+\frac{\alpha}{2})\|\nabla F(w_{k,t}^{\mathrm{md},m})\|^2 + (\gamma-\eta)^2\sigma^2$$
$$\leq (1-\alpha^{-1})^2(1+2\alpha^{-1})\|w_{k,t}^m - w_{k,t}^{\mathrm{md},m}\|^2 + (\gamma-\eta)^2(1+\frac{\alpha}{2})(\frac{2\alpha^2-\alpha}{2\alpha^2-1}\cdot 2L\Phi_{k,t}^m)$$
$$+ (\gamma-\eta)^2\sigma^2 \tag{24}$$

Now, let's compute $\mathbb{E}[\Phi_{k,t+1}^m|\mathcal{F}_{k,t}]$. We need to compute $\mathbb{E}[\|w_{k,t+1}^m - w^*\|^2|\mathcal{F}_{k,t}]$ and $\mathbb{E}[F(w_{k,t+1}^{\mathrm{ag},m}) - F^*|\mathcal{F}_{k,t}]$ first.

$$\mathbb{E}[\|w_{k,t+1}^m - w^*\|^2|\mathcal{F}_{k,t}]$$

$$= \mathbb{E}[\|(1-\alpha^{-1})w_{k,t}^m + \alpha^{-1}w_{k,t}^{\mathrm{md},m} - \gamma g_{k,t}^m - w^*\|^2|\mathcal{F}_{k,t}]$$

$$\leq \|(1-\alpha^{-1})w_{k,t}^m + \alpha^{-1}w_{k,t}^{\mathrm{md},m} - \gamma\nabla F(w_{k,t}^{\mathrm{md},m}) - w^*\|^2 + \gamma^2\sigma^2$$

$$\leq (1+\tfrac{1}{2}\alpha^{-1})\|(1-\alpha^{-1})w_{k,t}^m + \alpha^{-1}w_{k,t}^{\mathrm{md},m} - \gamma\nabla F(w_{k,t}^{\mathrm{md},m}) - w^*\|^2 + \gamma^2\sigma^2$$

$$= (1+\tfrac{1}{2}\alpha^{-1})\|(1-\alpha^{-1})w_{k,t}^m + \alpha^{-1}w_{k,t}^{\mathrm{md},m} - w^*\|^2 + \gamma^2(1+\tfrac{1}{2}\alpha^{-1})\|\nabla F(w_{k,t}^{\mathrm{md},m})\|^2$$

$$- 2\gamma(1+\tfrac{1}{2}\alpha^{-1})\langle(1-\alpha^{-1})w_{k,t}^m + \alpha^{-1}w_{k,t}^{\mathrm{md},m} - w^*, \nabla F(w_{k,t}^{\mathrm{md},m})\rangle + \gamma^2\sigma^2$$

$$\leq (1+\tfrac{1}{2}\alpha^{-1})\left((1-\alpha^{-1})\|w_{k,t}^m - w^*\|^2 + \alpha^{-1}\|w_{k,t}^{\mathrm{md},m} - w^*\|^2\right) + \gamma^2(1+\tfrac{1}{2}\alpha^{-1})$$

$$\cdot \|\nabla F(w_{k,t}^{\mathrm{md},m})\|^2 - 2\gamma(1+\tfrac{1}{2}\alpha^{-1})\langle(1-\alpha^{-1})w_{k,t}^m + \alpha^{-1}w_{k,t}^{\mathrm{md},m} - w^*, \nabla F(w_{k,t}^{\mathrm{md},m})\rangle + \gamma^2\sigma^2$$

It is easy to show $(1+\tfrac{1}{2}\alpha^{-1})(1-\alpha^{-1}) < 1 - \tfrac{1}{2}\alpha^{-1}, 1 + \tfrac{1}{2}\alpha^{-1} \leq \tfrac{3}{2}$. Due to these facts, we obtain

$$\mathbb{E}[\|w_{k,t+1}^m - w^*\|^2|\mathcal{F}_{k,t}]$$

$$\leq (1-\tfrac{1}{2}\alpha^{-1})\|w_{k,t}^m - w^*\|^2 + \tfrac{3}{2}\alpha^{-1}\|w_{k,t}^{\mathrm{md},m} - w^*\|^2 + \tfrac{3}{2}\gamma^2\|\nabla F(w_{k,t}^{\mathrm{md},m})\|^2$$

$$- 2\gamma(1+\tfrac{1}{2}\alpha^{-1})\langle(1-\alpha^{-1})w_{k,t}^m + \alpha^{-1}w_{k,t}^{\mathrm{md},m} - w^*, \nabla F(w_{k,t}^{\mathrm{md},m})\rangle + \gamma^2\sigma^2$$

$$\leq (1-\tfrac{1}{2}\alpha^{-1})\|w_{k,t}^m - w^*\|^2 + \tfrac{3}{2}\alpha^{-1}\|w_{k,t}^{\mathrm{md},m} - w^*\|^2 + \tfrac{3}{2}\gamma^2\|\nabla F(w_{k,t}^{\mathrm{md},m})\|^2$$

$$- 2\gamma(1+\tfrac{1}{2}\alpha^{-1})\langle(1-\alpha^{-1}(1-\beta^{-1}))w_{k,t}^m + \alpha^{-1}(1-\beta^{-1})w_{k,t}^{\mathrm{ag},m} - w^*, \nabla F(w_{k,t}^{\mathrm{md},m})\rangle + \gamma^2\sigma^2$$

Next, we compute the upper bound of $\mathbb{E}[F(w_{k,t+1}^{\mathrm{ag},m}) - F^*|\mathcal{F}_{k,t}]$.

$$\mathbb{E}[F(w_{k,t+1}^{\mathrm{ag},m}) - F^*|\mathcal{F}_{k,t}]$$

$$\leq \mathbb{E}[F(w_{k,t}^{\mathrm{md},m}) + \langle\nabla F(w_{k,t}^{\mathrm{md},m}), w_{k,t+1}^{\mathrm{ag},m} - w_{k,t}^{\mathrm{md},m}\rangle + \tfrac{L}{2}\|w_{k,t+1}^{\mathrm{ag},m} - w_{k,t}^{\mathrm{md},m}\|^2 - F^*|\mathcal{F}_{k,t}]$$

$$\leq F(w_{k,t}^{\mathrm{md},m}) - F^* - \eta\|\nabla F(w_{k,t}^{\mathrm{md},m})\|^2 + \tfrac{\eta^2 L}{2}\|\nabla F(w_{k,t}^{\mathrm{md},m})\|^2 + \tfrac{\eta^2 L}{2}\sigma^2$$

$$\leq F(w_{k,t}^{\mathrm{md},m}) - F^* - \tfrac{\eta}{2}\|\nabla F(w_{k,t}^{\mathrm{md},m})\|^2 + \tfrac{\eta^2 L}{2}\sigma^2 \ (\because 1 - \tfrac{\eta L}{2} \geq \tfrac{1}{2} \leftarrow \eta \in [0, \tfrac{1}{L}])$$

$$= (1-\tfrac{1}{2}\alpha^{-1})(F(w_{k,t}^{\mathrm{ag},m}) - F^*) + \tfrac{1}{2}\alpha^{-1}(F(w_{k,t}^{\mathrm{md},m}) - F^*)$$

$$+ (1-\tfrac{1}{2}\alpha^{-1})(F(w_{k,t}^{\mathrm{md},m}) - F(w_{k,t}^{\mathrm{ag},m})) - \tfrac{\eta}{2}\|\nabla F(w_{k,t}^{\mathrm{md},m})\|^2 + \tfrac{\eta^2 L}{2}\sigma^2$$

$$\leq (1-\tfrac{1}{2}\alpha^{-1})(F(w_{k,t}^{\mathrm{ag},m}) - F^*) - \tfrac{\mu\alpha^{-1}}{4}\|w_{k,t}^{\mathrm{md},m} - w^*\|^2 + \tfrac{1}{2}\alpha^{-1}\langle\nabla F(w_{k,t}^{\mathrm{md},m}), w_{k,t}^{\mathrm{md},m} - w^*\rangle$$

$$+ (1-\tfrac{1}{2}\alpha^{-1})\langle\nabla F(w_{k,t}^{\mathrm{md},m}), w_{k,t}^{\mathrm{md},m} - w_{k,t}^{\mathrm{ag},m}\rangle - \tfrac{\eta}{2}\|\nabla F(w_{k,t}^{\mathrm{md},m})\|^2 + \tfrac{\eta^2 L}{2}\sigma^2$$

$$= (1-\tfrac{1}{2}\alpha^{-1})(F(w_{k,t}^{\mathrm{ag},m}) - F^*) - \tfrac{\mu\alpha^{-1}}{4}\|w_{k,t}^{\mathrm{md},m} - w^*\|^2 - \tfrac{\eta}{2}\|\nabla F(w_{k,t}^{\mathrm{md},m})\|^2 + \tfrac{\eta^2 L}{2}\sigma^2$$

$$+ \tfrac{1}{2}\alpha^{-1}\langle\nabla F(w_{k,t}^{\mathrm{md},m}), 2\alpha\beta^{-1}w_{k,t}^m + (1-2\alpha\beta^{-1})w_{k,t}^{\mathrm{ag},m} - w^*\rangle$$

It is easy to show $\frac{1}{2}\alpha^{-1} = \frac{\gamma\mu}{3}(1 + \frac{1}{2}\alpha^{-1})$. Then, we bound $\mathbb{E}[\Phi^m_{k,t+1}|\mathcal{F}_{k,t}]$ by using the above results.

$$
\begin{aligned}
\mathbb{E}[\Phi^m_{k,t+1}|\mathcal{F}_{k,t}] &= \frac{\mu}{6}\mathbb{E}[\|w^m_{k,t+1} - w^*\|^2|\mathcal{F}_{k,t}] + \mathbb{E}[F(w^{\mathrm{ag},m}_{k,t+1}) - F^*|\mathcal{F}_{k,t}] \\
&\leq (1 - \frac{1}{2}\alpha^{-1})\Phi^m_{k,t} - \frac{2\eta - \gamma^2\mu}{4}\|\nabla F(w^{\mathrm{md},m}_{k,t})\|^2 + \frac{1}{2}(\frac{\gamma^2\mu}{3} + \eta^2 L)\sigma^2 \\
&\leq (1 - \frac{1}{2}\alpha^{-1})\Phi^m_{k,t} + \frac{1}{2}(\frac{\gamma^2\mu}{3} + \eta^2 L)\sigma^2 \quad (\because \gamma \leq \sqrt{\frac{\eta}{\mu}}) \\
&\leq (1 - \frac{1}{2}\alpha^{-1})\Phi^m_{k,t} + \frac{\gamma^2}{2}(\frac{\mu}{3} + L)\sigma^2
\end{aligned}
\tag{25}
$$

Plugging (24), (25) in (22) yields,

$$
\begin{aligned}
&\mathbb{E}[B^m_{k,t+1}|\mathcal{F}_{k,t}] \\
&\leq \Big(\frac{\mu\alpha^{-2}}{3}(1 - \beta^{-1})^2 + L\beta^{-2}\Big)\Big((1 - \alpha^{-1})^2(1 + 2\alpha^{-1})\|w^m_{k,t} - w^{\mathrm{md},m}_{k,t}\|^2 \\
&+ (\gamma - \eta)^2(1 + \frac{\alpha}{2}) \cdot (\frac{2\alpha^2 - \alpha}{2\alpha^2 - 1} \cdot 2L\Phi^m_{k,t}) + (\gamma - \eta)^2\sigma^2\Big) \\
&+ \gamma^2(\frac{\mu}{3} + L)\frac{2\alpha^2 - \alpha}{2\alpha^2 - 1} \cdot 2L\Big((1 - \frac{1}{2}\alpha^{-1})\Phi^m_{k,t} + \frac{\gamma^2}{2}(\frac{\mu}{3} + L)\sigma^2\Big) \\
&= (1 - \alpha^{-1})^2(1 + 2\alpha^{-1})\Big(\frac{\mu\alpha^{-2}}{3}(1 - \beta^{-1})^2 + L\beta^{-2}\Big)\|w^m_{k,t} - w^{\mathrm{md},m}_{k,t}\|^2 \\
&+ \Big(\Big(\frac{\mu\alpha^{-2}}{3}(1 - \beta^{-1})^2 + L\beta^{-2}\Big)(\gamma - \eta)^2(1 + \frac{\alpha}{2}) + (1 - \frac{1}{2}\alpha^{-1})\gamma^2(\frac{\mu}{3} + L)\Big) \\
&\cdot (\frac{2\alpha^2 - \alpha}{2\alpha^2 - 1} \cdot 2L\Phi^m_{k,t}) + \Big(\Big(\frac{\mu\alpha^{-2}}{3}(1 - \beta^{-1})^2 + L\beta^{-2}\Big)(\gamma - \eta)^2 + \gamma^4(\frac{\mu}{3} + L)^2\frac{2\alpha^2 - \alpha}{2\alpha^2 - 1}L\Big)\sigma^2
\end{aligned}
\tag{26}
$$

We can show that both coefficients of $\|w^m_{k,t} - w^{\mathrm{md},m}_{k,t}\|^2$ and $\frac{2\alpha^2 - \alpha}{2\alpha^2 - 1} \cdot 2L\Phi^m_{k,t}$ are upper bounded by $1 - \frac{1}{2}\alpha^{-1} + \frac{\frac{1}{2}\alpha^{-1}}{1 + \frac{1}{2}\alpha^{-1}}$.

$$
(1 - \alpha^{-1})^2(1 + 2\alpha^{-1}) \leq 1 - \frac{1}{2}\alpha^{-1} + \frac{\frac{1}{2}\alpha^{-1}}{1 + \frac{1}{2}\alpha^{-1}}(< 1)
\tag{27}
$$

$$
\Leftrightarrow 1 - \frac{1}{4}\alpha^{-2} + \frac{1}{2}\alpha^{-1} - (1 - \alpha^{-1})^2(1 + 2\alpha^{-1})(1 + \frac{1}{2}\alpha^{-1}) \geq 0
$$

Let's define $g_1(\alpha^{-1}) = 1 - \frac{1}{4}\alpha^{-2} + \frac{1}{2}\alpha^{-1} - (1 - \alpha^{-1})^2(1 + 2\alpha^{-1})(1 + \frac{1}{2}\alpha^{-1})$. Then, it is easy to check that $g_1(\alpha^{-1}) \geq 0$ for $0 < \alpha^{-1} \leq 1$. Moreover, we would like to show the below inequality.

$$
\begin{aligned}
&\Big(\frac{\mu\alpha^{-2}}{3}(1 - \beta^{-1})^2 + L\beta^{-2}\Big)(\gamma - \eta)^2(1 + \frac{\alpha}{2}) + (1 - \frac{1}{2}\alpha^{-1})\gamma^2(\frac{\mu}{3} + L) \\
&\leq \Big(\frac{\mu\alpha^{-2}}{3}(1 - \beta^{-1})^2 + L\beta^{-2}\Big)\gamma^2(1 + \frac{\alpha}{2}) + (1 - \frac{1}{2}\alpha^{-1})\gamma^2(\frac{\mu}{3} + L) \\
&\leq (1 - \frac{1}{2}\alpha^{-1} + \frac{\frac{1}{2}\alpha^{-1}}{1 + \frac{1}{2}\alpha^{-1}})\gamma^2(\frac{\mu}{3} + L)
\end{aligned}
\tag{28}
$$

Since $\frac{\mu\alpha^{-2}}{3}(1 - \beta^{-1})^2 + L\beta^{-2} = \frac{\mu}{3}(\frac{2\alpha - 1}{2\alpha^2 - 1})^2 + L(\frac{\alpha - 1}{2\alpha^2 - 1})^2 \leq (\frac{\mu}{3} + \frac{L}{4})(\frac{2\alpha - 1}{2\alpha^2 - 1})^2$, it is enough to show

$$
(\frac{\mu}{3} + \frac{L}{4})(\frac{2\alpha - 1}{2\alpha^2 - 1})^2\gamma^2(1 + \frac{\alpha}{2}) \leq \frac{\frac{1}{2}\alpha^{-1}}{1 + \frac{1}{2}\alpha^{-1}}\gamma^2(\frac{\mu}{3} + L)
$$

We also know that $\frac{\frac{\mu}{3}+L}{\frac{\mu}{3}+\frac{L}{4}} = 4 - \frac{1}{\frac{1}{3}+\frac{L}{\mu}\cdot\frac{1}{4}} > \frac{16}{7}$ ($\because \frac{L}{\mu} > 1$). Then, we only need to show

$$(\frac{2\alpha-1}{2\alpha^2-1})^2(1+\frac{\alpha}{2}) \le \frac{16}{7}\cdot\frac{\frac{1}{2}}{\alpha+\frac{1}{2}}$$

$$\Leftrightarrow \frac{8}{7}(2\alpha^2-1)^2 - (2\alpha-1)^2(1+\frac{\alpha}{2})(\alpha+\frac{1}{2}) \ge 0$$

Let's define $g_2(\alpha) = \frac{8}{7}(2\alpha^2-1)^2 - (2\alpha-1)^2(1+\frac{\alpha}{2})(\alpha+\frac{1}{2})$. Then, it is easy to check $g_2(\alpha) \ge 0$ for $\alpha \ge \frac{3}{2}$. As we assume $\gamma\mu \le \frac{3}{4}$, we can say $\alpha = \frac{3}{2\gamma\mu} - \frac{1}{2} \ge \frac{3}{2}$. This indicates that the inequality (28) is satisfied. Thus, from (26), (27), and (28) we finally get

$$\mathbb{E}[B^m_{k,t+1}|\mathcal{F}_{k,t}] \le (1 - \frac{1}{2}\alpha^{-1} + \frac{\frac{1}{2}\alpha^{-1}}{1+\frac{1}{2}\alpha^{-1}})B^m_{k,t}$$
$$+ \left(\left(\frac{\mu\alpha^{-2}}{3}(1-\beta^{-1})^2 + L\beta^{-2}\right)(\gamma-\eta)^2 + \gamma^4(\frac{\mu}{3}+L)^2\frac{2\alpha^2-\alpha}{2\alpha^2-1}L\right)\sigma^2$$

From this relationship between $B^m_{k,t+1}$ and $B^m_{k,t}$, we obtain the result of Proposition F.5.

$$\mathbb{E}[B^m_{k,t}] \le (1-\frac{1}{2}\alpha^{-1} + \frac{\frac{1}{2}\alpha^{-1}}{1+\frac{1}{2}\alpha^{-1}})^t\mathbb{E}[B^m_{k,0}] + \left(\left(\frac{\mu\alpha^{-2}}{3}(1-\beta^{-1})^2 + L\beta^{-2}\right)(\gamma-\eta)^2\right.$$

$$+ \gamma^4(\frac{\mu}{3}+L)^2\frac{2\alpha^2-\alpha}{2\alpha^2-1}L\Big)\sigma^2 \cdot \frac{1-(1-\frac{1}{2}\alpha^{-1}+\frac{\frac{1}{2}\alpha^{-1}}{1+\frac{1}{2}\alpha^{-1}})^t}{1-(1-\frac{1}{2}\alpha^{-1}+\frac{\frac{1}{2}\alpha^{-1}}{1+\frac{1}{2}\alpha^{-1}})}$$

$$\le \mathbb{E}[B^m_{k,0}] + \left(\left(\frac{\mu}{3}(\frac{2\alpha-1}{2\alpha^2-1})^2 + L(\frac{\alpha-1}{2\alpha^2-1})^2\right)\cdot(\gamma-\eta)^2 + \gamma^4(\frac{\mu}{3}+L)^2\frac{2\alpha^2-\alpha}{2\alpha^2-1}L\right)$$

$$\cdot\frac{1+\frac{1}{2}\alpha^{-1}}{\frac{1}{4}\alpha^{-2}}\cdot\left(1-(1-\frac{1}{2}\alpha^{-1}+\frac{\frac{1}{2}\alpha^{-1}}{1+\frac{1}{2}\alpha^{-1}})^t\right)\sigma^2$$

**Proposition F.6.** *Let F be $\mu$-strongly convex, and assume Assumption 2.2, 2.3, 2.4, then for $\alpha = \frac{3}{2\gamma\mu} - \frac{1}{2}, \beta = \frac{2\alpha^2-1}{\alpha-1}, \gamma \in [\eta, \sqrt{\frac{\eta}{\mu}}], \eta, \gamma \in (0, \frac{1}{L}], \gamma\mu \le \frac{3}{4}, \tau \ge 2$, FedAQ yields*

$$\frac{\mu}{6}\mathbb{E}[\|w^m_{k,\tau}-w_k\|^2] + \frac{L}{2}\mathbb{E}[\|w^{ag,m}_{k,\tau}-w^{ag}_k\|^2] \le \left(\gamma^2\mu(\frac{8}{3}\mu+2L) + 2\gamma^2L(\frac{\mu}{3}+L)\right)\tau^2\mathbb{E}[\Phi_k]$$
$$+ (\frac{\gamma^2\mu}{3}+\eta^2L)\tau\sigma^2$$
$$+ \left((\gamma-\eta)^2\gamma^2\mu^2(\frac{\mu}{3}+\frac{L}{4}) + \gamma^4(\frac{\mu}{3}+L)^2L\right)\frac{\tau^3\sigma^2}{2}$$

*Proof of Proposition F.6* We use the same upper bounds for $\mathbb{E}[\|w^m_{k,\tau}-w_k\|^2]$ and $\mathbb{E}[\|w^{ag,m}_{k,\tau}-w^{ag}_k\|^2]$ as in Proposition D.6.

$$\mathbb{E}[\|w^m_{k,\tau}-w_k\|^2] \le \tau\left(\sum_{t=0}^{\tau-1}2\alpha^{-2}(1-\beta^{-1})^2\mathbb{E}[\|w^m_{k,t}-w^{ag,m}_{k,t}\|^2] + 2\gamma^2\mathbb{E}[\|\nabla F(w^{md,m}_{k,t})\|^2]\right)$$
$$+ 2\tau\gamma^2\sigma^2$$

$$\mathbb{E}[\|w^{ag,m}_{k,\tau}-w^{ag}_k\|^2] \le \tau\left(\sum_{t=0}^{\tau-1}2\beta^{-2}\mathbb{E}[\|w^m_{k,t}-w^{ag,m}_{k,t}\|^2] + 2\eta^2\mathbb{E}[\|\nabla F(w^{md,m}_{k,t})\|^2]\right) + 2\tau\eta^2\sigma^2$$

Thus, by using the above results, we get

$$\frac{\mu}{6}\mathbb{E}[\|w_{k,\tau}^m - w_k\|^2] + \frac{L}{2}\mathbb{E}[\|w_{k,\tau}^{\text{ag},m} - w_k^{\text{ag}}\|^2]$$

$$\leq \tau\sum_{t=0}^{\tau-1}\left\{\left(\frac{\mu\alpha^{-2}}{3}(1-\beta^{-1})^2 + L\beta^{-2}\right)\mathbb{E}[\|w_{k,t}^m - w_{k,t}^{\text{ag},m}\|^2] + (\frac{\gamma^2\mu}{3} + \eta^2 L)\mathbb{E}[\|\nabla F(w_{k,t}^{\text{md},m})\|^2]\right\}$$

$$+ (\frac{\gamma^2\mu}{3} + \eta^2 L)\tau\sigma^2$$

$$\leq \tau\sum_{t=0}^{\tau-1}\left\{\left(\frac{\mu\alpha^{-2}}{3}(1-\beta^{-1})^2 + L\beta^{-2}\right)\mathbb{E}[\|w_{k,t}^m - w_{k,t}^{\text{ag},m}\|^2] + \gamma^2(\frac{\mu}{3} + L)\frac{2\alpha^2 - \alpha}{2\alpha^2 - 1}2L\mathbb{E}[\Phi_{k,t}^m]\right\}$$

$$+ (\frac{\gamma^2\mu}{3} + \eta^2 L)\tau\sigma^2 \ (\because (23))$$

$$= \tau\left(\sum_{t=0}^{\tau-1}\mathbb{E}[B_{k,t}^m]\right) + (\frac{\gamma^2\mu}{3} + \eta^2 L)\tau\sigma^2$$

By Proposition F.5 and the fact $\Phi_{k,0}^m = \Phi_k$, we obtain

$$\frac{\mu}{6}\mathbb{E}[\|w_{k,\tau}^m - w_k\|^2] + \frac{L}{2}\mathbb{E}[\|w_{k,\tau}^{\text{ag},m} - w_k^{\text{ag}}\|^2]$$

$$\leq \tau\left\{\sum_{t=0}^{\tau-1}\mathbb{E}[B_{k,0}^m] + \left(\left(\frac{\mu}{3}(\frac{2\alpha - 1}{2\alpha^2 - 1})^2 + L(\frac{\alpha - 1}{2\alpha^2 - 1})^2\right)(\gamma - \eta)^2 + \gamma^4(\frac{\mu}{3} + L)^2\frac{2\alpha^2 - \alpha}{2\alpha^2 - 1}L\right)\right.$$

$$\frac{1 + \frac{1}{2}\alpha^{-1}}{\frac{1}{4}\alpha^{-2}}\left(1 - (1 - \frac{1}{2}\alpha^{-1} + \frac{\frac{1}{2}\alpha^{-1}}{1 + \frac{1}{2}\alpha^{-1}})^t\right)\sigma^2\right\} + (\frac{\gamma^2\mu}{3} + \eta^2 L)\tau\sigma^2$$

$$= \tau^2\left(\left(\frac{\mu\alpha^{-2}}{3}(1-\beta^{-1})^2 + L\beta^{-2}\right)\mathbb{E}[\|w_k - w_k^{\text{ag}}\|^2] + \gamma^2(\frac{\mu}{3} + L)\frac{2\alpha^2 - \alpha}{2\alpha^2 - 1} \cdot 2L\mathbb{E}[\Phi_k]\right)$$

$$+ \tau\left(\left(\frac{\mu}{3}(\frac{2\alpha - 1}{2\alpha^2 - 1})^2 + L(\frac{\alpha - 1}{2\alpha^2 - 1})^2\right) \cdot (\gamma - \eta)^2 + \gamma^4(\frac{\mu}{3} + L)^2\frac{2\alpha^2 - \alpha}{2\alpha^2 - 1}L\right)\frac{1 + \frac{1}{2}\alpha^{-1}}{\frac{1}{4}\alpha^{-2}}$$

$$\cdot\left(\sum_{t=0}^{\tau-1}1 - (1 - \frac{1}{2}\alpha^{-1} + \frac{\frac{1}{2}\alpha^{-1}}{1 + \frac{1}{2}\alpha^{-1}})^t\right)\sigma^2 + (\frac{\gamma^2\mu}{3} + \eta^2 L)\tau\sigma^2$$

Before we get to the final result, let's find the upper bound for $\|w_k - w_k^{\text{ag}}\|^2$, $\sum_{t=0}^{\tau-1}\left(1 - (1 - \frac{1}{2}\alpha^{-1} + \frac{\frac{1}{2}\alpha^{-1}}{1 + \frac{1}{2}\alpha^{-1}})^t\right)$

$$\|w_k - w_k^{\text{ag}}\|^2 = \|w_k - w^* - (w_k^{\text{ag}} - w^*)\|^2$$

$$\leq (1 + \frac{1}{3})\|w_k - w^*\|^2 + (1 + 3)\|w_k^{\text{ag}} - w^*\|^2$$

$$\leq \frac{4}{3}\|w_k - w^*\|^2 + 4 \cdot \frac{2}{\mu}\left(F(w_k^{\text{ag}}) - F^* - \langle\nabla F(w^*), w_k^{\text{ag}} - w^*\rangle\right)$$

$$= \frac{4}{3}\|w_k - w^*\|^2 + \frac{8}{\mu}(F(w_k^{\text{ag}}) - F^*) = \frac{8}{\mu}\Phi_k$$

$$\sum_{t=0}^{\tau-1}\left(1-(1-\frac{1}{2}\alpha^{-1}+\frac{\frac{1}{2}\alpha^{-1}}{1+\frac{1}{2}\alpha^{-1}})^t\right) = \tau - \sum_{t=0}^{\tau-1}(1-\frac{1}{2}\alpha^{-1}+\frac{\frac{1}{2}\alpha^{-1}}{1+\frac{1}{2}\alpha^{-1}})^t$$

$$= \tau - \frac{1-(1-\frac{1}{2}\alpha^{-1}+\frac{\frac{1}{2}\alpha^{-1}}{1+\frac{1}{2}\alpha^{-1}})^\tau}{1-(1-\frac{1}{2}\alpha^{-1}+\frac{\frac{1}{2}\alpha^{-1}}{1+\frac{1}{2}\alpha^{-1}})}$$

$$\leq \tau - \frac{1-(1-\frac{\frac{1}{4}\alpha^{-2}}{1+\frac{1}{2}\alpha^{-1}}\tau+(\frac{\frac{1}{4}\alpha^{-2}}{1+\frac{1}{2}\alpha^{-1}})^2\frac{\tau(\tau-1)}{2})}{\frac{\frac{1}{4}\alpha^{-2}}{1+\frac{1}{2}\alpha^{-1}}}$$

$$= \frac{\frac{1}{4}\alpha^{-2}}{1+\frac{1}{2}\alpha^{-1}}\cdot\frac{\tau(\tau-1)}{2} \leq \frac{\frac{1}{4}\alpha^{-2}}{1+\frac{1}{2}\alpha^{-1}}\cdot\frac{\tau^2}{2}$$

Therefore, we obtain

$$\frac{\mu}{6}\mathbb{E}[\|w_{k,\tau}^m - w_k\|^2] + \frac{L}{2}\mathbb{E}[\|w_{k,\tau}^{\text{ag},m} - w_k^{\text{ag}}\|^2]$$

$$\leq \left(\frac{8}{3}\alpha^{-2}(1-\beta^{-1})^2 + \frac{8L}{\mu}\beta^{-2} + \gamma^2(\frac{\mu}{3}+L)\frac{2\alpha^2-\alpha}{2\alpha^2-1}\cdot 2L\right)\tau^2\mathbb{E}[\Phi_k] + (\frac{\gamma^2\mu}{3}+\eta^2L)\tau\sigma^2$$

$$+ \left(\left(\frac{\mu}{3}(\frac{2\alpha-1}{2\alpha^2-1})^2 + L(\frac{\alpha-1}{2\alpha^2-1})^2\right)\cdot(\gamma-\eta)^2 + \gamma^4(\frac{\mu}{3}+L)^2\frac{2\alpha^2-\alpha}{2\alpha^2-1}L\right)\cdot\frac{\tau^3\sigma^2}{2} \quad (29)$$

Moreover, we can simplify the above inequality by replacing $\alpha, \beta$ with $\gamma, \mu$. It is easy to show $\frac{2\alpha^2-\alpha}{2\alpha^2-1} \leq 1, \frac{2\alpha-1}{2\alpha^2-1} \leq \frac{1}{\alpha} = \frac{2\gamma\mu}{3-\gamma\mu} \leq \gamma\mu$. Then, we can further show

$$\frac{8}{3}\alpha^{-2}(1-\beta^{-1})^2 + \frac{8L}{\mu}\beta^{-2} + \gamma^2(\frac{\mu}{3}+L)\frac{2\alpha^2-\alpha}{2\alpha^2-1}\cdot 2L$$

$$= \frac{8}{3}(\frac{2\alpha-1}{2\alpha^2-1})^2 + \frac{8L}{\mu}(\frac{\alpha-1}{2\alpha^2-1})^2 + \gamma^2(\frac{\mu}{3}+L)\frac{2\alpha^2-\alpha}{2\alpha^2-1}\cdot 2L$$

$$\leq (\frac{8}{3}+\frac{2L}{\mu})(\frac{2\alpha-1}{2\alpha^2-1})^2 + \gamma^2(\frac{\mu}{3}+L)2L$$

$$\leq (\frac{8}{3}+\frac{2L}{\mu})\alpha^{-2} + \gamma^2(\frac{\mu}{3}+L)2L$$

$$\leq \gamma^2\mu(\frac{8}{3}\mu+2L) + 2\gamma^2L(\frac{\mu}{3}+L) \quad (30)$$

We also get

$$\left(\frac{\mu}{3}(\frac{2\alpha-1}{2\alpha^2-1})^2 + L(\frac{\alpha-1}{2\alpha^2-1})^2\right)\cdot(\gamma-\eta)^2 + \gamma^4(\frac{\mu}{3}+L)^2\frac{2\alpha^2-\alpha}{2\alpha^2-1}L$$

$$\leq (\frac{\mu}{3}+\frac{L}{4})(\frac{2\alpha-1}{2\alpha^2-1})^2(\gamma-\eta)^2 + \gamma^4(\frac{\mu}{3}+L)^2L$$

$$\leq (\gamma-\eta)^2\gamma^2\mu^2(\frac{\mu}{3}+\frac{L}{4}) + \gamma^4(\frac{\mu}{3}+L)^2L \quad (31)$$

Finally, from (29), (30), and (31), we conclude as below

$$\frac{\mu}{6}\mathbb{E}[\|w_{k,\tau}^m - w_k\|^2] + \frac{L}{2}\mathbb{E}[\|w_{k,\tau}^{\text{ag},m} - w_k^{\text{ag}}\|^2] \leq \left(\gamma^2\mu(\frac{8}{3}\mu+2L) + 2\gamma^2L(\frac{\mu}{3}+L)\right)\tau^2\mathbb{E}[\Phi_k]$$

$$+ (\frac{\gamma^2\mu}{3}+\eta^2L)\tau\sigma^2$$

$$+ \left((\gamma-\eta)^2\gamma^2\mu^2(\frac{\mu}{3}+\frac{L}{4}) + \gamma^4(\frac{\mu}{3}+L)^2L\right)\frac{\tau^3\sigma^2}{2}$$

*Proof of Lemma F.1* By the definition of $\Phi_k$, $\Phi_{k,t}$ and Proposition F.2,

$$\mathbb{E}[\Phi_{k+1}] = \mathbb{E}[\Phi_{k,\tau}] + \frac{\mu}{6}\mathbb{E}[\|w_{k+1} - \bar{w}_{k,\tau}\|^2] + \mathbb{E}[F(w_{k+1}^{\text{ag}}) - F(\bar{w}_{k,\tau}^{\text{ag}})]$$

Applying Proposition F.3 and Proposition F.4, we have

$$\mathbb{E}[\Phi_{k+1}]$$

$$\leq (1 - \frac{1}{3}\gamma\mu)^\tau \mathbb{E}[\Phi_k] + (\frac{\eta^2 L}{2} + \frac{\gamma^2\mu}{6})\frac{\tau\sigma^2}{M} + \gamma\tau \cdot \max_{0 \leq t < \tau} \mathbb{E}[\|\nabla F(\bar{w}_{k,t}^{\mathrm{md}}) - \frac{1}{M}\sum_{m=1}^M \nabla F(w_{k,t}^{\mathrm{md},m})\|^2]$$

$$+ \frac{q\mu}{6M^2}\sum_{m=1}^M \mathbb{E}[\|w_{k,\tau}^m - w_k\|^2] + \frac{qL}{2M^2}\sum_{m=1}^M \mathbb{E}[\|w_{k,\tau}^{\mathrm{ag},m} - w_k^{\mathrm{ag}}\|^2]$$

$$\leq (1 - \frac{1}{3}\gamma\mu)^\tau \mathbb{E}[\Phi_k] + (\frac{\eta^2 L}{2} + \frac{\gamma^2\mu}{6})\frac{\tau\sigma^2}{M} + \gamma\tau \cdot \max_{0 \leq t < \tau} \mathbb{E}[\|\nabla F(\bar{w}_{k,t}^{\mathrm{md}}) - \frac{1}{M}\sum_{m=1}^M \nabla F(w_{k,t}^{\mathrm{md},m})\|^2]$$

$$+ \frac{q}{M}\Big[\Big(\gamma^2\mu(\frac{8}{3}\mu + 2L) + 2\gamma^2 L(\frac{\mu}{3} + L)\Big)\tau^2 \mathbb{E}[\Phi_k] + (\frac{\gamma^2\mu}{3} + \eta^2 L)\tau\sigma^2$$

$$+ \Big((\gamma - \eta)^2\gamma^2\mu^2(\frac{\mu}{3} + \frac{L}{4}) + \gamma^4(\frac{\mu}{3} + L)^2 L\Big)\frac{\tau^3\sigma^2}{2}\Big]$$

$$= D(\gamma, \tau)\mathbb{E}[\Phi_k] + (\frac{\eta^2 L}{2} + \frac{\gamma^2\mu}{6})\frac{\tau\sigma^2}{M} + \gamma\tau \cdot \max_{0 \leq t < \tau} \mathbb{E}[\|\nabla F(\bar{w}_{k,t}^{\mathrm{md}}) - \frac{1}{M}\sum_{m=1}^M \nabla F(w_{k,t}^{\mathrm{md},m})\|^2]$$

$$+ \frac{q}{M}(\frac{\gamma^2\mu}{3} + \eta^2 L)\tau\sigma^2 + \frac{q}{2M}\Big((\gamma - \eta)^2\gamma^2\mu^2(\frac{\mu}{3} + \frac{L}{4}) + \gamma^4(\frac{\mu}{3} + L)^2 L\Big)\tau^3\sigma^2$$

The second inequality comes from Proposition F.6. $D(\gamma, \tau)$ is defined as below.

$$D(\gamma, \tau) = (1 - \frac{1}{3}\gamma\mu)^\tau + \frac{q}{M}\Big(\gamma^2\mu(\frac{8}{3}\mu + 2L) + 2\gamma^2 L(\frac{\mu}{3} + L)\Big)\tau^2$$

## F.2 Proof of Theorem F.7

**Theorem F.7.** *Let F be $\mu$-strongly convex, and assume Assumption 2.1, 2.2, 2.3, 2.4, then for the parameter condition set (2), $\tau \geq 2$, if the learning rate $\gamma$ satisfies*

$$\Big(\frac{1}{9}\mu^2 + \frac{q}{M}\Big(\mu(\frac{8}{3}\mu + 2L) + 2L(\frac{\mu}{3} + L)\Big)\Big)\gamma\tau \leq \frac{1}{6}\mu \tag{32}$$

*FedAQ yields*

$$\mathbb{E}[\Phi_K] \leq \exp\Big(-\frac{1}{6}\max(\eta\mu, \sqrt{\frac{\eta\mu}{\tau}})K\tau\Big)\Phi_0 + \frac{2(2q+1)\eta^{\frac{1}{2}}\sigma^2}{\mu^{\frac{1}{2}}M\tau^{\frac{1}{2}}} + \frac{8(q+25)\eta^2 L^2\tau\sigma^2}{\mu}$$

$$+ \frac{3q\Big(\mu^2(\frac{\mu}{3} + \frac{L}{4}) + L(\frac{\mu}{3} + L)^2\Big)\eta^{\frac{3}{2}}\tau^{\frac{1}{2}}\sigma^2}{\mu^{\frac{5}{2}}M} + \frac{3qL(\frac{\mu}{3} + L)^2\eta^3\tau^2\sigma^2}{\mu M}$$

*Proof of Theorem F.7* At first, due to the condition (32) in Theorem F.7, we get

$$D(\gamma, \tau) = (1 - \frac{1}{3}\gamma\mu)^\tau + \frac{q}{M}\Big(\gamma^2\mu(\frac{8}{3}\mu + 2L) + 2\gamma^2 L(\frac{\mu}{3} + L)\Big)\tau^2$$

$$\leq 1 - \frac{1}{3}\gamma\mu\tau + \frac{1}{9}\gamma^2\mu^2\tau^2 + \frac{q}{M}\gamma^2\Big(\mu(\frac{8}{3}\mu + 2L) + 2L(\frac{\mu}{3} + L)\Big)\tau^2$$

$$= 1 - \frac{1}{3}\gamma\mu\tau + \Big(\frac{1}{9}\mu^2 + \frac{q}{M}\Big(\mu(\frac{8}{3}\mu + 2L) + 2L(\frac{\mu}{3} + L)\Big)\Big)\gamma^2\tau^2$$

$$\leq 1 - \frac{1}{6}\gamma\mu\tau \ (\because \text{ condition (32)})$$

It is trivial that $\gamma = \max(\eta, \sqrt{\frac{\eta}{\mu\tau}}) \in [\eta, \sqrt{\frac{\eta}{\mu}}]$. Thus, we can use Lemma F.1. By using Lemma F.1 and the above result, we obtain

$\mathbb{E}[\Phi_{k+1}]$

$$\leq (1 - \frac{1}{6}\gamma\mu\tau)\mathbb{E}[\Phi_k] + (\frac{\eta^2 L}{2} + \frac{\gamma^2\mu}{6})\frac{\tau\sigma^2}{M} + \gamma\tau \cdot \max_{0\leq t<\tau} \mathbb{E}[\|\nabla F(\bar{w}_{k,t}^{\mathrm{md}}) - \frac{1}{M}\sum_{m=1}^{M}\nabla F(w_{k,t}^{\mathrm{md},m})\|^2]$$

$$+ \frac{q}{M}(\frac{\gamma^2\mu}{3} + \eta^2 L)\tau\sigma^2 + \frac{q}{2M}\left((\gamma-\eta)^2\gamma^2\mu^2(\frac{\mu}{3} + \frac{L}{4}) + \gamma^4(\frac{\mu}{3} + L)^2 L\right)\tau^3\sigma^2 \tag{33}$$

By the Lemma C.14 in Yuan and Ma (2020), we know that the below quantity is bounded.

$$\max_{0\leq t<\tau} \mathbb{E}[\|\nabla F(\bar{w}_{k,t}^{\mathrm{md}}) - \frac{1}{M}\sum_{m=1}^{M}\nabla F(w_{k,t}^{\mathrm{md},m})\|^2] \leq B'$$

$$B' = \begin{cases} 4\eta^2 L^2\tau\sigma^2\left(1 + \frac{\gamma^2\mu}{\eta}\right)^{2\tau}, & \text{if } \gamma \in \left(\eta, \sqrt{\frac{\eta}{\mu}}\right] \\ 4\eta^2 L^2\tau\sigma^2, & \text{if } \gamma = \eta \end{cases}$$

Telescoping (33) yields

$$\mathbb{E}[\Phi_K] \leq (1 - \frac{1}{6}\gamma\mu\tau)^K\Phi_0 + \left(\sum_{k'=0}^{K-1}(1 - \frac{1}{6}\gamma\mu\tau)^{k'}\right) \cdot \left[(\frac{\eta^2 L}{2} + \frac{\gamma^2\mu}{6})\frac{\tau\sigma^2}{M} + \frac{q}{M}(\frac{\gamma^2\mu}{3} + \eta^2 L)\tau\sigma^2\right.$$

$$+ \frac{q}{2M}\left((\gamma-\eta)^2\gamma^2\mu^2(\frac{\mu}{3} + \frac{L}{4}) + \gamma^4(\frac{\mu}{3} + L)^2 L\right)\tau^3\sigma^2 + \gamma\tau B'\bigg]$$

$$\leq \exp\left(-\frac{\gamma\mu\tau K}{6}\right)\Phi_0 + \frac{3\eta^2 L\sigma^2}{\gamma\mu M} + \frac{\gamma\sigma^2}{M} + \frac{6B'}{\mu} + 2q\left(\frac{\gamma\sigma^2}{M} + \frac{3\eta^2 L\sigma^2}{\gamma\mu M}\right)$$

$$+ \frac{3q}{M}\left((\gamma-\eta)^2\gamma\mu(\frac{\mu}{3} + \frac{L}{4}) + \frac{\gamma^3(\frac{\mu}{3} + L)^2 L}{\mu}\right)\tau^2\sigma^2$$

The last inequality comes from the fact that $\sum_{k'=0}^{K-1}(1 - \frac{1}{6}\gamma\mu\tau)^{k'} \leq \frac{6}{\gamma\mu\tau}$. Since we plug in $\gamma = \max(\eta, \sqrt{\frac{\eta}{\mu\tau}})$, we can use Lemma C.15 in Yuan and Ma (2020). Therefore, we obtain

$$\mathbb{E}[\Phi_K] \leq \exp\left(-\frac{1}{6}\max(\eta\mu, \sqrt{\frac{\eta\mu}{\tau}})K\tau\right)\Phi_0 + \frac{2(2q+1)\eta^{\frac{1}{2}}\sigma^2}{\mu^{\frac{1}{2}}M\tau^{\frac{1}{2}}} + \frac{4(2q+1)\eta^2 L^2\tau\sigma^2}{\mu}$$

$$+ \frac{24e^2\eta^2 L^2\tau\sigma^2}{\mu} + \frac{3q\tau^2\sigma^2}{M}\max\left(\frac{\eta^{\frac{3}{2}}\mu(\frac{\mu}{3} + \frac{L}{4})}{\mu^{\frac{3}{2}}\tau^{\frac{3}{2}}} + \frac{\eta^{\frac{3}{2}}(\frac{\mu}{3} + L)^2 L}{\mu^{\frac{5}{2}}\tau^{\frac{3}{2}}}, \frac{\eta^3(\frac{\mu}{3} + L)^2 L}{\mu}\right)$$

The first term stems directly from Lemma C.15 in Yuan and Ma (2020). Also, the last term comes from the fact that

$$(\gamma-\eta)^2\gamma\mu(\frac{\mu}{3} + \frac{L}{4}) + \frac{\gamma^3(\frac{\mu}{3} + L)^2 L}{\mu} \leq \begin{cases} \gamma^3\mu(\frac{\mu}{3} + \frac{L}{4}) + \frac{\gamma^3(\frac{\mu}{3}+L)^2 L}{\mu}, & \text{if } \gamma \neq \eta \\ \frac{\eta^3(\frac{\mu}{3}+L)^2 L}{\mu}, & \text{if } \gamma = \eta \end{cases}$$

Therefore, by simple inequalities such as $\max(a, b) \leq a + b$ and $\min(a, b) \leq a$, we ultimately get

$$\mathbb{E}[\Phi_K] \leq \exp\left(-\frac{1}{6}\max(\eta\mu, \sqrt{\frac{\eta\mu}{\tau}})K\tau\right)\Phi_0 + \frac{2(2q+1)\eta^{\frac{1}{2}}\sigma^2}{\mu^{\frac{1}{2}}M\tau^{\frac{1}{2}}} + \frac{8(q+25)\eta^2 L^2\tau\sigma^2}{\mu}$$

$$+ \frac{3q\left(\mu^2(\frac{\mu}{3} + \frac{L}{4}) + L(\frac{\mu}{3} + L)^2\right)\eta^{\frac{3}{2}}\tau^{\frac{1}{2}}\sigma^2}{\mu^{\frac{5}{2}}M} + \frac{3qL(\frac{\mu}{3} + L)^2\eta^3\tau^2\sigma^2}{\mu M} \tag{34}$$

### F.3  Proof of Corollary F.8

**Corollary F.8.** *Let $D_1, D_2$, and $\eta_0$ as below. Note that $T = K\tau$.*

$$D_1 = \frac{\left(\mu^2(\frac{\mu}{3} + \frac{L}{4}) + L(\frac{\mu}{3} + L)^2)\right)q}{\mu^{\frac{5}{2}}}, \quad D_2 = \frac{q(\frac{\mu}{3} + L)^2 L}{\mu}$$

$$\eta_0 = \frac{36\tau}{\mu T^2} \log^2 \left( e + \min(\frac{\mu MT\Phi_0}{(2q+1)\sigma^2}, \frac{\mu^3 T^4 \Phi_0}{(q+25)L^2\tau^3\sigma^2}, \frac{\mu^3 MT^3 \Phi_0}{(\mu^{\frac{3}{2}}D_1 + 6^3 D_2)\tau^2\sigma^2}) \right)$$

*Then for $\eta = \min(\frac{1}{L}, \eta_0)$, FedAQ yields*

$$\mathbb{E}[\Phi_K] \le \min \left( \exp(-\frac{\mu T}{6L}), \exp(-\frac{\mu^{\frac{1}{2}}T}{6L^{\frac{1}{2}}\tau^{\frac{1}{2}}}) \right)\Phi_0$$

$$+ \frac{13(2q+1)\sigma^2}{\mu MT} \log^2 \left( e + \frac{\mu MT\Phi_0}{(2q+1)\sigma^2} \right) \tag{35}$$

$$+ \frac{10369(q+25)L^2\tau^3\sigma^2}{\mu^3 T^4} \log^4 \left( e + \frac{\mu^3 T^4 \Phi_0}{(q+25)L^2\tau^3\sigma^2} \right) \tag{36}$$

$$+ \frac{649(\mu^{\frac{3}{2}}D_1 + 216D_2)\tau^2\sigma^2}{\mu^3 MT^3} \log^6 \left( e + \frac{\mu^3 MT^3 \Phi_0}{(\mu^{\frac{3}{2}}D_1 + 216D_2)\tau^2\sigma^2} \right) \tag{37}$$

*Proof of Corollary F.8*  Let's decompose the final result (34) of the Theorem F.7 into a decreasing term and an increasing term. We denote the decreasing term $\phi_1$ and the increasing term $\phi_2$ as below.

$$\phi_1(\eta) = \exp \left( -\frac{1}{6} \max(\eta\mu, \sqrt{\frac{\eta\mu}{\tau}})T \right)\Phi_0$$

$$\phi_2(\eta) = \frac{2(2q+1)\eta^{\frac{1}{2}}\sigma^2}{\mu^{\frac{1}{2}}M\tau^{\frac{1}{2}}} + \frac{8(q+25)\eta^2 L^2\tau\sigma^2}{\mu} + \frac{3q\left(\mu^2(\frac{\mu}{3} + \frac{L}{4}) + L(\frac{\mu}{3} + L)^2\right)\eta^{\frac{3}{2}}\tau^{\frac{1}{2}}\sigma^2}{\mu^{\frac{5}{2}}M}$$

$$+ \frac{3qL(\frac{\mu}{3} + L)^2\eta^3\tau^2\sigma^2}{\mu M}$$

Since $\phi_1$ is the decreasing term, we have

$$\phi_1(\eta) \le \phi_1(\frac{1}{L}) + \phi_1(\eta_0) \tag{38}$$

where

$$\phi_1(\frac{1}{L}) = \min \left( \exp(-\frac{\mu T}{6L}), \exp(-\frac{\mu^{\frac{1}{2}}T}{6L^{\frac{1}{2}}\tau^{\frac{1}{2}}}) \right)\Phi_0$$

$$\phi_1(\eta_0) \le \exp \left( -\frac{1}{6}\sqrt{\frac{\eta_0\mu}{\tau}}T \right)$$

$$= \left( e + \min(\frac{\mu MT\Phi_0}{(2q+1)\sigma^2}, \frac{\mu^3 T^4 \Phi_0}{(q+25)L^2\tau^3\sigma^2}, \frac{\mu^3 MT^3 \Phi_0}{(\mu^{\frac{3}{2}}D_1 + 6^3 D_2)\tau^2\sigma^2}) \right)^{-1}\Phi_0$$

$$\le \frac{(2q+1)\sigma^2}{\mu MT} + \frac{(q+25)L^2\tau^3\sigma^2}{\mu^3 T^4} + \frac{(\mu^{\frac{3}{2}}D_1 + 6^3 D_2)\tau^2\sigma^2}{\mu^3 MT^3}$$

Since $\phi_2$ is the increasing term, we have

$\phi_2(\eta)$

$\leq \phi_2(\eta_0)$

$\leq \dfrac{12(2q+1)\sigma^2}{\mu MT} \log\left(e + \dfrac{\mu MT\Phi_0}{(2q+1)\sigma^2}\right) + \dfrac{8 \cdot 36^2(q+25)L^2\tau^3\sigma^2}{\mu^3 T^4} \log^4\left(e + \dfrac{\mu^3 T^4\Phi_0}{(q+25)L^2\tau^3\sigma^2}\right)$

$\quad + \dfrac{3 \cdot 6^3 D_1\tau^2\sigma^2}{\mu^{\frac{3}{2}} MT^3} \log^3\left(e + \dfrac{\mu^3 MT^3\Phi_0}{(\mu^{\frac{3}{2}}D_1 + 6^3 D_2)\tau^2\sigma^2}\right)$

$\quad + \dfrac{3 \cdot 36^3 D_2\tau^5\sigma^2}{\mu^3 MT^6} \log^6\left(e + \dfrac{\mu^3 MT^3\Phi_0}{(\mu^{\frac{3}{2}}D_1 + 6^3 D_2)\tau^2\sigma^2}\right)$

$\leq \dfrac{12(2q+1)\sigma^2}{\mu MT} \log\left(e + \dfrac{\mu MT\Phi_0}{(2q+1)\sigma^2}\right) + \dfrac{8 \cdot 36^2(q+25)L^2\tau^3\sigma^2}{\mu^3 T^4} \log^4\left(e + \dfrac{\mu^3 T^4\Phi_0}{(q+25)L^2\tau^3\sigma^2}\right)$

$\quad + \dfrac{3 \cdot 6^3(\mu^{\frac{3}{2}}D_1 + 6^3 D_2)\tau^2\sigma^2}{\mu^3 MT^3} \log^6\left(e + \dfrac{\mu^3 MT^3\Phi_0}{(\mu^{\frac{3}{2}}D_1 + 6^3 D_2)\tau^2\sigma^2}\right)$ \hfill (39)

The last inequality comes from $\frac{\tau}{T} \leq 1$. Therefore, by combining (38) and (39), we finally get

$\mathbb{E}[\Phi_K] \leq \phi_1(\eta) + \phi_2(\eta)$

$\quad\quad \leq \phi_1(\tfrac{1}{L}) + \phi_1(\eta_0) + \phi_2(\eta_0)$

$\quad\quad \leq \min\left(\exp(-\dfrac{\mu T}{6L}), \exp(-\dfrac{\mu^{\frac{1}{2}}T}{6L^{\frac{1}{2}}\tau^{\frac{1}{2}}})\right)\Phi_0 + \dfrac{13(2q+1)\sigma^2}{\mu MT} \log^2\left(e + \dfrac{\mu MT\Phi_0}{(2q+1)\sigma^2}\right)$

$\quad\quad\quad + \dfrac{10369(q+25)L^2\tau^3\sigma^2}{\mu^3 T^4} \log^4\left(e + \dfrac{\mu^3 T^4\Phi_0}{(q+25)L^2\tau^3\sigma^2}\right)$

$\quad\quad\quad + \dfrac{649(\mu^{\frac{3}{2}}D_1 + 216 D_2)\tau^2\sigma^2}{\mu^3 MT^3} \log^6\left(e + \dfrac{\mu^3 MT^3\Phi_0}{(\mu^{\frac{3}{2}}D_1 + 216 D_2)\tau^2\sigma^2}\right)$

### F.4 Complexity of combining the quantization and multiple accelerated local updates

Our theoretical goal is to show that the convergence rate of FedAQ is just linearly slower than the convergence rate of FedAC, and FedAQ achieves the best convergence rate among quantization-based federated algorithms. Thus, we should find the tight upper bound of the emerging terms due to quantization so that FedAQ fully takes advantage of acceleration and achieves our goal. To be specific, since a server aggregates two quantized local updates $Q(w^m_{k,\tau} - w_k), Q(w^{\mathrm{ag},m}_{k,\tau} - w^{\mathrm{ag}}_k)$ from all clients, the additional error from variances of the unbiased quantizer $Q$ on two local updates $w^m_{k,\tau} - w_k, w^{\mathrm{ag},m}_{k,\tau} - w^{\mathrm{ag}}_k$ should be well-bounded (See Proposition F.4). This is where the difficulty of the theory of applying quantization to FedAC stems. The other quantization-based federated algorithms only care about one quantized local update, but we need to consider two quantized local updates that amplify the instability of FedAQ. In spite of this challenge, we obtain the tight upper bound of the error from $w^m_{k,\tau} - w_k, w^{\mathrm{ag},m}_{k,\tau} - w^{\mathrm{ag}}_k$ with the insight that two local parameters $w^m_{k,t}$, $w^{\mathrm{ag},m}_{k,t}$ become closer as $k, t$ increase and both parameters converge to $w^*$ (See Proposition F.5, F.6, the definition of $B^m_{k,t}$ in Appx. F).

## G Discussion

To sum up, we propose a novel communication-efficient federated optimization algorithm, FedAQ, that successfully incorporates accelerated multiple local updates and quantization with solid theoretical guarantees. We achieve the best convergence rate and the smallest number of communication rounds required for a linear speedup in $M$ in strongly-convex and homogeneous settings. In the future, further theoretical guarantees of FedAQ on convex and non-convex functions should be discussed. Also, the convergence analysis of FedAQ on heterogeneous settings can be an interesting topic. Even though Federated Learning systems provide some level of privacy to the clients as their explicit data is not shared with the servers, careful examination of FL systems including FedAQ is necessary to

examine how much privacy do they actually provide as information is shared in form of the iterates. More interesting and challenging open problems can be found in Kairouz et al. (2019); Wang et al. (2021a).

