# OpenReview forum: "Accelerated Federated Optimization with Quantization"
_NeurIPS.cc/2022/Workshop/Federated_Learning — FL-NeurIPS 2022 Oral_

### Official Review · Reviewer_qjEQ · 2022-09-30
**Good paper relevant to the workshop**

This paper studies local methods for federated learning and proposes an accelerated local method with quantization. The authors prove that the method achieves a linear speed-up in the number of workers after $O(M^\frac{1}{3})$ iterations, which matches the bound of FedAC but has a quantization operation on top of it. The authors also compare the proposed algorithm (FedAQ) on 1) logistic regression with MNIST data, 2) MLP on MNIST, 3) CNN on cifar-10. The study is rigorous, the paper is well written and I think the method would be of interest to some attendees of the workshop.

I think this paper is a good match for the workshop and should be accepted.

---

### Official Review · Reviewer_3oUq · 2022-10-18
**This paper proposes a novel algorithm that combine the FedAC and QSGD works together, with theoretical guarantees. The author provide a solid proof for its convergence based on strongly-convex and homogeneous local data distribution, but it’s not enough to understand its property especially on non-iid situations on FL.**

1.	Novelty, relevance, significance
    The novelty is limited. The author combine an acceleration techniques using FedAC with quantization method using QSGD.

2.	Soundness
The proof  mainly follows FedAC, except for the quantization part. I have checked most of its proof.


3.Summary
By combining acceleration technique and quantization technique, the authors propose a novel algorithm and provide theoretical guanrantee with strongly-convex and iid situation.
Strong points:
(1)The proof is solid.
(2)It successfully combines acceleration and quantitation techniques together.

Weak points:
(1) The novelty is limited. Its seems simply to employ QSGD to FedAC.
(2) Lack of considering more complex situations,  such as non-iid, non-convex in convergence analysis.

---

### Official Review · Reviewer_yG4m · 2022-10-18
**I think paper's contribution is sufficient for a workshop.**

This paper considers the communication-efficiency aspect of Federated Learning. Authors try to extend to add quantization element to acceleration, thus extending the results in [Yuan, H. and Ma, T. (2020)]. I think this paper opens up interesting questions in Federated optiimzation. Therefore, I recommend acceptance for this paper.

---

### Decision · Program_Chairs · 2022-10-20

Accept (Oral)